# Hypoxia induces histone clipping and H3K4me3 loss in neutrophil progenitors resulting in long-term impairment of neutrophil immunity

**A list of authors and their affiliations appears at the end of the paper**

The long-term impact of systemic hypoxia resulting from acute respiratory distress syndrome (ARDS) on the function of short-lived innate immune cells is unclear. We show that patients 3–6 months after recovering from ARDS have persistently impaired circulating neutrophil effector functions and an increased susceptibility to secondary infections. These defects are linked to a widespread loss of the activating histone mark H3K4me3 in genes that are crucial for neutrophil activities. By studying healthy volunteers exposed to altitude-induced hypoxemia, we demonstrate that oxygen deprivation alone causes this long-term neutrophil reprogramming. Mechanistically, mouse models of systemic hypoxia reveal that persistent loss of H3K4me3 originates in proNeu and preNeu progenitors within the bone marrow and is linked to N-terminal histone 3 clipping, which removes the lysine residue for methylation. Thus, we present new evidence that systemic hypoxia initiates a sustained maladaptive reprogramming of neutrophil immunity by triggering histone 3 clipping and H3K4me3 loss in neutrophil progenitors.

Hypoxemia is a feature of acute respiratory distress syndrome (ARDS), with neutrophils contributing to the vascular and alveolar epithelial injury that together result in the progression to respiratory failure[1–3]. Extending beyond local effects in the lung, we and others have observed altered circulating neutrophil frequency, phenotype and functions at time of presentation in patients with ARDS[4–6]. Together with emerging evidence in murine models of hypoxic acute lung injury that exposure to systemic hypoxia alone can acutely shape the circulating myeloid cell response through the regulation of myelopoiesis[7], this supports the concept that acute bone-marrow (central) reprogramming by hypoxia can inform neutrophil effector functions in the tissue. Besides changes in emergency hematopoiesis in response to systemic hypoxia, a bias toward myelopoiesis has been reported in the hematopoietic stem cell progenitor compartment in response to Bacillus Calmette–Guérin (BCG) vaccination[8]. This transcriptional response is associated with long-term functional changes in the circulating neutrophil compartment[9]. Moreover, in murine model systems, β-glucan-mediated training of granulopoiesis has been implicated in neutrophil reprogramming toward an antitumor response[10], and in disease tolerance to influenza A[11].

Changes in chromatin accessibility and histone methylation marks have been observed in both the granulocyte monocyte progenitor and circulating blood neutrophil compartments and consequently implicated in this innate immune training. The high turnover of circulating neutrophils ensures modifications within the bone marrow are rapidly manifest in the blood neutrophil pool. In peripheral tissues, hypoxia promotes hyperinflammatory neutrophil responses that augment tissue injury[12], resulting in impaired disease tolerance defined by the functional loss of tissue integrity. It remains unknown whether in human disease states following a hypoxic injurious insult central reprogramming mechanisms can be engaged to try to overcome these damaging responses by promoting disease tolerance. The long-term consequence of centrally altering the balance between responses aimed at evading or neutralizing pathogens (host resistance) and

✉e-mail: sarah.walmsley@ed.ac.uk

those enabling disease tolerance[13] also warrants investigation given its potential to sustain maladaptive reprogramming of the circulating neutrophil compartment.

Here, we show that patients who survive an admission with ARDS have sustained changes in neutrophil phenotype and function 3–6 months following recovery. These long-term neutrophil perturbations are associated with a widespread loss of histone 3 lysine 4 trimethylation (H3K4me3) and impaired infection outcomes. Longitudinal studies of healthy human volunteers exposed to altitude-induced hypoxemia and mice subject to low oxygen levels reveal that hypoxia alone is sufficient to trigger these neutrophil defects. Notably, we uncover a central re-shaping of the bone-marrow epigenetic landscape where hypoxia induces N-terminal histone 3 (H3) clipping linked to H3K4me3 downregulation in neutrophil-committed progenitors providing a mechanism for persistent neutrophil reprogramming.

## Results

### Impaired neutrophil function months after ARDS recovery

To test whether an episode of ARDS can trigger long-term changes in the neutrophil compartment, we studied circulating neutrophils from healthy donors and ARDS survivors 3–6 months following admission with Berlin criteria-defined moderate–severe ARDS[14] (Supplementary Table 1). Neutrophil phenotype and effector functions were assessed during steady-state and upon activation (Fig. 1a). Flow cytometry analysis of freshly isolated neutrophils showed that ARDS survivors retain high neutrophil blood counts but do not show the expansion of the immature neutrophil compartment associated with the acute disease state reported in the literature (Fig. 1b and Extended Data Fig. 1a)[6]. Surface analysis of the marker CD66b showed modest elevation in abundance with a concomitant decreased presence of the adhesion molecule CD62L (Fig. 1c,d), indicating anomalous neutrophil activation in unstimulated conditions. Neutrophil activities are closely connected to their metabolic status, which determines energy availability and metabolite supply for enzymatic reactions. Neutrophils also rely on dynamic rearrangement of their proteome to facilitate these end-effector functions[15]. Therefore, we undertook a targeted analysis following liquid chromatography–mass spectrometry (LC–MS) of blood neutrophil metabolome and proteome to expand on the characterization of this abnormal phenotypic status. Our data unmasked perturbations in central metabolic programs illustrated by diminished abundance of pyruvate and lactate in combination with a persistent increase of acetyl-CoA (Fig. 1e) in survivors of ARDS. These metabolic changes were not associated with alterations in tricarboxylic acid cycle intermediaries or amino acids such as methionine and glutamate, and did not deprive neutrophils energetically (Extended Data Fig. 1b–d). The parallel proteomic survey of more than 4,000 proteins (Extended Data Fig. 1e) revealed system-wide differences following ARDS (Extended Data Fig. 1f). While we observed conservation of global cellular processes evidenced by equivalent mitochondrial, nuclear envelope-related, ribosomal and eukaryotic initiation factor 4F protein content (Extended Data Fig. 1g), we detected a modest increase in cytoskeletal protein content accompanied by a significant contraction of the azurophilic granule cargo compartment (Fig. 1f and Extended Data Fig. 1h).

Notably, these changes in neutrophil activation, metabolite and protein abundance were associated with alterations in neutrophil effector functions upon challenge. Neutrophils isolated from patients 3–6 months following ARDS demonstrated reduced degranulation of α-1-antitrypsin when cultured ex vivo (Fig. 1g). ARDS also triggered long-term enhanced lipopolysaccharide (LPS) neutrophil survival and an impairment in opsonic phagocytosis of *Staphylococcus aureus* SH1000 (Fig. 1h,i). These abnormal innate immune antimicrobial responses occurred alongside a high prevalence of patients with secondary infections evidenced by positive microbiological samples following discharge from the critical care unit (Fig. 1j). Thus, we provide evidence of impaired neutrophil antimicrobial capacity (loss of granule protein expression and secretion, reduced opsonic phagocytosis) with increased susceptibility to secondary bacterial infection in individuals 3–6 months following admission with ARDS.

### Metabolic rewiring of neutrophils during ARDS presentation

Given the long-term consequences of ARDS for neutrophil biology detected in survivors, we decided to explore changes in neutrophils during the acute phase of ARDS that could link to a persistent reprogramming of the neutrophil compartment (Fig. 2a). In line with previous work from our group and others[6,16,17], data from patients involved in long-term studies showed preserved circulating total leukocyte counts but an accentuated elevation of the neutrophil pool at the time of presentation with ARDS (Fig. 2b). These patients also demonstrated marked hypoxemia[18] despite high levels of oxygen supplementation, in keeping with ventilatory failure (Fig. 2c).

Persistent modification of the neutrophil compartment with an epigenetic basis has been shown after administration of training agents such as BCG vaccine or β-glucan[9,10]. The epigenetic status is highly interconnected with metabolic programs and therefore, oxygen availability[19]. Therefore, we analyzed metabolic and proteomic datasets (accession number PXD023834) obtained from highly pure normal-density neutrophils from the blood of patients presenting with ARDS in previous studies[6] to interrogate metabolic pathways and enzymes relevant for epigenetic processes, including acetylation and methylation (Fig. 2d,e). Despite neutrophils heavily relying on glycolysis[20], a switch to enable the utilization of fatty acids was manifested in ARDS neutrophils. This was indicated by elevated copy number of enzymes and transporters required for mitochondrial lipid entry and β-oxidation (ACSL1, CPT1A, SLC25A20, ACAA2 and HADH) when compared to healthy control neutrophils, and an associated increase in acetyl-CoA abundance (Fig. 2d), an essential metabolite for histone acetylation. ARDS neutrophils also displayed an enhanced capacity to generate metabolic intermediaries important for the addition of methylation marks through the increase in one-carbon metabolism (Fig. 2e). This was evidenced by an uplift in the copy number of SHMT2 and abundance of methionine accompanied by increased levels of the enzyme MAT2A (Fig. 2e).

We next questioned whether ARDS-driven proteomic changes in neutrophils could include changes in epigenetic enzymes and complexes. We detected an expanded expression of catalytic (HDAC2) and accessory (MTA2 and RBBP4) components of the NuRD chromatin remodeling complex, which regulates chromatin acetylation[21] (Extended Data Fig. 2a). ARDS also affected the abundance of proteins regulating H3K4 methylation states, with diminished levels of both the H3K4 methyltransferase COMPASS complex components ASH2L and RBBP5 (Extended Data Fig. 2b). Taken together, these results reveal that during acute presentation with ARDS circulating neutrophils display changes in epigenetic writers, erasers and metabolic intermediaries that have the capacity to influence chromatin accessibility.

### Loss of H3K4me3 in blood neutrophils 3–6 months post-ARDS

Human BCG vaccination studies have previously reported that neutrophil antimicrobial activities are influenced by H3K4me3 abundance[9]. Given the observed changes in the histone methylation machinery during ARDS (Extended Data Fig. 2b) we therefore questioned whether long-term perturbations in neutrophil effector function were linked to changes in H3K4me3. Chromatin immunoprecipitation sequencing (ChIP-seq) analysis of the blood neutrophil pool of ARDS survivors 3–6 months post-hospitalization revealed inter-individual heterogeneity in the distribution of H3K4me3 marks (Fig. 3a). Despite this heterogeneity, we observed extensive loss of H3K4me3 across the gene body irrespective of treatment groups (Fig. 3b,c and Extended Data Fig. 3a–c). Gene Ontology (GO) enrichment analysis subsequently revealed neutrophil degranulation to be the top ranked

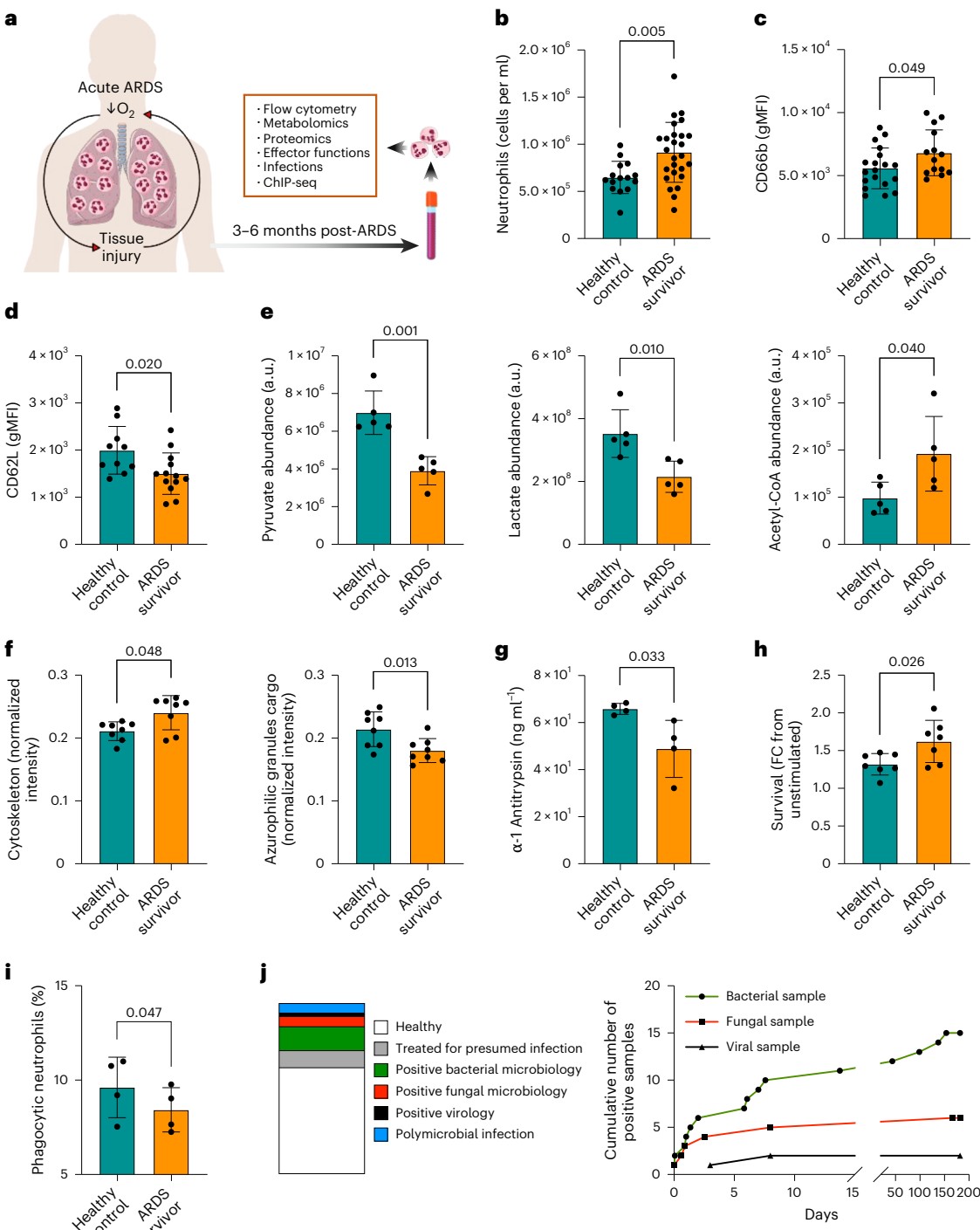

**Fig. 1 | ARDS induces long-term alterations of neutrophil phenotypes and functions. a–i**, Phenotypic and functional analysis of circulating neutrophils collected from healthy controls and survivors of ARDS 3–6 months post-hospital admission. **a**, Schematic representation of the analyses performed. Human silhouette adapted from Wikipedia commons (https://commons.wikimedia.org/wiki/File:Man_shadow_-_upper.png); lungs, neutrophils and blood tube adapted from Servier license under CC-BY 3.0 Unported. **b**, Circulating neutrophil counts obtained by flow cytometry (*n* = 19 healthy control and *n* = 26 ARDS survivor). **c,d**, Surface expression of the neutrophil activation markers CD66b and CD62L measured by flow cytometry (*n* = 15 healthy control and *n* = 14 ARDS survivor (**c**), *n* = 10 healthy control and *n* = 13 ARDS survivor (**d**)). **e**, Metabolite abundance of pyruvate, lactate and acetyl-CoA obtained by LC–MS analysis (*n* = 5 for both experimental groups). **f**, Proteomic analysis by LC–MS showing abundance of cytoskeletal (GO:0005856) and azurophilic granule cargo proteins (GO:0035578) (*n* = 8 for both experimental groups).

**g**, Ex vivo quantification of α-1-antitrypsin from neutrophil culture supernatants performed by ELISA (*n* = 4 for both experimental groups, two technical replicates per sample). **h**, Ex vivo neutrophil survival in response to LPS stimulation after 20 h of culture evaluated by microscopy analysis (*n* = 7 for both experimental groups, two technical replicates per sample). **i**, Phagocytic capacity of opsonized *S. aureus* SH1000 measured by flow cytometry as a percentage of the total neutrophil population analyzed (*n* = 4 for both experimental groups, two technical replicates per sample). **j**, Infections recorded in ARDS survivors, with their infection etiologies expressed as proportion or as cumulative positive microbiology results over the course of 6 months post-ARDS. **b–i**, Data show mean ± s.d., with each value representing an individual. Significant *P* values depicted (for *P* < 0.05) and obtained by Shapiro–Wilk normality test followed by a two-tailed *t*-test (**b–i**) or two-tailed Mann–Whitney *U*-test (**c**, cytoskeleton (**f**)). FC, fold change; gMFI, geometric mean fluorescence intensity; a.u., arbitrary units.

pathway to display differential levels of H3K4me3. Other pathways identified to show differential expression profiles related to core neutrophil processes, including apoptosis regulation and response to viral and bacterial infections (Fig. 3d). A closer inspection of tracks of individual genes relevant to neutrophil functional processes such as granule proteins and maturation (*LYZ*, *SERPINA1*, *RAB3D* and *FOS*) demonstrated a profound loss of H3K4me3 levels in these genes (Fig. 3e). Similarly, important genes involved in calcium signaling (such as *CALM1-3* and *CALR*) and inflammatory response genes (including *IFNAR2*, *IRF7*, *CXCR2* or *TNFRSF1A*) also showed a substantial downregulation of H3K4me3 abundance in circulating neutrophils months after ARDS (Fig. 3f,g and Extended Data Fig. 3d).

Collectively, these results demonstrate an altered H3K4me3 epigenomic profile in genes relevant for core neutrophil activities triggered by ARDS that could translate into functional innate immune memory.

## Systemic hypoxia mirrors long-term ARDS neutrophil defects

To dissect the contribution of systemic hypoxia to the persistent reprogramming of neutrophil responses, we undertook a parallel longitudinal study in a cohort of healthy volunteers exposed to a 7-day period of altitude-induced hypoxemia (Fig. 4a). Peripheral blood neutrophils were isolated from individuals at baseline and 3–4 months following recovery from a substantial acute hypoxic exposure (Fig. 4b). Although the total neutrophil number in the circulation post-altitude was equivalent to baseline conditions (Fig. 4c), surface expression of the activation markers CD66b and CD62L followed their post-ARDS profile, namely an increment of CD66b abundance with concomitant reduction of CD62L in unstimulated conditions (Fig. 4d).

An equivalent proteomic approach to the one employed post-ARDS was used to compare circulating neutrophil protein abundance at baseline and post-altitude (Extended Data Fig. 4a–c). In keeping with observations in ARDS survivors, cells collected from individuals 3–4 months after altitude-induced hypoxemia demonstrated an uplift in cytoskeleton-related proteins with a concomitant contraction in granule cargo proteins (Fig. 4e). This was accompanied by impaired ex vivo release of myeloperoxidase (Fig. 4f) and decreased phagocytic activity (Fig. 4g). These results place the hypoxic component of ARDS as a key regulator of neutrophil phenotype and performance in the long-term.

To test whether systemic hypoxia was sufficient to impart the epigenetic signature observed in neutrophils months after ARDS, we undertook H3K4me3 ChIP–qPCR analysis of core neutrophil effector genes. We observed hypoxia-driven loss of H3K4me3 in genes that are notable for granule proteins and maturation (*LYZ* and *RAB3D*), calcium signaling (*CALM1* and *CALR*) and inflammatory responses (*IFNAR2* and *CXCR2*) (Fig. 4h–j). This supports our previous ChIP-seq findings and assigns a central role to hypoxia in the epigenetic reprogramming of human neutrophils.

## Loss of H3K4me3 in progenitors impairs neutrophil immunity

To address the mechanisms driving sustained changes in H3K4me3 levels in response to hypoxia, we used murine models of systemic

hypoxia and hypoxic lung injury. We first validated hypoxia-mediated neutrophil perturbations in host defense[22] in a *S. aureus* SH1000 skin infection model. In keeping with observations in ARDS survivors[23] (Fig. 1j), exposure of mice to 7 days of systemic hypoxia followed by a 5-week reoxygenation period before bacterial challenge (Fig. 5a) led to an impaired antimicrobial response with increased abscess size and bacterial counts following infection (Fig. 5b,c), despite equivalent circulating neutrophil abundance (Extended Data Fig. 5a). Next, we used a hypoxic lung injury recovery mouse model by employing LPS nebulization followed by a 1-week hypoxic exposure and a 5-week period of normoxia (Fig. 5d) to explore the consequence of hypoxic exposure on the abundance of H3K4me3 in mature bone-marrow neutrophils by immunoblot. Our results confirmed a drop in H3K4me3 levels in this mouse model (Fig. 5e), in keeping with the ARDS survivors' ChIP-seq epigenetic profile (Fig. 3b).

Previous reports have demonstrated a myelopoiesis bias with granulocyte monocyte progenitor (GMP) expansion and changes in chromatin accessibility as mechanisms underlying long-term neutrophil reprogramming in response to agents like β-glucan or chronic inflammation[10,24]. Therefore, we postulated that mouse bone-marrow GMPs could be the precursor population transmitting a persistent reduction in H3K4me3 to mature neutrophils in response to hypoxia (Fig. 5f). To test this hypothesis, we performed Cut&Run sequencing analysis of the FACS-isolated bone-marrow GMP compartment from mice experiencing 1 week of hypoxia plus 5 weeks recovery in normoxia (Fig. 5d). Of note, inspection of individual tracks of genes showing H3K4me3 loss in response to hypoxia in human neutrophils (Fig. 4h–j) did not display a hypoxia-mediated lasting epigenetic change in GMPs for this histone mark (Fig. 5g). Additional LPS challenge in combination with hypoxia did not alter H3K4me3 profiles in these genes and the frequencies of both GMP and common myeloid progenitor (CMP) populations were also unchanged (Extended Data Fig. 5b). Thus, we hypothesized alternative proliferating progenitor populations further across the neutrophil lineage could be responsible for a hypoxia-driven lasting reduction in H3K4me3. Further neutrophil-committed bone-marrow progenitor populations with self-renewal capacity have been reported in the literature in recent years, including proNeu1, proNeu2 and preNeu[25] (Fig. 5f). A flow cytometry-based approach was used to identify these populations in mouse bone marrow 3 months after a 1-week hypoxic exposure to evaluate global levels of H3K4me3 and H3 through intracellular staining. Although the abundance of these populations was equivalent to normoxia control mice (Extended Data Fig. 5c–e), a significant reduction in H3K4me3 signal was detected in proNeu1, proNeu2 and preNeu months following a single hypoxic exposure (Fig. 5h–j). These findings place neutrophil-committed progenitors at the basis of the long-term epigenetic reprogramming observed in circulating cells.

## N-terminal clipping in progenitors drives H3K4me3 loss

Early studies have highlighted a role for histone clipping in cell development and specification, including differentiation of mast

---

**Fig. 2 | Circulating neutrophils during ARDS show metabolic rewiring that can influence epigenetic reactions. a**, Diagram of the studies performed in circulating neutrophils from patients during the acute phase of ARDS and healthy control samples. Human silhouette adapted from Wikimedia commons (https://commons.wikimedia.org/wiki/File:Man_shadow_-_upper.png); lungs, neutrophils and blood tube adapted from Servier license under CC-BY 3.0 Unported. **b**, Circulating white cell and neutrophil counts during the acute phase of ARDS obtained from ARDS survivors in Fig. 1 (*n* = 43). **c**, Lowest partial pressures of oxygen (*P*aO$_2$) and highest fraction of inspired oxygen (*F*iO$_2$) from clinical arterial blood samples during the acute phase of ARDS obtained from ARDS survivors in Fig. 1 (shadowed area depicts normal *P*aO$_2$ range and dashed line reflects 21% O$_2$) (*n* = 24). **d,e**, Abundance of proteins and metabolites measured in circulating neutrophils from healthy control and acute

ARDS patients by LC–MS analysis obtained from previous datasets[6]. **d**, Schema of relevant steps involved in fatty acid transport and oxidation, with dashed lines summarizing multiple enzymatic steps alongside quantification of the abundance of proteins and metabolites (*n* = 4 healthy control and *n* = 11 ARDS for ACSL1, CPT1A, SLC25A20, ACAA2 and HADH, *n* = 5 healthy control and *n* = 6 ARDS for Acetyl-CoA). **e**, Schema of proteins and metabolites linked to one-carbon metabolism pathway alongside quantification of their abundance (*n* = 4 healthy control and *n* = 11 ARDS for SHMT2 and MAT2A, *n* = 5 healthy control and *n* = 15 ARDS for methionine). Data as mean ± s.d., with each value representing an individual. Significant *P* values depicted (for *P* < 0.05) and obtained by Shapiro–Wilk normality test followed by two-tailed *t*-test (ACSL1, Acetyl-CoA, SHMT1, MAT2A and methionine (**d**,**e**)) or two-tailed Mann–Whitney *U*-test (CPT1A, SLC25A20, ACAA2 and HADH (**d**,**e**)).

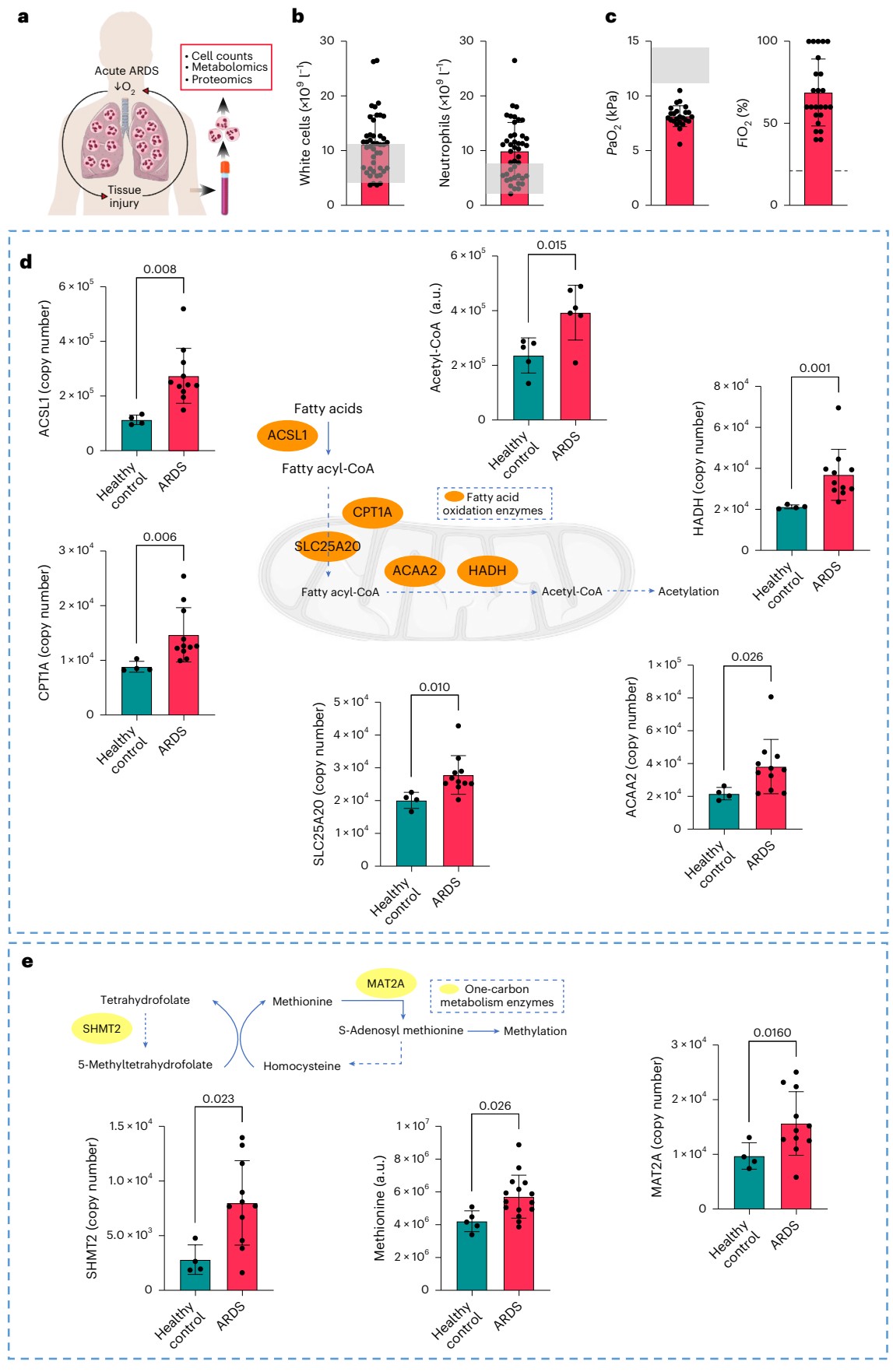

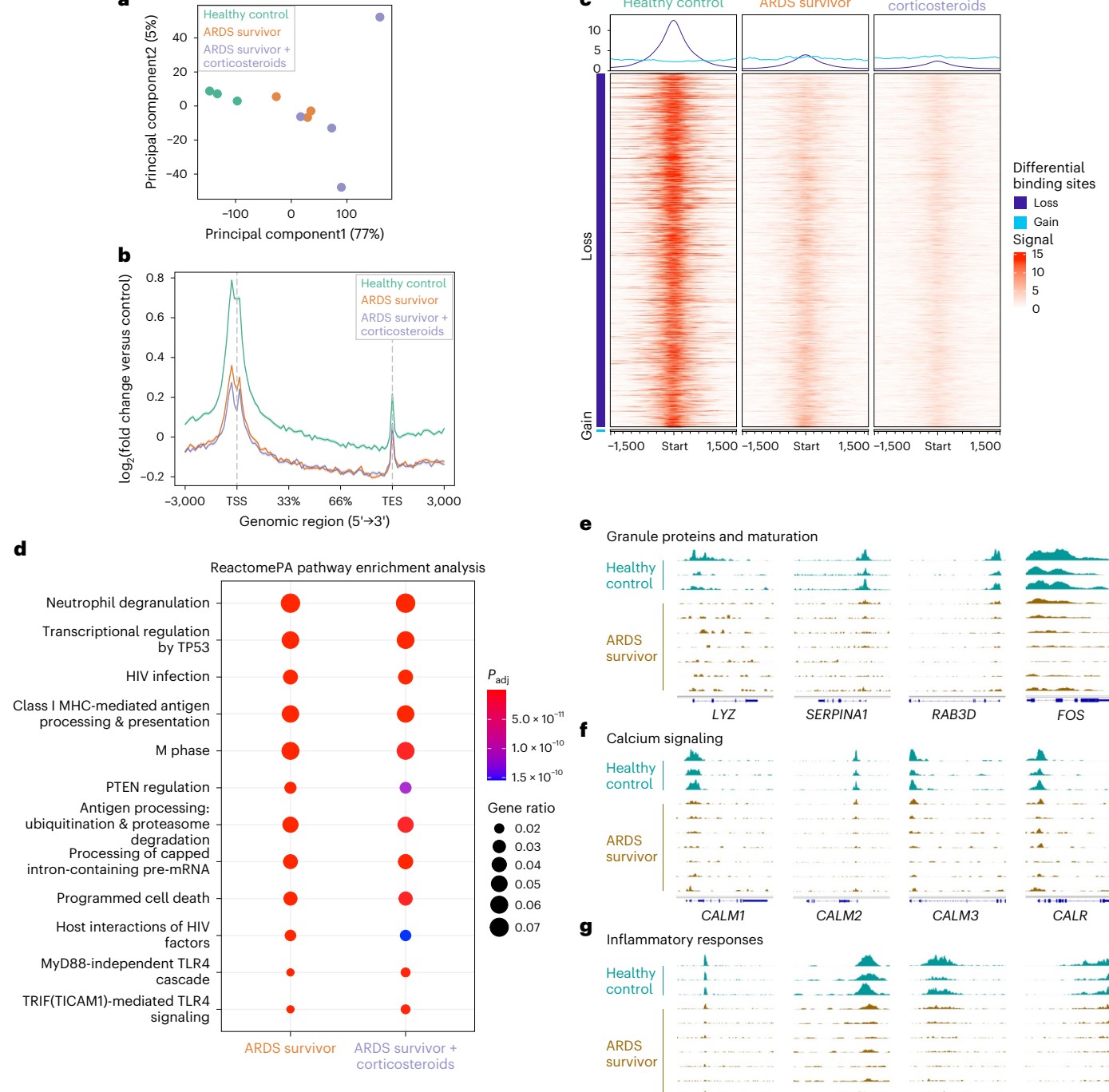

**Fig. 3 | Genes involved in core functions in circulating neutrophils express diminished H3K4me3 levels months following ARDS. a–g,** Chromatin immunoprecipitation sequencing analysis of genomic regions associated with H3K4me3 in healthy donors' blood neutrophils compared to those from patients surviving ARDS 3–6 months post-hospital admission depicting corticosteroid treatment. **a,** Principal-component analysis of the three sample groups with each dot representing an individual. **b,** Metagene H3K4me3 profiles 3-kb upstream from the transcription start site (TSS) and downstream from the transcription end site (TES) of the three sample groups, with the s.e.m. shown as shading around the mean curve. Data are expressed as log₂FC versus input control levels.

**c,** Differential peaks obtained through the R package Diffbind when comparing the three sample groups. **d,** GO analysis by Reactome pathway enrichment in the ARDS survivors' groups versus healthy controls. **e–g,** Individual gene tracks for genes involved in granule proteins and maturation (**e**), calcium signaling (**f**) and inflammatory responses (**g**) depicting H3K4me3 profiles for healthy controls and ARDS survivors. The numbers of biological replicates were healthy controls ($n = 3$), ARDS survivors who did not receive corticosteroid treatment ($n = 3$) and corticosteroid-treated ARDS survivors ($n = 4$). The false discovery rate was applied to generate **d**. Details on normalization and quality control analysis in Extended Data Fig. 3 and Methods.

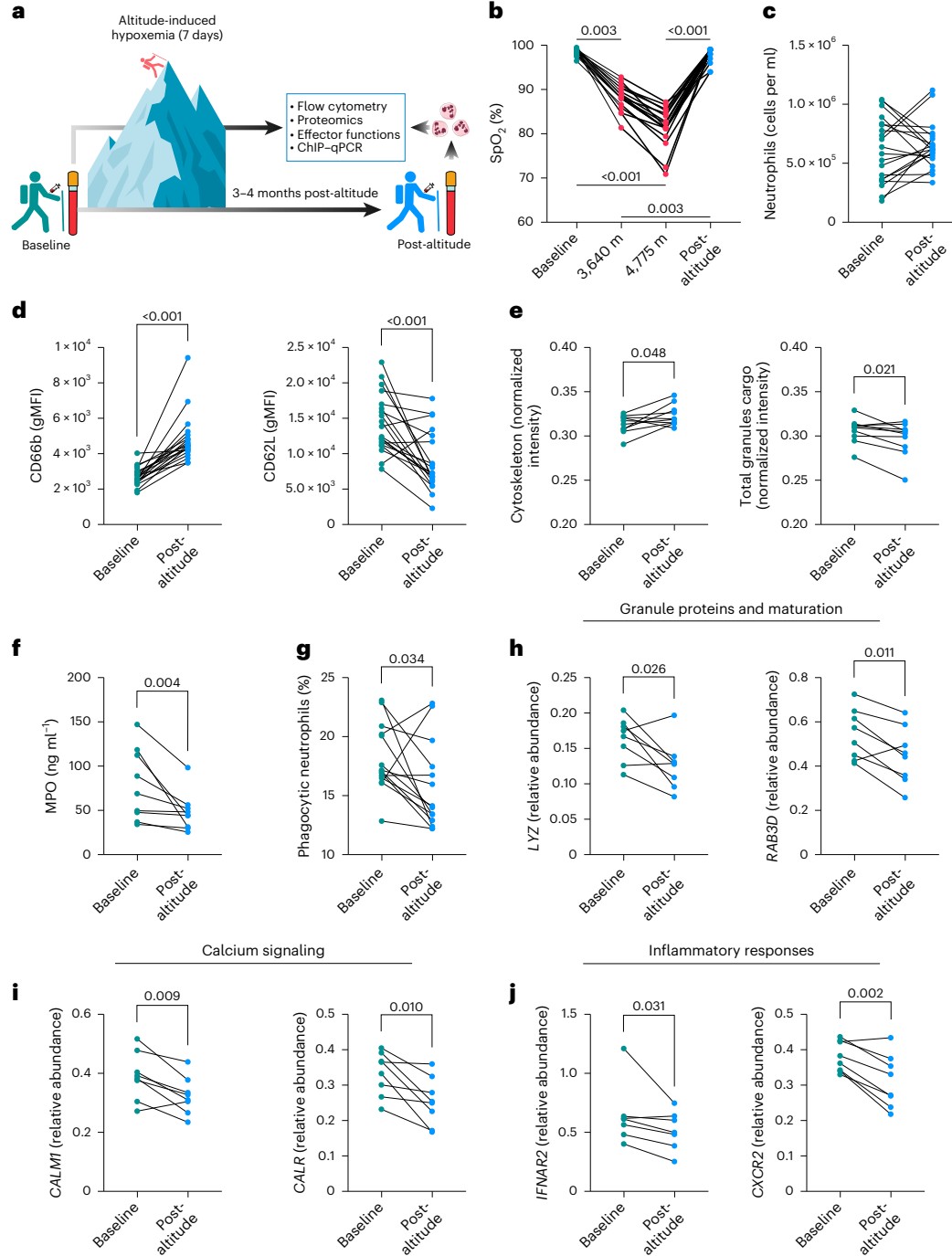

**Fig. 4 | Long-term neutrophil changes in response to ARDS are mirrored by altitude-induced hypoxemia. a–j,** Paired longitudinal analysis of blood neutrophils from volunteers at baseline and 3–4 months after a 7-day period of altitude-induced hypoxemia (post-altitude). **a,** Diagram of the analyses performed with the altitude-induced hypoxemia human cohort. Hiker adapted from Wikimedia commons (https://commons.wikimedia.org/wiki/File:Big_guy_637%27s_hiking_icon.svg); mountain adapted from Unsplash; neutrophils and blood tube adapted from Servier under license CC-BY 3.0 Unported. **b,** Oxygen saturation levels at baseline, at day 5 of an acclimatization period at 3,640 m of altitude, at day 7 after ascending to 4,775 m of altitude, and post-altitude ($n = 18$ for the different time points, exact $P$ value = $0.4 \times 10^{-7}$ for depicted $P < 0.001$). SpO$_2$, peripheral oxygen saturation. **c,** Circulating neutrophil counts obtained by flow cytometry ($n = 20$ both time points). **c,** Surface expression of the neutrophil activation markers CD66b and CD62L measured by flow cytometry ($n = 20$ for both time points, exact $P$ value = $0.2 \times 10^{-5}$ for CD66b and $P$ value = 0.0002 for CD62L). **e,** Proteomic analysis by LC−MS showing abundance

of cytoskeletal (GO:0005856) and total granule cargo proteins (GO:0035578, GO:0035580, and GO:1904724) ($n = 10$ for both time points). **f,** Ex vivo quantification of myeloperoxidase (MPO) from neutrophil culture supernatants performed by ELISA ($n = 9$ for both time points, two technical replicates per sample). **g,** Phagocytic capacity of opsonized *S. aureus* SH1000 measured by flow cytometry ($n = 14$ for both time points, two technical replicates per sample). **h–j,** Analysis of H3K4me3 levels in genes involved in granule proteins and maturation (**h**) ($n = 8$ for both time points, two technical replicates per sample), calcium signaling (**i**) ($n = 14$ for both time points, two technical replicates per sample) and inflammatory responses (**j**) ($n = 7$ for *IFNAR2* and $n = 8$ for *CXCR2* for both time points, two technical replicates per sample) by ChIP−qPCR expressed as % of input. Each set of values linked through a line represents an individual. Significant $P$ values depicted (for $P < 0.05$) and obtained by Shapiro−Wilk normality test followed by Friedman test (**b**), two-tailed paired *t*-test (**c**, CD62L (**d**), cytoskeleton (**e**), **h,i**, *CXCR2* (**j**)) or two-tailed Wilcoxon test (CD66b (**d**), total granules cargo (**e**), **f,g**, *IFNAR2* (**j**)).

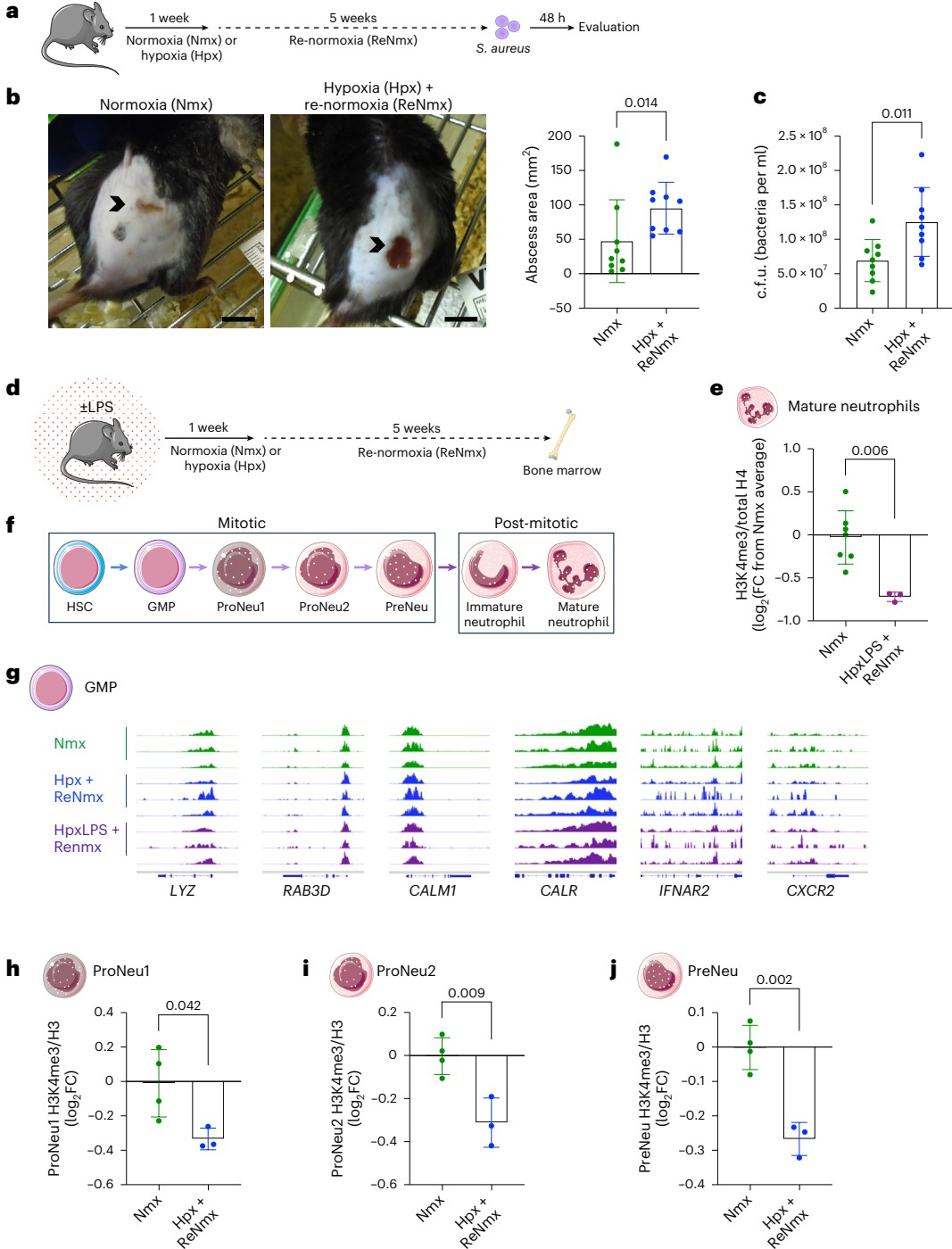

**Fig. 5 | Systemic hypoxia compromises bactericidal capacity and produces H3K4me3 loss in neutrophil progenitors. a**, Schema of the murine infection model used: mice were subjected to 10% O₂ for 1 week followed by a 5-week reoxygenation period before subcutaneous infection with *S. aureus* SH1000 (Hpx + ReNmx). Normoxia counterparts (Nmx) were used as experimental controls to evaluate infection outcomes 48 h post-challenge. **b,c**, Representative abscess (black arrowhead) images 24 h following infection, including abscess size quantification (**b**) and bacterial burden (c.f.u.) (**c**) 48 h post-infection (*n* = 9 for both experimental groups). Scale bar, 1 cm. **d**, Schema of the murine lung injury model used in **e,g**: mice were nebulized (HpxLPS + ReNmx) or not (Hpx + ReNmx) with LPS and subjected to 10% O₂ for 1 week followed by a 5-week reoxygenation period, when bone-marrow cells were collected. Normoxia counterparts (Nmx) were used as experimental controls. **e**, H3K4me3 levels normalized by total histone 4 content (H4) obtained by immunoblot in mature neutrophils isolated by FACS from the mouse model in **d** (*n* = 7 Nmx and *n* = 3 HpxLPS + ReNmx). log₂FC

from average Nmx is depicted. **f**, Simplified representation of the neutrophil hematopoietic lineage, including hematopoietic stem cells (HSCs), GMPs and the neutrophil-committed progenitors proNeu1, proNeu2 and preNeu. **g**, Single gene tracks reflecting H3K4me3 occupancy obtained by Cut&Run sequencing analysis in GMPs isolated from the mouse model in **d**. **h–j**, Levels of H3K4me3 as a ratio of total histone 3 (H3) in bone-marrow proNeu1 (**h**), proNeu2 (**i**) and preNeu (**j**) from mice subjected to 10% O₂ for 1 week followed by a 3-month reoxygenation period (Hpx + ReNmx) compared to normoxia control mice (Nmx) obtained by flow cytometry (*n* = 4 Nmx and *n* = 3 Hpx + ReNmx). log₂FC from average Nmx is depicted. Each data point represents a mouse. Data show mean ± s.d. Significant *P* values are depicted (for *P* < 0.05) and were obtained by a Shapiro–Wilk normality test followed by two-tailed *t*-test (**c**,**e**,**h**–**j**) or two-tailed Mann–Whitney *U*-test (**b**). Mouse in panel **a**,**d** adapted from Servier under license CC-BY 3.0 Unported; femur in panel **d** and cells in panels **e**–**j** adapted from Servier (https://smart.servier.com/smart_image) under license CC-BY 4.0.

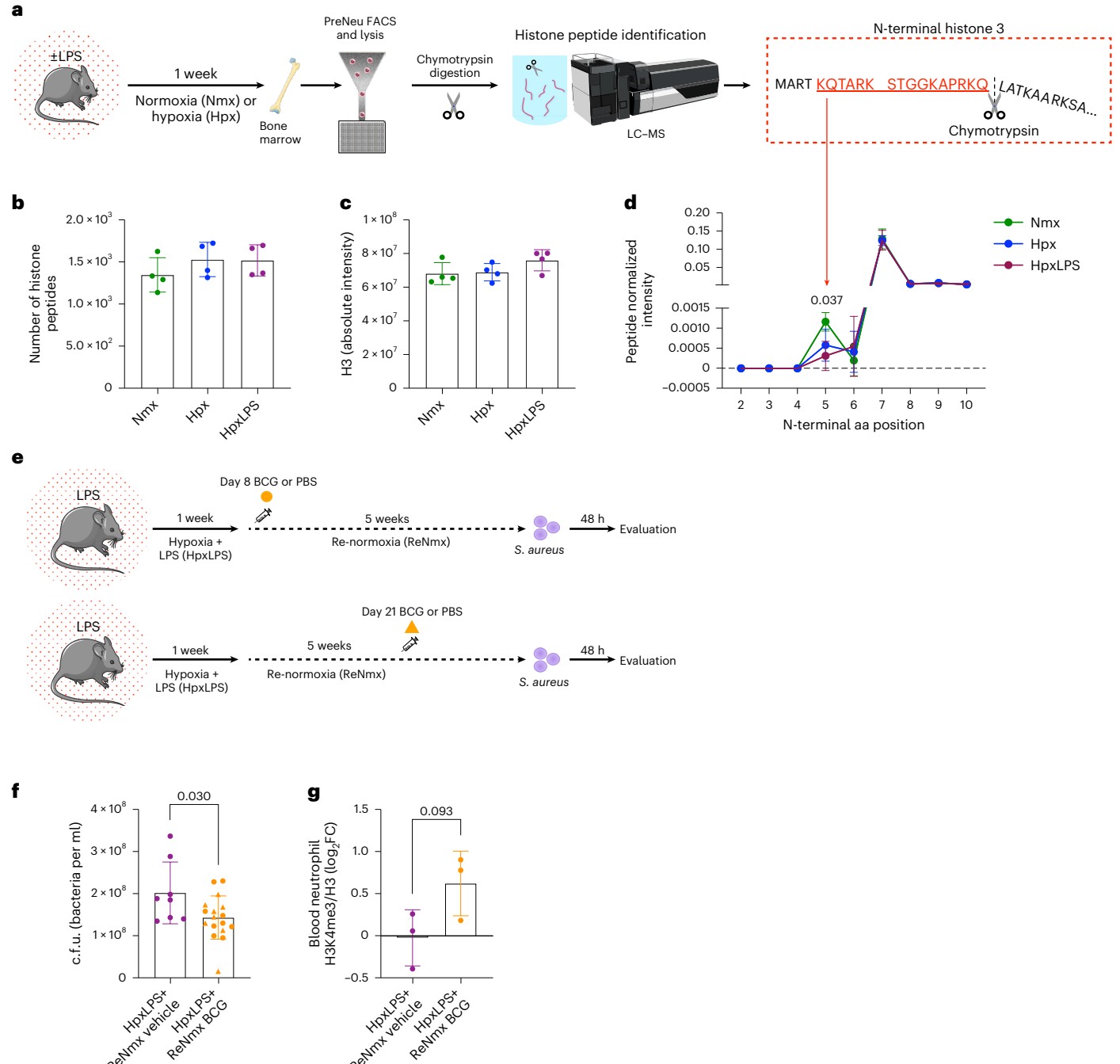

**Fig. 6 | Hypoxia-induced loss of H3K4me3 and impaired host defense are associated with H3 tail clipping and can be partially restored by BCG vaccination. a**, Schema of the acute systemic hypoxia murine model used for histone studies in **b**–**d**: mice were nebulized (HpxLPS) or not (Hpx) with LPS and subjected to 10% $O_2$ for 1 week, with normoxia counterparts (Nmx) used as experimental controls. Bone-marrow preNeu were isolated by FACS, lysed and submitted to chymotrypsin digestion. LC–MS was used to identify histone peptides. **b**, Total number of histone peptides identified. **c**, Total intensity for the peptides corresponding to H3. **d**, Abundance of N-terminal H3 peptides normalized by total H3 at different peptide start sites within the H3 protein sequence in **a** ($n = 4$ for all the experimental groups, two technical replicates per condition). Highlighted region in red depicts the fraction containing the trimethylation site lost in hypoxic conditions. **e**, Schema of the murine infection model used: mice were exposed to hypoxic lung injury with LPS nebulization followed by 1 week of hypoxia (10% $O_2$, HpxLPS). Following return to normoxia

(ReNmx), mice were vaccinated with BCG (HpxLPS + ReNmx BCG) or PBS control (HpxLPS + ReNmx vehicle) on either day 8 (circle) or day 21 (triangle) and, after 5 weeks recovery, challenged with a subcutaneous infection of *S. aureus*. **f**, Abscess c.f.u. counts 48 h post-infection ($n = 8$ HpxLPx + ReNmx vehicle and $n = 17$ HpxLPS + ReNmx BCG). **g**, Levels of H3K4me3 as a ratio of total histone 3 (H3) in circulating neutrophils obtained by flow cytometry ($n = 3$ for both experimental groups). $\log_2$FC from average HpxLPS + ReNmx vehicle is depicted. Each data point represents one mouse (**b**,**c**,**f**,**g**) or $n = 4$ mice per experimental group (**d**). Data show mean ± s.d. *P* values are depicted and were obtained by a Shapiro–Wilk normality test followed by a two-tailed *t*-test (**f**,**g**) or two-way analysis of variance (**b**–**d**). Mouse icons in panel **a**,**e** adapted from Servier under license CC-BY 3.0 Unported; femur in panel **a** adapted from Servier (https://smart.servier.com/smart_image) under license CC-BY 4.0; plate and mass spectrometer from panel **a** reproduced from Servier under licenses CC11.0 Universal and CC-BY 4.0 Unported, respectively.

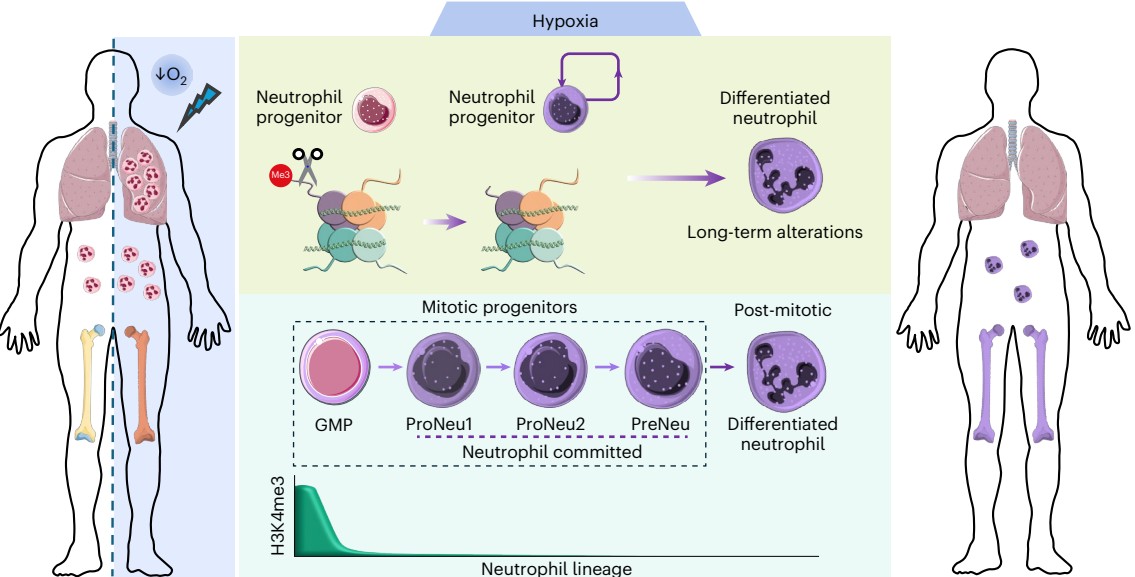

**Fig. 7 | Summary diagram.** In response to hypoxia, clipping of the N-terminal region of H3 in neutrophil-committed progenitors leads to loss of the H3K4 lysine trimethylation site. This widespread reduction in H3K4me3 is transmitted to differentiated neutrophils and perpetuated due to the mitotic capacity of neutrophil precursors. As a result, newly formed neutrophils present alterations in their phenotype and effector function in the long term with consequences for infection outcomes. Human silhouette adapted from Wikimedia commons (https://commons.wikimedia.org/wiki/File:Man_shadow_-_upper.png); femur, neutrophils and stem cells adapted from Servier (https://smart.servier.com/smart_image) under license CC-BY 4.0; neutrophils and DNA adapted from Servier under license CC-BY 3.0 Unported; nueclosome adapted from Servier under license CC0 1.0 Universal.

cells[26,27]. More recently, Cheung et al.[28] described how N-terminal H3 proteolytic cleavage by serine proteases epigenetically regulates monocyte-to-macrophage transition. N-terminal H3 clipping is an irreversible histone modification that eliminates the distal portion of the histone tail[29], where post-translational modifications like H3K4me3 lay. Therefore, we questioned whether acute hypoxia induces H3 clipping in neutrophil progenitors resulting in long-term loss of H3K4me3. To test this hypothesis, we exposed mice to 1 week of hypoxia or to hypoxic lung injury and isolated bone-marrow pre-Neu. Sorted cells were lysed, treated with chymotrypsin to promote the generation of N-terminal histone fragments, and resulting peptides analyzed by LC–MS (Fig. 6a). An equivalent number of total histone peptides was detected in hypoxic preNeu when compared to normoxia, indicating consistent histone abundance (Fig. 6b), with unchanged absolute H3 intensity (Fig. 6c). To quantify N-terminal histone clipping, we grouped all the peptides generated within the N-terminal region by starting amino acid and analyzed their abundance. Our results showed a significant hypoxia-mediated loss of the N-terminal H3 peptide KQTARKSTGGKAPRKQ, containing the lysine where trimethylation would occur (Fig. 6d). A similar loss was not observed in peptides at this site of the N-terminal region of histone H2B, suggesting site-specificity for this process (Extended Data Fig. 5f,g).

Agents promoting both enhanced or attenuated neutrophil responses to subsequent challenges have been reported in the literature[30]. BCG vaccination in humans results in a sustained increase in H3K4me3 levels in circulating neutrophils associated with functional reprogramming and improved antimicrobial activities[9]. Conversely, our data shows a long-lasting widespread loss of H3K4me3 in neutrophils linked to impaired responses and deficient pathogen control. We therefore questioned whether BCG vaccination could be used to restore the hypoxia-mediated impaired antibacterial capacity that we observe in vivo (Fig. 5b,c). To test this, we used a murine hypoxic lung injury model for 1 week followed by a 5-week recovery period in normoxia, in which BCG was administered on either day 8 or day 21. Upon completion of the recovery period, mice were infected with *S. aureus* and both abscess colony forming units (c.f.u.) and blood neutrophil H3K4me3 levels by flow cytometry were quantified as described above

(Fig. 6e). Our data revealed a reduction in c.f.u. recovered from the infected tissue, demonstrating improved pathogen control in vaccinated versus PBS-vehicle control mice (Fig. 6f). This was paired with the partial restoration of H3K4me3 abundance in circulating neutrophils, relating dynamic changes in H3K4me3 to infection outcomes in vivo (Fig. 6g).

In summary, these findings uncover a new role for hypoxia determining outcome of infections linked to a long-lasting suppression of H3K4me3 in circulating neutrophils that affects crucial genes for neutrophil responses. This H3K4me3 loss originates in self-renewing neutrophil progenitors, which experience N-terminal H3 clipping under low oxygen tensions leading to loss of trimethylation site (Fig. 7).

## Discussion

A proportionate neutrophil response to tissue injury is a hallmark of effective immunity. Conversely, dysfunctional neutrophilic inflammation underpins the tissue damage characteristic of many inflammatory diseases. One of the greatest therapeutic challenges for treating the tissue damage caused by neutrophilic inflammation remains how to mediate sustained reprogramming of short-lived rapidly turned-over inflammatory neutrophils enabling disease tolerance while preserving host resistance.

Our data speak to the ability of human neutrophils to alter core metabolic, epigenetic and functional programs relevant for disease states after an episode of ARDS that are retained over time and manifest in newly formed circulatory neutrophils. Paired analysis of individuals following exposure to altitude-induced hypoxemia reveals that the changes we describe in the disease setting are at least in part a consequence of the hypoxemia that defines ARDS. Our findings suggest that, in the context of acute inflammation, systemic hypoxia can act as a danger signal that activates a central program to redress the balance between host resistance and disease tolerance to limit tissue injury. While this enables the host to survive the acute injurious challenge, it sustains a lasting maladaptive response with consequences for subsequent host defense strategies. Of note, maladaptive responses following sepsis in the peripheral immune compartment have also been observed, with the emergence of neutrophil T cell

immunosuppressive phenotypes[31]. Together, these reprogrammed responses offer a putative link with the reported increased susceptibility to infections in convalescent patients with ARDS[23].

ARDS is a heterogenous condition, with multiple etiologies, defined in part by the presence of hypoxemia[32]. It is remarkable that, despite this heterogeneity and modest sample size, we detect conserved defects in neutrophil functions including activation, granule release, cell survival and bacterial phagocytosis. Conservation of hypoxic neutrophil reprogramming to a longitudinal cohort of high-altitude exposed volunteers and murine model systems also speaks to the biological relevance of these observations. Notably, neutrophils are at the front line of host defense and experience a turnover rate of 100 billion per day in homeostasis[33]. Therefore, we need to consider that relatively modest changes induced by neutrophil reprogramming at a cellular level have the potential to translate into biologically relevant nonlinear effects when exhibited across the whole population.

Altered chromatin accessibility has previously been associated with changes in neutrophil differentiation within the bone marrow[34] and the presence of inflammatory subsets within the blood[35]. Specifically, earlier studies have reported a granulopoiesis bias associated with changes in the epigenetic landscape in human hematopoietic stem and progenitor cells (HSPCs) following sepsis and COVID-19, which have been linked to persistent myeloid perturbations[31,36]. Using ChIP-seq, ChIP–qPCR, flow cytometry and LC–MS, our data reveal a widespread reduction in the levels of H3K4me3 in both newly formed differentiated neutrophils and their progenitors in response to systemic hypoxia that persists over time. Of note, our mouse model studies show that this does not manifest in the GMPs. Instead, this epigenetic change originates in the proNeu and preNeu committed progenitors within the neutrophil lineage, suggesting a restrictive and cell-specific pattern. This supports the concept of differential capacity within the hematopoietic lineage to respond to environmental triggers and transmit epigenetic cues that sustain the long-term reprogramming of newly formed immune populations.

We propose hypoxia-induced N-terminal clipping of histone 3 during neutrophil differentiation as a mechanism underlying the sustained widespread reduction in H3K4me3. Given recent evidence that neutrophil serine proteases can mediate histone clipping and regulate monocyte-to-macrophage differentiation[28], we speculate that the hypoxia-induced H3 clipping observed in our study may similarly depend on neutrophil granule protease activity. There is existing evidence that azurophilic granules can be disrupted by the generation of reactive oxygen species[37]. With perturbations in core metabolic programs and contraction of granule cargo proteins observed during acute hypoxic presentation with ARDS[6], future research will be directed to identifying potential specific proteases involved and delineate the mechanisms by which hypoxia facilitates N-terminal histone cleavage.

In neutrophils, the master regulator of cellular responses to hypoxia, HIF, critically regulates tissue survival, effector functions and core metabolic programs[12,20,38]. With the epigenetic landscape both regulating and being regulated by HIF activity[39], future work addressing the contribution of HIF to the reported H3K4me3 loss would likely shed light into the interplay between hypoxia and epigenetics in this context. Moreover, it will also be important to establish whether this central reprogramming is a direct consequence of local bone-marrow hypoxia or mediated via circulating cues returning to the bone marrow from the peripheral tissues.

It is now well established in human and murine systems that systemic inflammatory cues can initiate trained immune responses through changes in HSPCs within the bone-marrow compartment. Type I and II interferons (IFNs), interleukin (IL)-1 and granulocyte–macrophage colony-stimulating factor have all been implicated in training of granulopoiesis by β-glucan and BCG vaccination[8,30,40,41]. Our preliminary data following in vivo vaccination with BCG suggests

that training adjuvants can in part overcome the dysfunctional neutrophilic inflammation following exposure to systemic hypoxia. Future work will be required to delineate the mechanisms by which BCG vaccination mediates changes in the epigenetic landscape of bone-marrow populations with consequences for de novo neutrophil production and rescue of tissue neutrophil effector functions.

Neutrophils can regulate intracellular protein stores and suppress proinflammatory granule proteins through the activation of intrinsic circadian programs[42]. Conversely, tissue neutrophils have the capacity to fuel de novo protein synthesis and granule protein production through activation of hypoxia-sensitive protein scavenging pathways[12]. It is therefore likely that the distinct effector functions acquired by neutrophils at sites of infection and injury[43] will in part reflect tissue cues acting in concert with intrinsic reprogramming of newly formed neutrophil populations in the bone-marrow compartment. It may also explain in part why healthy individuals exposed to altitude-induced hypoxemia, while demonstrating impaired levels of neutrophil opsonic phagocytosis, do not report secondary infections in the following 6 months, in marked contrast to the ARDS cohort.

Our work has broader relevance not only to the long-term sequelae of severe COVID-19 (refs. 44–47), but to chronic inflammatory disease states in which neutrophils contribute to the pathogenesis of disease. For example, systemic changes in blood neutrophils and their precursors have been reported in early-stage chronic obstructive pulmonary disease[48], where changes in neutrophil state and abundance are associated with lung function decline. Understanding the core mechanisms that regulate this central reprogramming of neutrophil effector functions will finally enable therapeutic strategies that allow the long-term manipulation of neutrophilic inflammation in human disease states.

## Online content

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

Manuel A. Sanchez-Garcia [1], Pranvera Sadiku[1,24], Brian M. Ortmann[2,24], Niek Wit[2,24], Yutaka Negishi[3,4,5,24], Patricia Coelho[1], Ailiang Zhang[1], Chinmayi Pednekar[6], Andrew J. M. Howden [7], David M. Griffith [8], Rachel Seear[2], Jessica D. Kindrick[9], Janine Mengede[3,4,5], George Cooper[1], Tyler Morrison [1], Emily R. Watts[1], Benjamin T. Shimeld[1], Leila Reyes[1], Ananda S. Mirchandani [1], Simone Arienti [1], Xiang Xu [1], Alexander Thomson[1], Alejandro J. Brenes [7,10], Helena A. Turton[11], Rebecca Dowey [11], Rebecca C. Hull[11], Hazel Davidson-Smith[12], Amy McLaren[13], Andrew Deans[13], Gourab Choudhury[13], Katherine Doverman[14], David Hope[14], Oliver Vick [1,15], Alastair Woodhead[1,15], Isla Petrie [1,15], Suzanne Green [1,15], Nina M. Rzechorzek [15,16,17], Lance Turtle[18,19], Peter J. M. Openshaw [20], Malcolm G. Semple [18,21], the PHOSP-COVID Study Collaborative Group*, Duncan Sproul[22], J. Kenneth Baillie [23], Alfred A. R. Thompson [11], David R. Mole [9], Alex von Kriegsheim [6], Moira K. B. Whyte [1], Musa M. Mhlanga [3,4,5], James A. Nathan [2] & Sarah R. Walmsley [1] ✉

[1]Centre for Inflammation Research, Institute for Regeneration and Repair, University of Edinburgh, Edinburgh, UK. [2]Cambridge Institute of Therapeutic Immunology and Infectious Disease (CITIID), Jeffrey Cheah Biomedical Centre, Department of Medicine, University of Cambridge, Cambridge, UK. [3]Department of Internal Medicine and Radboud Center for Infectious Diseases, Radboud University Medical Center, Nijmegen, the Netherlands. [4]Department of Cell Biology, Faculty of Science, Radboud Institute for Molecular Life Sciences, Radboud University Nijmegen, Nijmegen, the Netherlands. [5]Department of Human Genetics, Radboud University Medical Center, Nijmegen, the Netherlands. [6]CRUK Scotland Centre, IGC, University of Edinburgh, Edinburgh, UK. [7]Division of Cell Signalling and Immunology, University of Dundee, Dundee, UK. [8]The Usher Institute, University of Edinburgh, Royal Infirmary of Edinburgh, Edinburgh, UK. [9]NDM Research Building, Nuffield Department of Medicine, University of Oxford, Oxford, UK. [10]Centre for Gene Regulation and Expression, University of Dundee, Dundee, UK. [11]Division of Clinical Medicine, University of Sheffield, Sheffield, UK. [12]MRC Human Genetics Unit, MRC IGC, University of Edinburgh, Edinburgh, UK. [13]NHS Lothian, Respiratory Medicine, Edinburgh, UK. [14]Edinburgh Critical Care Research Group, NHS Lothian, Royal Infirmary of Edinburgh, Edinburgh, UK. [15]Altitude Physiology Expeditions (APEX), Sheffield, UK. [16]MRC Laboratory of Molecular Biology, Cambridge, UK. [17]Department of Engineering, University of Cambridge, Cambridge, UK. [18]Health Protection Research Unit in Emerging and Zoonotic Infections, Institute of Infection, Veterinary and Ecological Sciences, University of Liverpool, Liverpool, UK. [19]Tropical and Infectious Disease Unit, Liverpool University Hospitals NHS Foundation Trust (member of Liverpool Health Partners), Liverpool, UK. [20]Respiratory Infection Section, National Heart and Lung Institute, Imperial College London, London, UK. [21]Respiratory Department, Liverpool Institute of Child Health and Wellbeing, Alder Hey Children's Hospital NHS Foundation Trust, Liverpool, UK. [22]MRC Human Genetics Unit and CRUK Edinburgh Centre, MRC IGC, University of Edinburgh, Edinburgh, UK. [23]The Roslin Institute, University of Edinburgh, Edinburgh, UK. [24]These authors contributed equally: Pranvera Sadiku, Brian M. Ortmann, Niek Wit, Yutaka Negishi. *A full list of members and their affiliations appears in the Supplementary Information. ✉e-mail: sarah.walmsley@ed.ac.uk

## Methods

### Blood donors

Peripheral venous blood was taken from male and female healthy volunteers as a control group with written informed consent obtained before sample collection, as approved by the University of Edinburgh Centre for Inflammation Research Blood Resource Management Committee (AMREC 15-HV-013, 21-EMREC-041). The total number of control participants was 33, with a male:female ratio of 10:23 and an average age range of 31–40 years. Blood donations by male and female participants undergoing altitude-induced hypoxemia were approved by the Edinburgh Medical School Research Ethics Committee (EMREC, 21-EMREC-043) and written consent was obtained from the participants. The total number of participants for the altitude-induced hypoxemia study was 20, with a male:female ratio of 8:12 and an average age of 22 years. Blood was taken at baseline conditions before the volunteers were exposed to altitude-induced hypoxemia for 7 days at 4,775 m altitude (Huayna Potosí, Bolivia), with previous acclimatization for a 5-day period at 3,640 m altitude (La Paz, Bolivia). The same individuals were bled between 3–4 months after returning to baseline altitude conditions. Oxygen saturation levels were taken by using a pulse oximeter.

For ARDS samples, patient recruitment took place from April 2020 to April 2025 mainly at The Royal Infirmary of Edinburgh, UK, through the ARDS Neut study (20/SS/0002) and in association with the PHOSP study (20/YH/0225), the University of Cambridge (17/EE/0025) and the University of Sheffield (18/YH/0441). All data presented in this manuscript were obtained from prospectively recruited patient cohorts. The principal objective of these studies was to assess neutrophil properties in acute disease and during convalescence. Patients with ARDS included in this study were adult male and female intensive care unit (ICU) patients diagnosed with moderate–severe ARDS according to the Berlin criteria[14] at any time during ICU admission. The acute physiology and chronic health evaluation (APACHE II) score = acute physiology score + age points + chronic health points (minimum score = 0; maximum score = 71) was calculated, where increasing score is associated with increasing risk of hospital death[49]. Data on the assessment by clinicians upon patients' admission to the intensive care unit, or earliest time possible, and follow-up infection data, were collated by researchers at a later time. To obtain consent, a member of the clinical team carried out initial approaches to patients or relatives, welfare attorney or legal representative in cases where patients lacked capacity due to compromised health status. Information sheets were provided in all cases, and guided explanation was given through the information sheets. Separate contact after consideration was made by the research team with the patient or representatives and formal written consent obtained. Patients were recruited and blood sampling undertaken within 7 days of patients meeting the Berlin criteria definition of moderate–severe ARDS for leukocyte counts and 3–6 months post-hospital discharge for long-term studies. Due to limited sample volumes, cell availability and high demands on cell number for 'omics assays', it was not possible to conduct every assay on each patient sampled. Assays were performed sequentially as samples became available/cell number permitting with no previous selection. There was no participant compensation.

### Mouse studies

Animal experiments were conducted in accordance with the UK Home Office Animals (Scientific Procedures) Act of 1986. All animal studies were approved by The University of Edinburgh Animal Welfare and Ethical Review Board, adhered to the principles of '3Rs' (replacement, reduction and refinement) and complied with ARRIVE guidelines for animal research: inclusion of control groups, individual datapoints reflecting number of animals used in each experiment, animals terminated early if humane end points were reached, animals randomly assigned to the experimental groups, experimenter blind to treatments

and outcome measures and statistical analysis clearly stated in figure legends. Male 3–6-month-old C57BL/6 mice were used for these studies. Mice were housed under 12-h light–dark cycles and controlled temperature (20–23 °C) and humidity (45–65%), in accordance with UK Home Office guidance. All mice used for experiments were healthy, with quarterly and annual testing carried out in accordance with FELASA 2014 guidelines, using a mixture of environmental, random colony samples and sentinel testing by serology and PCR. Mice had ad libitum access to food (Special Diets Services, 801151) and water. Hypoxia exposure was achieved by incubation for one week at 10% $O_2$ (hypoxia) in a hypoxia chamber (Coy Labs), with excess $CO_2$ scavenged using Sofnolime soda lime chips (Molecular Products). Uniform and reproducible mouse hypoxic acute lung injury was performed by administering nebulized LPS from $P. aeruginosa$ 10 (Sigma Aldrich) before hypoxic exposure, as previously described[12]. Control normoxia or reoxygenation conditions were achieved by room $O_2$ exposure (21% $O_2$).

### Human blood neutrophil isolation

Neutrophil isolation from peripheral blood was performed as described in previous studies[6]. In brief, blood was collected in citrate tubes and spun down to obtain a cell-enriched layer. Dextran sedimentation was used to discard erythrocytes followed by discontinuous Percoll gradients to isolate highly pure neutrophils.

### Human flow cytometry neutrophil counts

A flow cytometry-based approach was utilized to quantify the concentration of unstained neutrophils in the peripheral blood of patients following ARDS and volunteers from the altitude-induced hypoxemia study. Immediately after blood withdrawal, 100 µl of blood was diluted with 1 ml of FACS-Lyse buffer (BD) and incubated for 10 min on ice. Samples were pelleted and Countbright Beads (Thermo Fisher Scientific) were added before analyzing the samples in a flow cytometer BD LSRFortessa (Becton Dickinson). The gating strategy was SSC-H/SSC-A > FSC-H/FSC-A > SSC-A/FSC-A (beads and granulocytes). Any eosinophils were excluded from the granulocytes gate based on their autofluorescent properties, as previously reported[50].

### Human flow cytometry phenotyping

Neutrophils were isolated as described above and stained with Zombie Aqua (BioLegend) to exclude dead cells. Human Fc block (BioLegend) followed by staining with an antibody mix containing CD66b (BioLegend, cat. 305114), CD16 (eBioscience, 11-0168-42), CD62L (BioLegend, 304814) and CD49d (BioLegend, 304322) was used. Stained neutrophils were fixed with 4% PFA and data acquired in a BD LSRFortessa flow cytometer (Becton Dickinson). Fluorescence Minus One were used as negative controls and were subtracted from stained samples to obtain geometric mean fluorescence intensity (gMFI) values. The gating strategy was SSC-H/SSC-A > FSC-H/FSC-A > SSC-A/Zombie Aqua > SSC-A/FSC-A (cells) > CD66b$^+$ CD49d$^-$ (neutrophils) > CD66b, CD16 and CD62L. Representative gating strategy plots are shown in Extended Data Fig. 7.

### Metabolomics analysis

This analysis was performed by LC–MS at the Institute of Genetics and Cancer, Edinburgh. Freshly isolated neutrophil samples were lysed in 2:2:1 methanol:acetonitrile:$H_2O$. A blank control tube containing only lysis buffer was used to check for a background signal. Cellular debris was discarded via centrifugation at 18,000$g$ for 15 min at 4 °C followed by collection of the supernatant containing metabolites. The samples were transferred to a 96-well PCR plate for sample loading. A Millipore Sequant ZIC-pHILIC analytical column (5 µm, 2.1 × 150 mm) with a 2.1 × 20 mm guard column (both 5-mm particle size) was used to separate the metabolites in solution. This column was equipped with a binary solvent system integrated by Solvent A, which consisted of 20 mM ammonium carbonate, 0.03% ammonium

hydroxide, and Solvent B, which was acetonitrile. The chromatographic gradient was run at a flow rate of 0.200 ml min$^{-1}$ as follows: 0–2 min, 80% B; 2–18 min, linear gradient from 80% B to 20% B; 18–18.5 min, linear gradient from 20% B to 80% B; and 18.5–27.5 min, hold at 80% B. The temperature of the column oven was kept constant at 40 °C and the temperature of the tray was held at 10 °C. An injection volume of 20 μl was used to randomize the samples, and a Thermo Scientific Q-Exactive Hybrid Quadrupole-Orbitrap mass spectrometer was connected to the HPLC system. The mass spectrometer was operated in full-scan, polarity-switching mode, with the spray voltage set to +4.5 kV/−3.5 kV. The heated capillary was kept at 320 °C, and the auxiliary gas heater held at 280 °C. For analysis of acetyl-CoA, samples were acquired in single ion monitoring mode with the machine in positive mode and scanning windows of $m/z = 8$ between 765.1147–874.1385.

Metabolite abundance was determined in a targeted manner, using Skyline software (v.21.1, MacCoss Lab Software). In-house generated standards were used to identify metabolites according to their expected retention time. The area under the curve was manually determined and expressed as arbitrary units of abundance.

## Proteomics analysis

Samples were processed for LC–MS analysis at the University of Dundee as previously described[6]. In short, neutrophil proteins were extracted in 5% SDS after a red blood cell lysis step to prevent erythrocyte contamination followed by sonication and alkylation. Protein lysates were processed for mass spectrometry using s-trap spin columns following the manufacturer's instructions (Protifi)[51]. The peptides were sequentially eluted and 2 μg of peptide analyzed on a Q-Exactive-HF-X[6,52] (Thermo Scientific) mass spectrometer in the case of the ARDS survivor's cohort and 1.5 μg of peptide analyzed on an Exploris 480[53] (Thermo Scientific) mass spectrometer in the case of the high-altitude study.

The data-independent acquisition data were analyzed with Spectronaut 14, for the post-ARDS cohort, or Spectronaut 16, for the high-altitude study, using the directDIA option[54]. The false discovery rate threshold was set to 1% Q-value at both the Precursor and Protein level. The directDIA data were searched against the human SwissProt database (July 2020 for the post-ARDS cohort and January 2021 for the high-altitude study) and included isoforms. Estimates of protein copy numbers per cell for proteomic data belonging to acute phase patients with ARDS were calculated using the proteomic ruler method[55]. Alternatively, proteomic data resulting from ARDS survivors and high-altitude study samples was expressed as normalized intensities to avoid copy-number estimation errors given that the total protein content for the groups analyzed was significantly different (Extended Data Fig. 6). Normalized intensity values were obtained by dividing the intensity values of each protein detected in a sample by the total sum intensity detected for that particular sample. Principal-component analysis plots were generated with Perseus software[56]. GO databases were used to analyze the protein abundance of components of mitochondria (GO:0005739), nuclear envelope (GO:0005635), ribosome (Kyoto Encyclopedia of Genes and Genomes annotation 03010), cytoskeleton (GO:0005856), azurophilic granule cargo proteins (GO:0035578) and total granule cargo proteins (GO:0035578, GO:0035580 and GO:1904724). The eukaryotic initiation factor 4F was represented considering normalized intensities of its components PABPC1, EIF4G1, EIF4E and EIF4A1.

## α-1-Antitrypsin and MPO detection

Neutrophils isolated as per above were cultured at 5 million per ml in RPMI medium (Gibco) supplemented with 5.5 mM glucose, 5% dialyzed FBS and 1% penicillin–streptomycin for 4 h. Culture supernatants were collected and free α-1-antitrypsin (Bethyl Laboratories) or myeloperoxidase (MPO) (Abcam) were quantified by ELISA as per the manufacturer's instructions.

## Ex vivo phagocytosis

The *S. aureus* SH1000 strain was stained with 9 μM CFS-E (Thermo Fisher Scientific) and heat-killed by incubation at 80 °C for 1 h. Dialyzed FBS (Thermo Fisher Scientific) was used to opsonize the bacteria for 30 min at 37 °C, before incubation with neutrophils. Isolated neutrophils as described above were distributed in a 96-well plate and incubated in a 5:1 bacteria to neutrophil ratio for 10 min at 37 °C. Cells were transferred into a fresh plate containing Trypan blue (Sigma) to quench any extracellular fluorescent signal before data acquisition in a Attune Nxt autosampler (Thermo Fisher Scientific). Samples were run in duplicates with neutrophils in the absence of bacteria used to establish CFS-E$^+$ events. The gating strategy was SSC-H/SSC-A > FSC-H/FSC-A > SSC-A/FSC-A (neutrophils) > CFS-E$^+$.

## Neutrophil survival

Isolated neutrophils were cultured at 5 million per ml in RPMI medium (Gibco) supplemented with 5.5 mM glucose, 5% of dialyzed FBS and 1% penicillin–streptomycin for 20 h in the absence or presence of 100 ng ml$^{-1}$ of LPS *Escherichia coli* serotype R515 (Enzo) at 37 °C and 5% $CO_2$. Cells were collected onto slides for staining by using a cytospin station (Thermo Shandon) and fixed with methanol. Subsequent staining with hematoxylin and eosin allowed for visual evaluation of neutrophil apoptosis under a bright-field microscope. The fold change of survival from LPS unstimulated conditions was used as measurement.

## H3K4me3 ChIP

We performed H3K4me3 ChIP-seq as described elsewhere[57] and in brief below, with the following modifications. Neutrophil pellets were incubated in 1% paraformaldehyde for 10 min followed by 0.125 M glycine for 10 min. Cells were then lysed with 150 μl of ChIP lysis buffer (50 mM Tris-HCl, pH 8.1, 1% SDS, 10 mM EDTA and Complete Mini EDTA-free protease inhibitor) for 10 min on ice. A 1:1 dilution was performed by adding 150 μl of ChIP dilution buffer (20 mM Tris-HCl, pH 8.1, 1% (v/v) Triton X-100, 2 mM EDTA and 150 mM NaCl) before chromatin shearing by sonication with Bioruptor (Diagenode), with 25 cycles of 30 s on followed by 30 s off. The samples were spun at 16,000*g* 10 min at 4 °C, and a 20 μl aliquot was created as input control and stored at −80 °C. The remainder of the samples was diluted by adding 1.2 ml of ChIP dilution buffer followed by immunoprecipitation with anti-H3K4me3 (Cell Signaling, 9751S) antibody. Immunocomplexes were magnetically labeled and eluted with 120 μl elution buffer (1% (w/v) SDS and 0.1 M Na-bicarbonate). Both inputs and immunoprecipitated samples underwent reversal of crosslinking and protein digestion. DNA was purified using the DNA minielute kit (QIAGEN) according to the manufacturer's guidelines.

## ChIP-seq analysis

Successful sample fragmentation was confirmed by analysis with TapeStation (Agilent) (Extended Data Fig. 3a,b). Raw sequencing reads were first quality trimmed using Trim Galore (v.0.6.7)[58] followed by read mapping with HISAT2 (v.2.2.1)[59] against the human genome (hg19) in paired-end mode. Reads mapping to blacklisted regions[60] were removed with the bedtools (v.2.30)[61] intersect tool and sorted by coordinate and indexed using SAMtools (v.1.15)[62] sort and index, respectively. From the resulting BAM files, duplicated reads were removed using the MarkDuplicates tool from Picard (Broad Institute) with the setting REMOVE_DUPLICATES = TRUE. Any potential differences in sequencing depth were corrected by down-sampling all samples to the same number of fragments. BAM files were merged per group with SAMtools merge to create subsequent metagene H3K4me3 profiles with ngs.plot (v.2.61)[63]. This software performs two normalization steps: the coverage vectors (gene regions) are normalized to be equal length, and the vectors are normalized against the corresponding library size (the total read count for the reads that pass quality filters) to generate the reads per million mapped reads. Additionally, data from

each individual was normalized by its corresponding input sample. We represented 10,000 bins around transcription start sites, which confirmed a higher concentration of peaks at these genomic areas across study groups (Extended Data Fig. 3c), as expected according to previous studies[64]. The random distribution of 10,000 bins along the DNA revealed equivalent low abundance profiles (Extended Data Fig. 3c). Also, preserved levels of H3K4me3 were detected in certain genomic regions including the genes *NBPF26*, *ACTN2*, *HRNR* and *PLPRP5* (Extended Data Fig. 3d). H3K4me3 peaks were identified using the MACS3 (v.3.0.0.a6)[65] callpeak function. Differential H3K4me3 peak analysis between conditions and enriched pathway analysis of differential peaks was performed with a custom R (v.4.2.0) script[66] using the Bioconductor packages DiffBind[67,68], ChIPseeker[69] and ReactomePA[70]. The BED files generated by MACS3 were first annotated using the HOMER (v.4.11)[71] annotatePeaks.pl script to obtain the nearest peak IDs and genes (Ensembl gene IDs). These were subsequently merged with the corresponding MACS3 _peaks.xls file to obtain the pileup values for each peak with a custom Python (v.3.10.8) script[66] using the pandas data analysis package[72]. Bigwig files were visualized with IGV or genome browser.

### ChIP–qPCR

ChIP using anti-H3K4me3 antibody (Cell Signaling, 9751S) was performed as described in 'H3K4me3 ChIP' above to isolate DNA from input controls and anti-H3K4me3 pulled down samples. qPCR analysis was performed by using PowerUp SYBR Green (Thermo Fisher), following the manufacturer's guidelines. Template DNA was amplified with ABI QuantStudio 5 and analyzed as percent of input[73]. Primer sequences were generated with Primer3 software: *LYZ* forward (5′-AATGGAT GGCTACAGGGGAATC-3′), *LYZ* reverse (5′-AGCCCCTTCTTCTTCTTC CTTC-3′), *RAB3D* forward (5′-TTCCAGGCAATCTGTCCCAC-3′), *RAB3D* reverse (5′-GCACCTTGGACTCGGATGAA-3′), *CALM1* forward (5′-AGC TGCGCTTAAAGGAGGTT-3′), *CALM1* reverse (5′-CTACGACCAAGTC CAGCTCC-3′), *CALR* forward (5′-AGTTTCTGGACGGAGGTAACG-3′), *CALR* reverse (5′-ACAACGCAGATCCAGGATCG-3′), *IFNAR2* forward (5′-AGCTGACTGGAGGGAAAACG-3′), *IFNAR2* reverse (5′-CAGGAG GAGGAGGAGGAGTC-3′), *CXCR2* forward (5′-AGAAGGAGGCTGACTG GGAA-3′) and *CXCR2* reverse (5′-GTACCTCCCTGTGTCCCAGA-3′).

### Skin infection

Mice were exposed to hypoxia or normoxia for 1 week, as described above. After a period of 5 weeks of reoxygenation, mice were depilated and subcutaneously injected with $5 \times 10^7$ *S. aureus* SH1000. Abscess development was monitored over a 48-h period and pictures were taken to quantify scab growth in Fiji[74]. Mice were killed 48 h post-infection, a sample of their blood was taken and their abscesses were collected and digested by using a tissue blender (Next Advance) and homogenizing beads (Precellys). The number of c.f.u. was quantified by performing serial dilutions of the lysate, plating onto blood agar plates and counting colonies after an overnight period at 37 °C and 5% $CO_2$. Blood was stained after a red blood cell lysis step (BioLegend) with Zombie Aqua (BioLegend) mouse Fc block (BioLegend, 101320) and an antibody mix containing anti-CD45 (BioLegend, 103128) and anti-Ly6G (BioLegend, 127628) antibodies. Stained neutrophils were fixed with 1.5% PFA for 10 min on ice and washed before data acquisition in a BD LSRFortessa flow cytometer (Becton Dickinson). Fluorescence Minus One were used as negative controls. The gating strategy was SSC-H/SSC-A > FSC-H/ FSC-A > SSC-A/Zombie Aqua⁻ > SSC-A/FSC-A > SSC-A/CD45⁺ > SSC-A/ Ly6G⁺ (neutrophils; % of CD45⁺).

### H3K4me3 immunoblot

Mice undergoing hypoxic lung injury were subjected to a recovery period of 5 weeks in normoxia, with normoxia counterparts used as control (see above). After the 5-week period of reoxygenation, bone-marrow tissue was collected and filtered through a 40-µm cell

trainer to obtain a cell suspension. A discontinuous Percoll gradient was created by overlaying HBSS–Percoll-based solutions with 81%, 62% and 55% Percoll and used to obtain a neutrophil-enriched layer after a 2,000*g* centrifugation at 21 °C for 30 min. A BD FACSAria Fusion flow cytometer fitted with a 70-µm nozzle was used to sort mature neutrophils based on their SSC/FSC profile.

An acid extraction of histones was performed as previously described[75]. In brief, a hypotonic lysis solution (10 mM Tris-HCl, pH 8.0, 1 mM KCl, 1.5 mM $MgCl_2$ and 1 mM dithiothreitol) was used to extract cell nuclei followed by incubation in 0.2 M of $H_2SO_4$ (Sigma) for an extra 30 min. Incubation with 33% w/v TCA (Sigma Aldrich) allowed the precipitation of histones followed by acetone washes and resuspension in ultrapure water. Protein quantification was performed with BCA protein assay kit (Thermo Scientific) as per manufacturer's guidelines. SDS–PAGE by using 4–12% Bis-Tris gels and 2 µg of histone sample was performed, followed by transfer into nitrocellulose membranes. A Li-cor Intercept system was used for primary antibody incubation and secondary antibody detection. Specifically, H3K4me3 primary antibody (Upstate, 07-473) and H4 primary antibody (Abcam, Ab31830, loading control) were followed by Li-cor anti-rabbit (925-32211, 800 CW) and Li-cor anti-mouse (cat. 925-68070, 680 RD) for band visualization in Li-cor Odyssey CLX imaging system and subsequent quantification.

### Cut&Run

Mice underwent hypoxia exposure or hypoxic lung injury as described above, with normoxia counterparts used as control. After a period of 5 weeks of reoxygenation, bone-marrow tissue was collected and filtered through a 40-µm cell strainer to obtain a cell suspension. A red blood cell lysis step (BioLegend) was used before staining with anti-CD16/32 antibody (BioLegend, 101326). A mix containing extra antibodies anti-lineage (BioLegend, 78022), anti-Sca1 (BioLegend, 108129), anti-cKit (BioLegend, 105835) and anti-CD34 (BioLegend, 152208) was subsequently added. The samples were sorted with a BD FACSAria Fusion cell sorter, after adding 4,6-diamidino-2-phenylindole (DAPI), to obtain a highly pure GMP population. The gating strategy was SSC-A/ FSC-A > SSC-H/SSC-A > FSC-H/FSC-A > Zombie Aqua⁻/Lin⁻ > cKit⁺/ Sca1⁻ > CD34⁺/CD16/32ʰⁱᵍʰ (GMP) or CD34⁺/CD16/32ˡᵒʷ (CMP). Representative gating strategy plots are shown in Extended Data Fig. 7.

Cells were fixed with 0.1% PFA for 2 min at 21 °C and crosslinking stopped by adding glycine solution (Cell Signaling) and incubating for 5 min. Cells were pelleted at 3,000*g* for 3 min to discard the supernatant before freezing to store. Cut&Run libraries were generated as per manufacturer's guidelines (Cell Signaling Technology). In brief, cells were thawed and bound with concanavalin A beads. After cell permeabilization by digitonin, anti-H3K4me3 antibody (Cell Signaling, 9751S) was added to the samples and incubated for 2 h at 4 °C. After removing unbound antibodies by washing, the samples were mixed with pAG-MNase and incubated for 1 h at 4 °C. After washing, pAG-MNase was activated by adding calcium chloride. DNA was digested for 30 min at 4 °C and the digestion reaction was stopped with EDTA. DNA fragments were collected by column-based DNA purification. DNA sequencing libraries were prepared by HAKA HyperPrep kit. The Cut&Run libraries were sequenced by NextSeq 2000. The quality of FASTQ files was assessed by the FastQC program. After trimming the adaptor sequences, the reads were aligned to the mouse genome mm10 by BWA and peaks were called by the MACS2 program. Bigwig files were generated by MACS2 and visualized with IGV or genome browser.

### H3K4me3 flow cytometry

Mice were exposed to hypoxia or normoxia as described above. After a period of 3 months of reoxygenation (21% $O_2$), bone-marrow tissue was collected and filtered through a 40-µm cell strainer to obtain a cell suspension. Cells were incubated with anti-CD16/32 antibody (BioLegend, 101333) followed by addition of a lineage cocktail containing

biotinylated anti-CD90.2 (eBioscience, 13-0902-82), anti-B220 (eBioscience, 13-0452-82), anti-NK1.1 (eBioscience, 13-5941-82), anti-Sca1 (eBioscience, 13-5981-82), anti-Flt3 (eBioscience, 13-1351-82) and anti-CD115 (eBioscience, 13-1152-82), anti-Ter119 (eBioscience, 13-5921-82) antibodies. Anti-biotin microbeads (Miltenyi Biotec) where used to remove lineage+ cells from the cell suspension by using LS magnetic columns (Miltenyi Biotec) as detailed by the manufacturer. Negatively selected eluted cells were stained with an antibody cocktail containing anti-CD34 (BioLegend, 152218), anti-Ly6G (BioLegend, 127628), anti-CD11b (BioLegend, 101257), anti-cKit (BioLegend, 105835), anti-CD106 (BioLegend, cat. no. 105716), anti-SiglecF (BioLegend, 155534), anti-Ly6C (BioLegend, 128041) and anti-Gr1 (BioLegend, 108417) plus streptavidin-BV650 (BioLegend). LIVE/DEAD Fixable Near IR (Thermo Fisher) was followed by fixation with 1.5% PFA for 10 min on ice. Cell permeabilization was achieved by resuspending the cells in 0.3% Triton X-100 (Sigma) for 10 min at 21 °C. Subsequent intracellular histone staining involved incubation with anti-H3 (Abcam, ab313347) and anti-H3K4me3 (Abcam, ab237342) antibodies for 30 min at 21 °C. Data acquisition was performed in a BD LSRFortessa flow cytometer (Becton Dickinson). Fluorescence Minus One were used as negative controls and were subtracted from stained samples to obtain gMFI values. The gating strategy was SSC-A/FSC-A > SSC-H/SSC-A > FSC-H/FSC-A > SSC-A/Live/Dead⁻ > Ly6G⁻/Lin⁻ > CD16/32high/Ly6C+ > cKithigh/CD34+ > CD11blow/CD106⁻ (ProNeu1) or CD11bhigh/CD106+ (ProNeu2). SSC-A/FSC-A > SSC-H/SSC-A > FSC-H/FSC-A > SSC-A/Live/Dead⁻ > Ly6G⁻/Lin⁻ > SiglecF⁻/CD11b+ > cKit+/Gr1+ > SSC-A /CD34⁻ (PreNeu). Representative gating strategy plots are shown in Extended Data Fig. 7.

## PreNeu mass spectrometry

Hypoxia-exposed or hypoxic lung injury mice were generated as described above, with normoxia counterparts used as controls. Bone-marrow preNeu were collected and stained as described above, except intracellular staining was not used and DAPI identified alive cells before FACS (Aria II, Becton Dickinson) into a 384-well plate (Thermo Scientific). Cells were lysed in lysis buffer containing 100 mM TEAB, 1 mM CaCl₂, 0.2% (v/v) DDM, 0.01% protease enhancer (Promega) and 6 ng μl⁻¹ chymotrypsin (Promega) in ultrapure water. The plate was sealed with Thermowell sealing tape (Corning, 6569) followed by a 10-min sonication and incubation at 50 °C for 2 min plus 4 h at 37 °C in a thermal cycler (PE Applied Biosystems, GeneAmp PCR System 9700). Next, the temperature was reduced to 20 °C and 3.5 μl of 1% trifluoroacetic acid was added to the samples. Samples were loaded onto purification and loading trap columns (Evosep, Evotips) according to the manufacturer's instructions. After loading, the Evotips were washed with 0.1% FA followed by a final wash with 100 μl of 0.1% FA and spinning for 10 s at 800g. The samples were then transferred into an Evosep One LC system for LC−MS/MS analyses. Peptides were analyzed on a timsTOF SCP using a standard DDA method with the following settings: 100−1,700 m/z, 1/k0 0.7−1.5, 100 ms ramp time, ten PASEF ramps, charge from 0−5, target intensity of 20,000 and intensity threshold of 500. An active PASEF precursor region was designed to exclude singly charged ions.

Data were analyzed on Fragpipe 22, using the LFQ-MBR setting, searching against a database containing murine histones and common contaminants. Variable modifications were K (methylation, dimethylation, trimethylation and acetylation), R (methylation and dimethylation), N-terminal (acetylation) and M (oxidation).

## BCG vaccination

Hypoxic lung injury mice were generated as per above. On day 8 or day 21 after returning to 21% O₂, mice received a single intravenous dose of 100 μl of BCG vaccine containing 2−8 × 10⁵ bacteria or 100 μl of PBS as vehicle control. Five weeks after returning to normoxia, the mice were infected via subcutaneous administration of *S. aureus*, as described above. At 48 h post-infection, c.f.u. were quantified in the abscess tissue, as described above. Peripheral blood was subjected to red blood cell lysis (BioLegend) and stained to measure H3K4me3 levels and H3 by flow cytometry, as described above. Neutrophils were identified based on Ly6G expression (BioLegend, 127628). The gating strategy was SSC-A/FSC-A > SSC-H/SSC-A > FSC-H/FSC-A > SSC-A/Live/Dead⁻ > Ly6G⁺.

## Statistics and reproducibility

Statistical analyses were performed using Prism v.10 software (Graph-Pad Software). A minimum of three samples were used to perform statistical tests, with sample size and specific tests detailed in the figure legends. No statistical methods were used to predetermine sample sizes but our sample sizes are similar to those reported in previous publications[7,12,20]. No data were excluded from the analyses. The experiments were not randomized. The Investigators were blinded to allocation during experiments and outcome assessment. The use of technical replicates is clarified in figure legends if applied. An animal not developing infection was removed from the experiment in Fig. 5a. Data were tested for normality and equal variances. Statistical significance was established by $P < 0.05$.

## Reporting summary

Further information on research design is available in the Nature Portfolio Reporting Summary linked to this article.

## Data availability

This study did not generate new unique reagents. The MS proteomics data have been deposited to the ProteomeXchange[76] Consortium via the PRIDE[77] partner repository with the dataset identifiers PXD065519 (ARDS survivors' cohort), PXD065461 (high-altitude study), PXD065562 (progenitor data) and supplementary data files. Sequencing data have been deposited at the Gene Expression Omnibus (https://www.ncbi.nlm.nih.gov/geo/) with the dataset identifiers GSE240723 (ChIP-seq) and GSE301027 (Cut&Run). Custom scripts for analysis can be accessed at https://zenodo.org/records/16921677 (ref. 66). Further information and requests for resources and reagents should be directed to, and will be fulfilled by, the lead contact, S.R.W. (sarah.walmsley@ed.ac.uk). Source data are provided with this paper.

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

## Acknowledgements

We thank the clinical support teams, patients and their families, who have contributed providing samples for this study. We also thank the CIR blood resource (21-EMREC-041) for the recruitment of healthy donors and the Altitude Physiology Expedition (APEX) 6 organizing members and expedition leaders, including S. Hattersley, E. Bennet, S. Lewis and D. Strončeková (Centre for Inflammation Research, Institute for Regeneration and Repair, University of Edinburgh, Edinburgh, UK and APEX, Sheffield, UK) for contributing to this study, as well as the donors involved. The APEX 6 Expedition was supported by APEX (registered charity number SC030345). This research used biological samples and data provided by The University of Sheffield from the Sheffield Teaching Hospitals Observational Study of Patients with Pulmonary Hypertension, Cardiovascular and Lung Disease, with support from the National Institute for Health Research (NIHR) Sheffield Clinical Research Facility. We thank the NIHR BioResource volunteers for their participation, and gratefully acknowledge NIHR BioResource centers, National Health Service (NHS) Trusts and staff for their contribution. We thank the NIHR, NHS Blood and Transplant, and Health Data Research UK as part of the Digital Innovation Hub Programme. This research was funded by Wellcome Trust Senior Clinical Fellowship awards 098516, 209220 (S.R.W.), Discovery science award 225778 (S.R.W.), UKRI/NIHR funding through the UK Coronavirus Immunology Consortium (UK-CIC) (S.R.W.), a Cancer Research UK cancer immunology project award C62207/A24495 (S.R.W.), Medical Research Council (MRC) SHIELD consortium (MRN0299X/1). Further funding was provided by a Wellcome Senior Clinical Research Fellowship to J.A.N. (215477/Z/19/Z). The Chan Zuckerberg Initiative, the Allen Institute, Gates MRI, The Gates Foundation provided funding to J.M., Y.N. and M.M.M. (M.M.M. is scientific founder of Lemba Tx). Also, a Wellcome Clinical Training Fellowship (108717) and Clinical Research Career Development Fellowship (224637/Z/21/Z) awarded to E.R.W., a Wellcome Clinical Training Fellowship award to T.M. (214383), a Wellcome Trust postdoctoral fellowship (110086) and iTPA grant (PIII052) awarded to A.S.M., support by Medical Research Foundation National Program in Antimicrobial Resistance studentship awarded to S.A. and funding by the MRC (MR/S022023/1 and MC_EX_MR/S022023/1) to N.M.R. supported this study. L.T. received a Wellcome Trust Fellowship (205228/Z/16/Z) and support by the UK Research and Innovation/NIHR through the UK Coronavirus Immunology Consortium and PHOSP-COVID. PHOSP-COVID is supported by a grant from the MRC-UK Research and Innovation and the Department of Health and Social Care through the NIHR rapid response panel to tackle COVID-19. L.T. was also supported by the US Food and Drug Administration Medical Countermeasures Initiative contract 75F40120C00085 and the NIHR Protection Research Unit in Emerging and Zoonotic Infections (NIHR200907) at University of Liverpool in partnership with the UK Health Security Agency, in collaboration with Liverpool School of Tropical Medicine and the University of Oxford. P.J.M.O. received support from NIHR, UKRI-CIC and PHOSP-COVID. M.G.S. was supported by the NIHR Protection Research Unit in Emerging and Zoonotic Infections (NIHR200907) at the University of Liverpool, NIHR UKRI-CIC and UK Health Security Agency. J.K.B. was supported by Wellcome Trust (223164), PHOSP-COVID, NIHR UKRI, Department of Health and Social Care (DHSC), Illumina, LifeArc, Sepsis Research (the Fiona Elizabeth Agnew Trust), The Intensive Care Society, Biotechnology and Biological Sciences Research Council (BBSRC) and the APEX charity. A.A.R.T. received a British Heart Foundation intermediated clinical fellowship (FS/18/13/33281) and support from the APEX charity. D.S. was a Cancer Research UK Career Development fellow (reference C47648/A20837), and work in his laboratory is also supported by an MRC university grant to the MRC Human Genetics Unit. D.R.M. was supported by the NIHR (NIHR-RP-2016-06-004) and J.D.K. was supported by the NIH-Oxford-Cambridge Scholars Program. Contributions by A.K. were supported by an MRC grant (MR/X01293X/1). The views expressed are those of the author(s) and not necessarily those of the UK NHS, the NIHR, the Department of Health, Department of Health and Social Care, or the UK Health Security Agency. We thank the flow cytometry and animal facilities at the Institute of Regeneration and Repair for their support during this work. We thank P. Puerto-Camacho for her editing support. Human silhouettes (Figs. 1a and 2a) and hiker (Fig. 4a) adapted from Wikimedia commons; lungs, neutrophil, mitochondrium, blood tube, mouse, cell diagrams adapted from Servier (https://bioicons.com) Servier (https://bioicons.com/, licensed under CC-BY 3.0 and public domain vectors (https://publicdomainvectors.org/, bound to Creative

## Author contributions

M.A.S.-G., P.S., B.M.O., N.W., Y.N., D.R.M., A.V.K., M.K.B.W., M.M.M, J.A.N. and S.R.W. conceived and designed the experiments. M.A.S.-G., P.S., B.M.O., N.W., Y.N., P.C., A.Z., C.P., A.J.M.H., R.S., J.D.K., J.M., T.M., B.T.S, L.R., A.T., H.A.T., R.D., R.C.H. and H.D.-S. performed the experiments. M.A.S.-G., P.S., B.M.O., N.W., Y.N., A.J.M.H., J.D.K., G.C., E.R.W., S.A., X.X., A.T. and A.J.B. analyzed the data. M.A.S.-G., P.S., B.M.O., N.W., Y.N., A.J.M.H., J.D.K., D.S., D.R.M., A.V.K, M.K.B.W., M.M.M, J.A.N. and S.R.W. interpreted the data. M.A.S.-G., P.C., D.M.G., A.S.M., A.M., A.D., G.C., K.D., D.H., O.V., A.W., I.P., S.G., N.M.R, L.T., P.J.M.O., M.G.S, P.-C.S.C.G., J.K.B., A.A.R.T., J.A.N. and S.R.W. facilitated obtaining human samples. S.R.W. obtained the funding. M.A.S.-G., M.M.M., J.A.N. and S.R.W. wrote the paper.

## Competing interests

The authors declare no competing interests.

## Additional information

**Extended data** is available for this paper at https://doi.org/10.1038/s41590-025-02301-9.

**Correspondence and requests for materials** should be addressed to Sarah R. Walmsley.

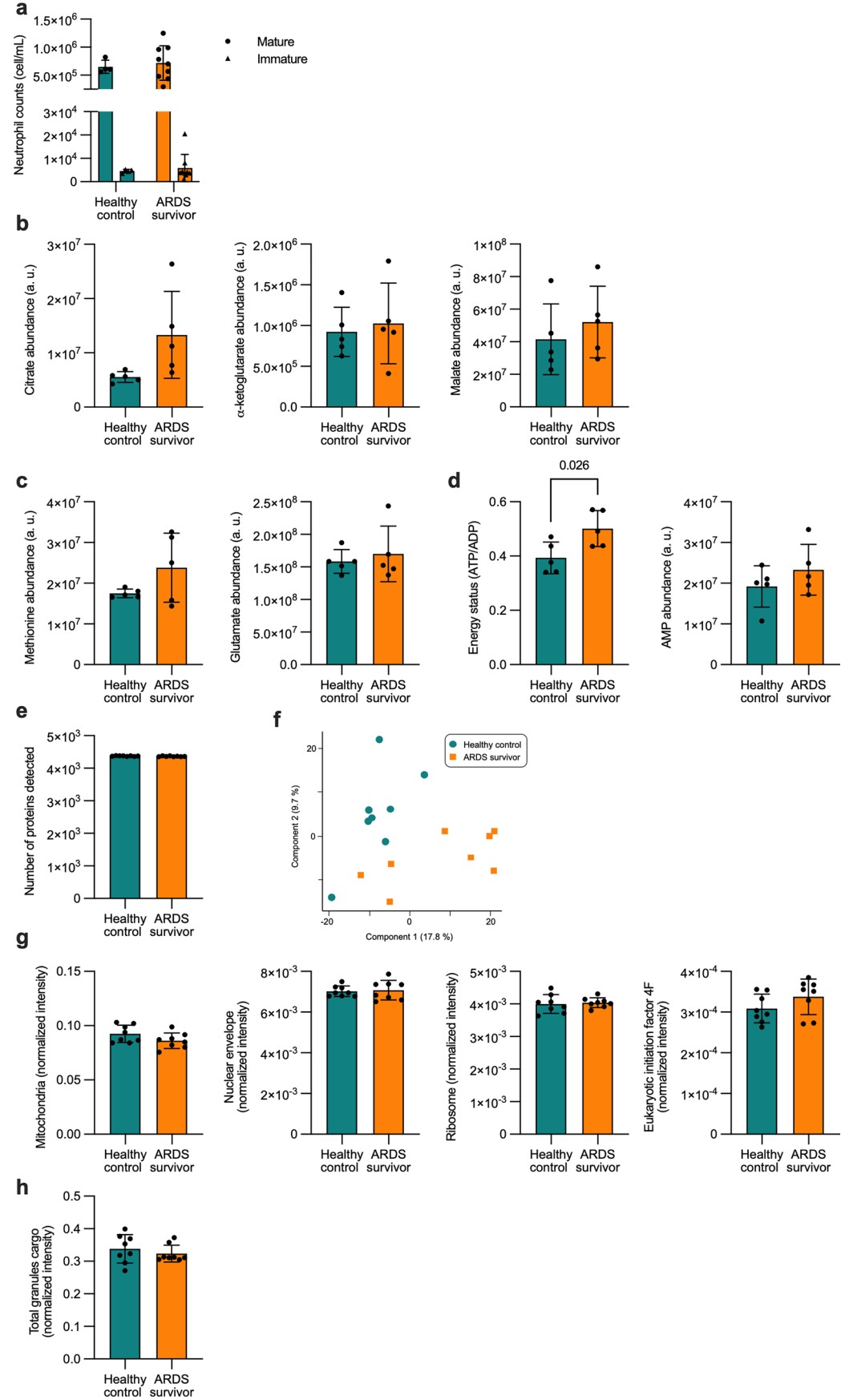

**Extended Data Fig. 1 | See next page for caption.**

**Extended Data Fig. 1 | Equivalent metabolic intermediaries and core molecular processes are observed in neutrophils months after ARDS. a–h**, Maturity, and metabolic and proteomic liquid chromatography-mass spectrometry analysis of circulating neutrophils harvested from healthy controls and acute respiratory distress syndrome (ARDS) survivors 3–6 months post-hospital admission. **a**. Mature (circle) and immature (triangle) blood neutrophil counts obtained by flow cytometry by analyzing CD16 surface expression (n = 4 healthy control and n = 9 ARDS survivor). **b**, Abundance of TCA metabolites. **c**, Abundance of metabolites related to one-carbon metabolism pathways. **d**, Energy status expressed as the ratio of ATP/ADP and AMP abundance (n = 5 for both experimental groups). **e**, Number of proteins identified by proteomic analysis. **f**, Principal-component analysis of proteomic data. **g**, Proteomic survey of the core cellular elements mitochondria (GO:0005739), nuclear envelope (GO:0005635), ribosome (Kyoto Encyclopedia of Genes and Genomes annotation 03010) and eukaryotic initiation factor 4 F. **h**, Total granule cargo proteins (GO:0035578, GO:0035580, and GO:1904724) (n = 8 for both sample groups). Data as mean ± s.d. (except **f**). Each value represents an individual. Significant p values depicted (for p < 0.05) and obtained by Shapiro–Wilk normality test followed by 2-way ANOVA test (**a**), two-tailed t-test (**b**–**e**, **g**) or two-tailed Mann–Whitney test (**h**).

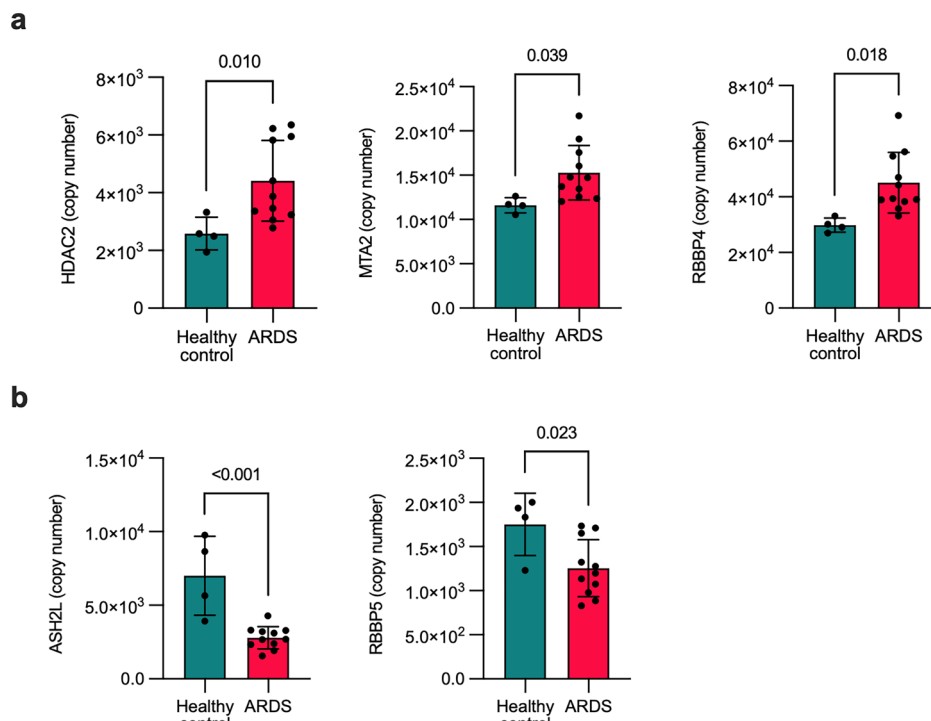

**Extended Data Fig. 2 | Circulating neutrophils from ARDS patients during the acute phase of the syndrome present changes in the epigenetic machinery.** **a**, **b**, Abundance of proteins measured in circulating neutrophils from healthy control and acute respiratory distress syndrome (ARDS) patients by chromatography-mass spectrometry analysis obtained from previous datasets[6] (n = 4 healthy control and n = 11 ARDS). Each value represents an individual. Data as mean ± s.d. Significant p values depicted (for p < 0.05) and obtained by Shapiro–Wilk normality test followed by two-tailed t-test (MTA2, RBBP4, ASH2L, RBBP5 (**a**, **b**)) or two-tailed Mann–Whitney test (HDAC2 (**a**)).

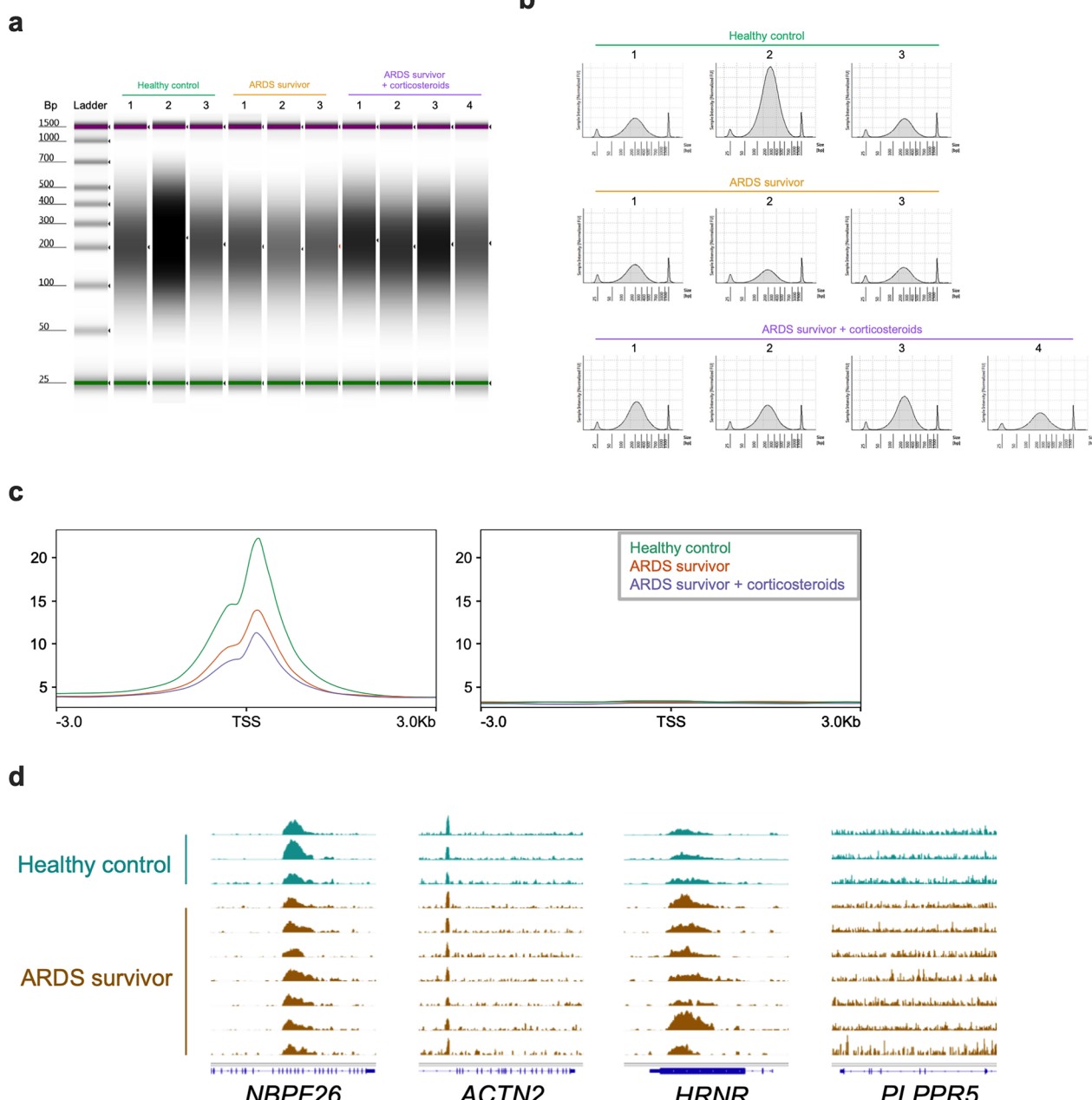

**Extended Data Fig. 3 | H3K4me3 peaks are enriched at the transcription start site and show preserved levels in certain genomic regions. a, b,** Tapestation profiles (**a**) of the fragment size distribution (**b**) detected in the input control samples from the healthy control group and acute respiratory distress syndrome (ARDS) survivor subgroups according to corticosteroid treatment. **c,** Representation of H3K4me3 levels when distributing 10000 bins around transcription start sites (TSS, left) or randomly across the genome (right) in the clinical groups included in (**a**). **d,** Individual gene tracks for unaltered H3K4me3 profiles in the sample groups included in (**a**). The number of biological replicates was healthy control (n = 3), ARDS survivor who did not receive corticosteroid treatment (n = 3), and corticosteroid-treated ARDS survivor (n = 4). Details on normalization and QC analysis in methods section.

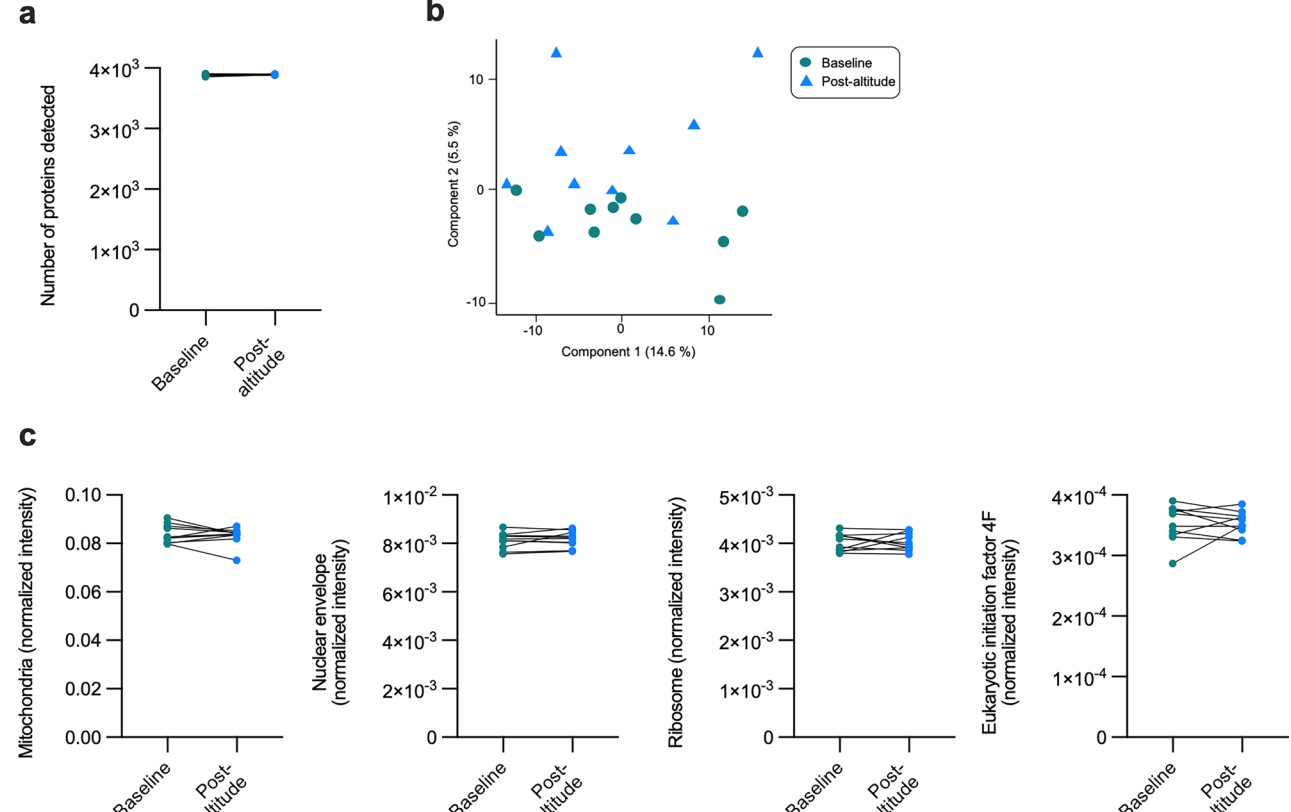

**Extended Data Fig. 4 | Proteomic study of blood neutrophils at baseline and after altitude-induced hypoxemia reveals preservation of core cellular processes. a–c**, Liquid chromatography-mass spectrometry analysis of the proteome of circulating neutrophils from healthy volunteers at baseline and 3–4 months after a 7-day period of altitude-induced hypoxemia (post-altitude) (n = 10 for both time points). **a**, Number of proteins identified. **b**, Principal-component analysis. **c**, Proteomic survey of the core cellular elements mitochondria (GO:0005739), nuclear envelope (GO:0005635), ribosome (Kyoto Encyclopedia of Genes and Genomes annotation 03010) and eukaryotic initiation factor 4 F. Each set of values linked through a line represents an individual. Significant p values depicted (for p < 0.05) and obtained by Shapiro–Wilk normality test followed by two-tailed paired t-test (**a**, **c**) or two-tailed Wilcoxon test (mitochondria (**c**)).

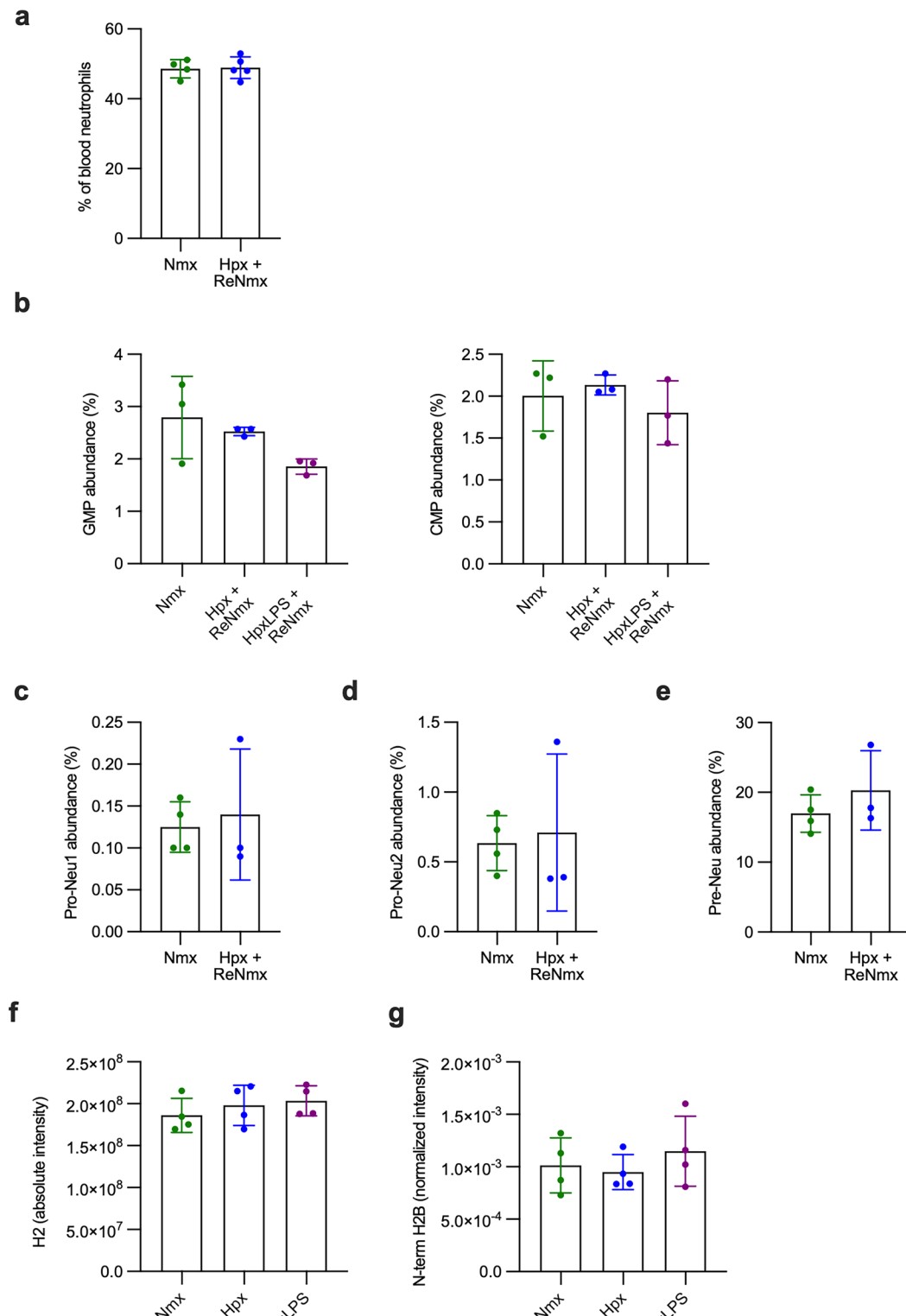

**Extended Data Fig. 5 | See next page for caption.**

**Extended Data Fig. 5 | Exposure to hypoxia does not affect the frequencies of circulating neutrophils upon infection, progenitor abundance or N-terminal H2B clipping in mice. a**, Circulating neutrophil abundance obtained by flow cytometry in mice exposed to 10 % $O_2$ for a week followed by a 5-week reoxygenation period prior to subcutaneous infection with *S. aureus* SH1000 (Hpx + ReNmx). Normoxia counterparts (Nmx) were used as experimental controls. Data expressed as a percentage from CD45$^+$ cells 48 h post-challenge (n = 4 Nmx and n = 5 Hpx + ReNmx). **b**, Granulocyte monocyte progenitor (GMP) and Common myeloid progenitor (CMP) abundance quantification by flow cytometry in the bone marrow of mice nebulized (HpxLPS + ReNmx) with LPS or not (Hpx + ReNmx) and exposed to 10 % $O_2$ for a week followed by a 5-week reoxygenation period. Normoxia counterparts (Nmx) were used as experimental controls (n = 3 for all experimental conditions). **c**–**e**, Quantification of bone marrow proNeu1, proNeu2 and preNeu from mice subjected to 10 % $O_2$ for a

week followed by a 3-month reoxygenation period (Hpx + ReNmx) compared to normoxia control mice (Nmx) obtained by flow cytometry as a proportion of Alive/Lin- cells (n = 4 Nmx and n = 3 Hpx + ReNmx). **f**, **g**, Mice were nebulized (HpxLPS) or not (Hpx) with LPS and subjected to 10 % $O_2$ for a week, with normoxia counterparts (Nmx) used as experimental controls. Bone marrow preNeu were harvested and lysed followed by chymotrypsin digestion. Liquid chromatography-mass spectrometry (LC−MS) was used to identify histone peptides. **f**, Total intensity for the peptides corresponding to histone 2 (H2). **g**, Abundance of N-terminal histone H2B peptide corresponding to amino acids 5-20 in the protein sequence (n = 4 for all experimental conditions). Data as mean ± s.d. Each value represents a mouse, except (**b**), with 4 mice pooled to obtain each data point. Significant p values depicted (for p < 0.05) and obtained by Shapiro-Wilk normality test followed by two-tailed t-test (**a**, **c**, **e**), 1-way ANOVA (**b**), two-tailed Mann–Whitney test (**d**), or 2-way ANOVA (**f**, **g**).

**a**

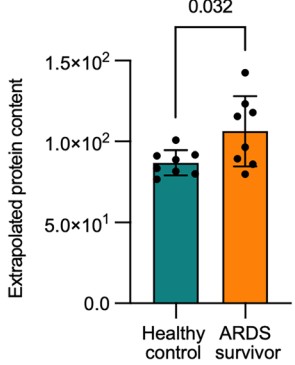
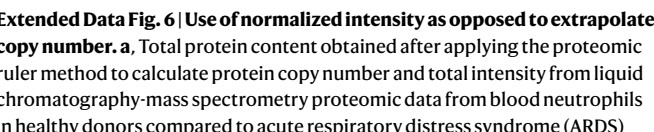
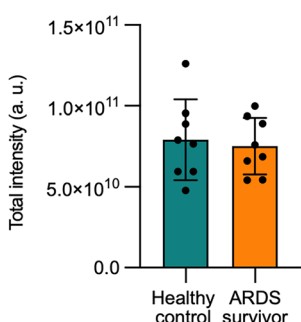

**Extended Data Fig. 6 | Use of normalized intensity as opposed to extrapolated copy number. a**, Total protein content obtained after applying the proteomic ruler method to calculate protein copy number and total intensity from liquid chromatography-mass spectrometry proteomic data from blood neutrophils in healthy donors compared to acute respiratory distress syndrome (ARDS) survivors 3–6 months post-hospital admission (n = 8 for both experimental groups). Data as mean ± s.d., with each value representing an individual. Significant p values depicted (for p < 0.05) and obtained by Shapiro-Wilk normality test followed by two-tailed t-test.

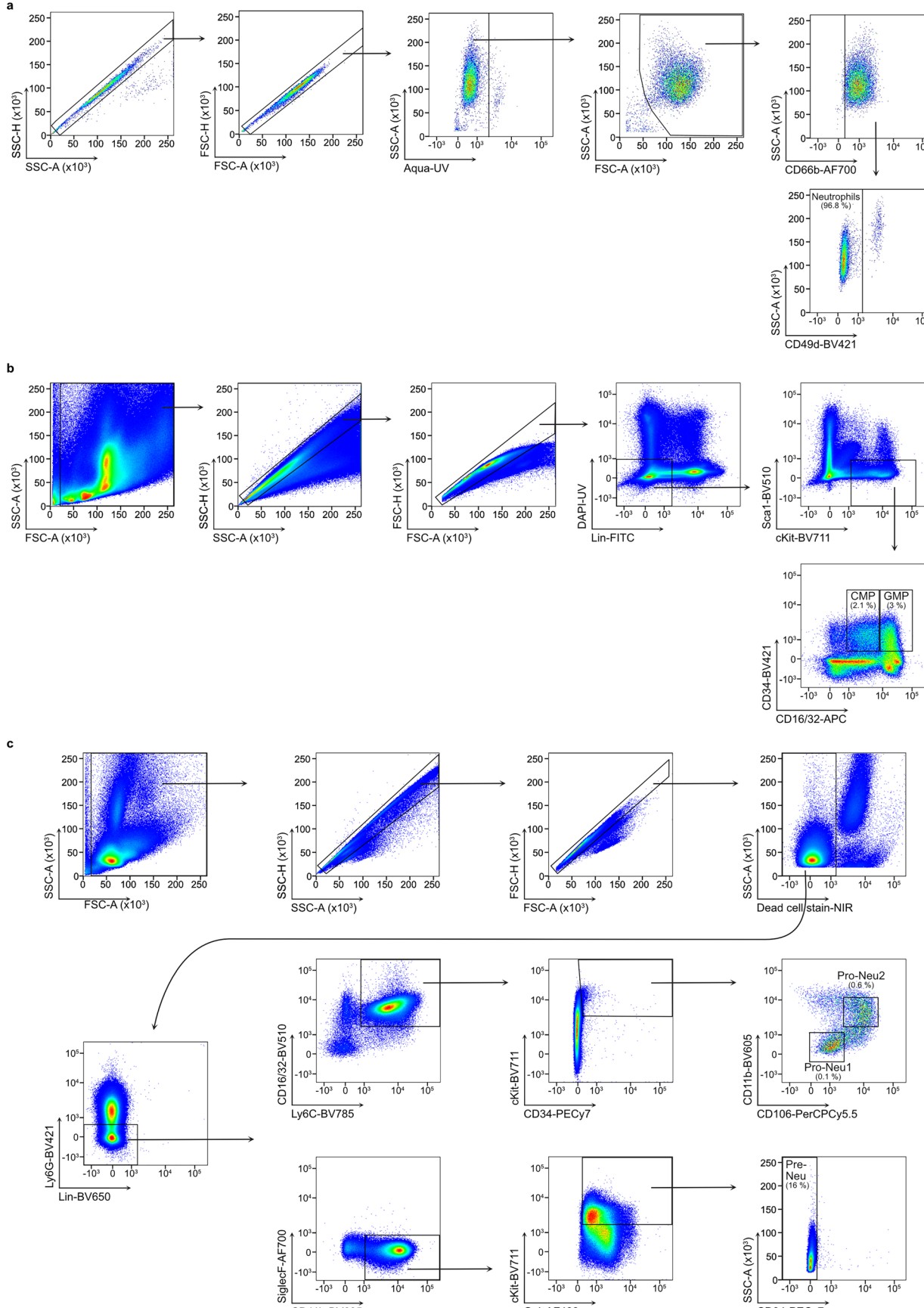

**Extended Data Fig. 7 | Representative gating strategy. a–c,** Pseudocolor plots containing gates depicting the sequential gating strategy followed for human blood neutrophil phenotypic and maturation experiments (**a**), mouse bone marrow granulocyte monocyte progenitor (GMP) and common myeloid progenitor (CMP) FACS sorting (**b**), and mouse bone marrow survey of proNeu1, proNeu2 and preNeu neutrophil-committed progenitors (**c**) for identification by flow cytometry or FACS. Percentages refer to Alive (**a**) or Alive/Lin- (**b**, **c**) gates.

# Reporting Summary

## Statistics

For all statistical analyses, confirm that the following items are present in the figure legend, table legend, main text, or Methods section.

| n/a | Confirmed | |
|---|---|---|
| ☐ | ☒ | The exact sample size (*n*) for each experimental group/condition, given as a discrete number and unit of measurement |
| ☐ | ☒ | A statement on whether measurements were taken from distinct samples or whether the same sample was measured repeatedly |
| ☐ | ☒ | The statistical test(s) used AND whether they are one- or two-sided<br>*Only common tests should be described solely by name; describe more complex techniques in the Methods section.* |
| ☒ | ☐ | A description of all covariates tested |
| ☐ | ☒ | A description of any assumptions or corrections, such as tests of normality and adjustment for multiple comparisons |
| ☐ | ☒ | A full description of the statistical parameters including central tendency (e.g. means) or other basic estimates (e.g. regression coefficient) AND variation (e.g. standard deviation) or associated estimates of uncertainty (e.g. confidence intervals) |
| ☐ | ☒ | For null hypothesis testing, the test statistic (e.g. *F*, *t*, *r*) with confidence intervals, effect sizes, degrees of freedom and *P* value noted<br>*Give P values as exact values whenever suitable.* |
| ☒ | ☐ | For Bayesian analysis, information on the choice of priors and Markov chain Monte Carlo settings |
| ☒ | ☐ | For hierarchical and complex designs, identification of the appropriate level for tests and full reporting of outcomes |
| ☒ | ☐ | Estimates of effect sizes (e.g. Cohen's *d*, Pearson's *r*), indicating how they were calculated |

*Our web collection on statistics for biologists contains articles on many of the points above.*

## Software and code

Policy information about availability of computer code

| Data collection | Custom scripts generated have been deposited in Zenodo. Details for access included in the manuscript. |
|---|---|
| Data analysis | Proteomics data was analysed with Spectronaut 14 and 16 using the directDIA option.<br>PCA plots were generated with Perseus software.<br>Metabolomics analysis was performed with Skyline software (21.1).<br>ChIPseq data was processed and analysed by using Trim Galore (0.6.7), HISAT (2.2.1), bedtools (2.30), samtools (1.15), Picard (3.1.0), ngs.plot (2.61), MACS3 (3.0.0.a6), custom scripts of the R packages DiffBind, ChIPseeker, and ReactomePA, HOMER (4.11), and a custom Python 3.10.8 script.<br>Cut&Run data was generated aligning trimmed reads with BWA and peaks called by MACS2.<br>FlowJo (10.8.1) was used to analyse flow cytometry data.<br>Data was analysed with Graphpad prism (10.5.0). |

For manuscripts utilizing custom algorithms or software that are central to the research but not yet described in published literature, software must be made available to editors and reviewers. We strongly encourage code deposition in a community repository (e.g. GitHub). See the Nature Portfolio guidelines for submitting code & software for further information.

# Data

Policy information about availability of data

All manuscripts must include a data availability statement. This statement should provide the following information, where applicable:
- Accession codes, unique identifiers, or web links for publicly available datasets
- A description of any restrictions on data availability
- For clinical datasets or third party data, please ensure that the statement adheres to our policy

> Proteomic data from ARDS survivors, high-altitude study volunteers and preNeu has been deposited in PRIDE and accession codes are provided. ChiPseq and Cut&Run data have been uploaded to GEO and accession codes are provided. Accession tokens have been generated and are readily available.

# Research involving human participants, their data, or biological material

Policy information about studies with human participants or human data. See also policy information about sex, gender (identity/presentation), and sexual orientation and race, ethnicity and racism.

| | |
|---|---|
| Reporting on sex and gender | Human data from both sexes were included in this study and recorded by the clinical teams upon hospital visit or as self-reported by the participant. No sex stratification studies have been performed. The acute respiratory disease syndrome patient cohort included 52 individuals, with a male:female ratio of 32:20. The altitude induced hypoxemia cohort included 20 participants, with a male:female ratio of 8:12. The healthy donors used as control group was integrated by 33 participants, with a male:female ratio of 10:23. We have ethical approval to show individual anonymized data. |
| Reporting on race, ethnicity, or other socially relevant groupings | We have not performed any analysis based on race, ethnicity or any other socially relevant grouping. |
| Population characteristics | The population characteristics for the acute respiratory disease syndrome patient cohort are collated in Table 1. The average age of the participants involved in the altitude induced hypoxia study was 22 years old. The average age range of the healthy donors used as controls was 31-40 years old. No extra characteristics were allowed to be recorded from either the altitude induced hypoxia or the control cohorts. |
| Recruitment | Patient recruitment took place from April 2020-April 2025 mainly at The Royal Infirmary of Edinburgh, UK, the University of Cambridge and the University of Sheffield. The acute respiratory distress syndrome patients included in this study were males and females diagnosed according to the Berlin criteria. All sequential samples from patients identified based on the ethics inclusion criteria were processed. Survivors were samples 3-6 months post-hospital admission. This study was performed with informed consent.<br>The altitude induced hypoxemia cohort was restricted to participants who had voluntarily enrolled in the Apex6 expedition and written consent was taken from all of them. |
| Ethics oversight | Acute respiratory distress syndrome studies were done through the ARDS Neut study (20/SS/0002) and in association with the PHOSP study (20/YH/0225), the University of Cambridge (17/EE/0025) and the University of Sheffield (18/YH/0441).<br>The altitude induced hypoxia study was approved by the Edinburgh Medical School Research Ethics Committee (EMREC, 21-EMREC-043).<br>The healthy donors blood donations were regulated by the University of Edinburgh Centre for Inflammation Research Blood Resource Management Committee (AMREC 15-HV-013, 21-EMREC-041). |

Note that full information on the approval of the study protocol must also be provided in the manuscript.

# Field-specific reporting

Please select the one below that is the best fit for your research. If you are not sure, read the appropriate sections before making your selection.

☒ Life sciences      ☐ Behavioural & social sciences      ☐ Ecological, evolutionary & environmental sciences

For a reference copy of the document with all sections, see nature.com/documents/nr-reporting-summary-flat.pdf

# Life sciences study design

All studies must disclose on these points even when the disclosure is negative.

| | |
|---|---|
| Sample size | Groups of at least 3 biological replicates were used in each experiment in each treatment or experimental group. Data from individual replicate experiments were pooled as detailed in the figure legends. No sample calculations were performed pre-hoc. Sample size was based on pilot experiments and previous experience with similar experiments in the laboratory. |
| Data exclusions | One of the mice in the experiment detailed in Figure 5b-c was not included in the analysis as did not develop infection at all. |
| Replication | Experiments were replicated and pooled as detailed in the figure legends. |

| | |
|---|---|
| Randomization | For human studies, due to limited sample volumes, cell availability and high demands on cell number for "omics assays", it was not possible to conduct every assay on each patient sampled. Assays were performed sequentially as samples became available/cell number permitting with no prior selection.<br>In mouse studies, mice were randomly allocated to normoxic and hypoxic groups. Mixing of cages was not permitted where male mice were used but were purchased as part of the same cohort. Mice were age- and sex-matched within and between experiments. |
| Blinding | Experimental processing, data acquisition, and data analysis were performed on a blinded way whenever possible. |

# Reporting for specific materials, systems and methods

We require information from authors about some types of materials, experimental systems and methods used in many studies. Here, indicate whether each material, system or method listed is relevant to your study. If you are not sure if a list item applies to your research, read the appropriate section before selecting a response.

## Materials & experimental systems

| n/a | Involved in the study |
|---|---|
| ☐ | ☒ Antibodies |
| ☒ | ☐ Eukaryotic cell lines |
| ☒ | ☐ Palaeontology and archaeology |
| ☐ | ☒ Animals and other organisms |
| ☒ | ☐ Clinical data |
| ☒ | ☐ Dual use research of concern |
| ☒ | ☐ Plants |

## Methods

| n/a | Involved in the study |
|---|---|
| ☐ | ☒ ChIP-seq |
| ☐ | ☒ Flow cytometry |
| ☒ | ☐ MRI-based neuroimaging |

## Antibodies

| | |
|---|---|
| Antibodies used | CD16 eBioCB16 11-0168-42 2253005 FITC Ebioscience 1:200<br>CD66b G10F5 305114 B292150 AF700 Biolegend 1:200<br>CD62L DREG-56 304814 B291262 APC-Cy7 Biolegend 1:200<br>CD49d 9F10 304322 B302258 BV421 Biolegend 1:200<br>CD45 30-F11 103128 B380021 AF700 Biolegend 1:200<br>Ly6G 1A8 127628 B431781 BV421 Biolegend 1:200<br>Fc block 93 101320 B411434 Unconjugated Biolegend 1:100<br>H3K4me3 C42D8 9751S 15 Unconjugated Cell Signaling 1:50<br>H3K4me3 Polyclonal 07-473 2289139 Unconjugated Upstate 1:10000<br>Goat anti-rabbit Polyclonal 925-32211 D00804-06 IRDye 800CW Li-cor 1:10000<br>H4 Sp2/0-Ag14 Ab31830 GR3204774-3 Unconjugated Abcam 1:1000<br>Goat anti-mouse Polyclonal 925-68070 C90910-20 IRDye 680RD Li-cor 1:1000<br>CD16/32 93 101326 B418060 APC Biolegend 1:200<br>Lineage n/a 78022 B281150 FITC Biolegend 1:200<br>Sca1 D7 108129 B444060 BV510 Biolegend 1:100<br>c-Kit 2B8 105835 B249350 BV711 Biolegend 1:200<br>CD34<br>SA376A4 152208 B419071 BV421 Biolegend 1:200<br>CD16/32 93 101333 B418160 BV510 Biolegend 1:100<br>CD90.2 53-2.1 13-0902-82 3020994 Biotin Ebioscience 1:200<br>B220 (CD45R) RA3-6B2 13-0452-82 2899900 Biotin Ebioscience 1:200<br>NK1.1 PK136 13-5941-82 2791123 Biotin Ebioscience 1:200<br>Sca1 D7 13-5981-82 2993028 Biotin Ebioscience 1:200<br>Flt3 A2F10 13-1351-82 2924264 Biotin Ebioscience 1:200<br>CD115 AFS98 13-1152-82 2986854 Biotin Ebioscience 1:200<br>Ter119 TER-119 13-5921-82 2609055 Biotin Ebioscience 1:200<br>CD34<br>SA376A4 152218 B438235 PE-Cy7 Biolegend 1:100<br>Ly6G 1A8 127628 B431781 BV421 Biolegend 1:200<br>CD11b M1/70 101257 B431815 BV605 Biolegend 1:200<br>c-Kit 2B8 105835 B429186 BV711 Biolegend 1:200<br>CD106 429 105716 B399866 PerCP-Cy5.5 Biolegend 1:200<br>SiglecF S17007L 155534 NB424840 AF700 Biolegend 1:200<br>Ly6C Hk1.4 128041 B406937 BV785 Biolegend 1:500<br>Gr1 RB6-8C5 108417 B410115 AF488 Biolegend 1:500<br>H3 EPR17785 ab313347 1076340-2 AF647 Abcam 1:500<br>H3K4me3 EPR20551-225 ab237342 1027184-1 PE Abcam 1:1000 |
| Validation | All the antibodies were used according to manufacturer's guidelines.<br>CD16 11-0168-42 Ebioscience Santana-Hernandez S, et al. 2024. J Exp Clin Cancer Res.<br>CD66b 305114 Biolegend Giamarellos-Bourboulis EJ, et al. 2020. Cell. |

CD62L 304814 Biolegend Ajith A, et al. 2021. Front Immunol.
CD49d 304322 Biolegend Qi Q, et al. 2020. Blood.
CD45 103128 Biolegend Radtke AJ, et al. 2022. Nat Protoc.
Ly6G 127628 Biolegend Hutter K, et al. 2022. Front Immunol.
Fc block 101320 Biolegend Ajith A, et al. 2021. Front Immunol.
H3K4me3 9751S Cell Signaling Noshita et al. 2023. J Cell Sci.
H3K4me3 07-473 Upstate Suijker et al. 2015. Oncotarget
Goat anti-rabbit 925-32211 Li-cor This antibody was tested by dot blot and and/or solid-phase. The conjugate has been specifically tested and qualified for Western blot and In-Cell Western™ Assay applications.
H4 Ab31830 Abcam Samejima et al. 2022. Mol Cell.
Goat anti-mouse 925-68070 Li-cor This antibody was tested by dot blot and and/or solid-phase. The conjugate has been specifically tested and qualified for Western blot and In-Cell Western™ Assay applications.
CD16/32 101326 Biolegend Lopez DA, et al. 2022. Cell Rep.
Lineage 78022 Biolegend Mirchandani AS, et al. 2022. Nat Immunol.
Sca1 108129 Biolegend Sandovici I, et al. 2022. Dev Cell.
c-Kit 105835 Biolegend Schönberger K, et al. 2022. Cell Stem Cell
CD34 152208 Biolegend Vanneste D, et al. 2023. Nat Immunol.
CD16/32 101333 Biolegend Al-Rifai R, et al. 2022. Nat Commun.
CD90.2 13-0902-82 Ebioscience Bi R, et al. 2023. Nat Commun.
B220 (CD45R) 13-0452-82 Ebioscience Vos WG, et al. 2024. Front. Immunol.
NK1.1 13-5941-82 Ebioscience Chen Y, et al. 2024. Cell Commun Signal.
Sca1 13-5981-82 Ebioscience Griffin KH, et al. 2023. PNAS.
Flt3 13-1351-82 Ebioscience Qiu J, et al. 2024. Stemm Cell Res Ther.
CD115 13-1152-82 Ebioscience Eren RO, et al. 2024. Nat Commun.
Ter119 13-5921-82 Ebioscience Tundidor I, et al. 2023. Nat Commun.
CD34 152218 Biolegend Biolegend's QC testing. Xu Y, et al. 2023. EMBO Rep. for same clone ref.
Ly6G 127628 Biolegend Hutter K, et al. 2022. Front Immunol.
CD11b 101257 Biolegend Gallizioli M, et al. 2020. Cell Rep.
c-Kit 105835 Biolegend Schönberger K, et al. 2022. Cell Stem Cell.
CD106 105716 Biolegend Nahrendorf W, et al. 2021. eLife.
SiglecF 155534 Biolegend Biolegend's QC testing. Senatus L, et al. 2023. Commun Biol. for same clone ref.
Ly6C 128041 Biolegend Grandjean CL, et al. 2021. Sci Adv.
Gr1 108417 Biolegend Haratani K, et al. 2019. J Clin Invest.
H3 ab313347 Abcam Abcam's flow cytometry testing service
H3K4me3 ab237342 Abcam Abcam's flow cytometry testing service

## Animals and other research organisms

Policy information about studies involving animals; ARRIVE guidelines recommended for reporting animal research, and Sex and Gender in Research

| Laboratory animals | Animal experiments were conducted in accordance with the UK Home Office Animals (Scientific Procedures) Act of 1986. All animal studies were approved by The University of Edinburgh Animal Welfare and Ethical Review Board, adhered to the principles of "3Rs" (replacement, reduction, refinement), and complied with ARRIVE guidelines for animal research. Mice of 3-6 months old were used for these studies. Mice were housed in IVC cages under 12 h light/darkness cycles and controlled temperature (20-23°C) in accordance with UK Home Office guidance. All mice used for experiments were healthy with quarterly and annual testing carried out in accordance with FELASA 2014 Guidelines, using a mixture of environmental, random colony samples and sentinel testing by serology and PCR. Mice had ad libitum access to food and water. |
|---|---|
| Wild animals | N/A |
| Reporting on sex | 92 male mice were used in this study. According to our experience using LPS models and to the literature (Aziz et al. 2007, Mock et al. 2023), age and weight of the mice may have an impact on inflammatory outcomes but sex, per se, does not have a significant effect. Additionally, any changes in outcomes are in magnitude or duration rather than character. We therefore performed experiments only in male mice to avoid potential magnitude variability and reduce the number of animals needed for our studies, in keeping with the 3R perspective on experimental animal usage. |
| Field-collected samples | N/A |
| Ethics oversight | All animal studies were approved by The University of Edinburgh Animal Welfare and Ethical Review Board. |

Note that full information on the approval of the study protocol must also be provided in the manuscript.

## Plants

| Seed stocks | N/A |
|---|---|
| Novel plant genotypes | N/A |

| Authentication | N/A |
| --- | --- |

# ChIP-seq

## Data deposition

☒ Confirm that both raw and final processed data have been deposited in a public database such as GEO.

☒ Confirm that you have deposited or provided access to graph files (e.g. BED files) for the called peaks.

**Data access links**
*May remain private before publication.*

GSE240723

**Files in database submission**

H3K4me3: .bigwig, .bed, and .fastq files
2_H3K4me3-sort-bl-dedupl-downscaled-norm.bw
3_H3K4me3-sort-bl-dedupl-downscaled-norm.bw
4_H3K4me3-sort-bl-dedupl-downscaled-norm.bw
8_H3K4me3-sort-bl-dedupl-downscaled-norm.bw
9_H3K4me3-sort-bl-dedupl-downscaled-norm.bw
10_H3K4me3-sort-bl-dedupl-downscaled-norm.bw
11_H3K4me3-sort-bl-dedupl-downscaled-norm.bw
13_H3K4me3-sort-bl-dedupl-downscaled-norm.bw
14_H3K4me3-sort-bl-dedupl-downscaled-norm.bw
15_H3K4me3-sort-bl-dedupl-downscaled-norm.bw
2_H3K4me3-sort-bl-dedupl-downscaled_2.bam.bed
3_H3K4me3-sort-bl-dedupl-downscaled_3.bam.bed
4_H3K4me3-sort-bl-dedupl-downscaled_4.bam.bed
8_H3K4me3-sort-bl-dedupl-downscaled_3.bam.bed
9_H3K4me3-sort-bl-dedupl-downscaled_4.bam.bed
10_H3K4me3-sort-bl-dedupl-downscaled_5.bam.bed
11_H3K4me3-sort-bl-dedupl-downscaled_1.bam.bed
13_H3K4me3-sort-bl-dedupl-downscaled_3.bam.bed
14_H3K4me3-sort-bl-dedupl-downscaled_4.bam.bed
15_H3K4me3-sort-bl-dedupl-downscaled_5.bam.bed
WTCHG_917224_73305306_1.fastq.gz
WTCHG_917224_73305306_2.fastq.gz
WTCHG_917224_73315307_1.fastq.gz
WTCHG_917224_73315307_2.fastq.gz
WTCHG_917224_73325308_1.fastq.gz
WTCHG_917224_73325308_2.fastq.gz
WTCHG_917224_73365312_1.fastq.gz
WTCHG_917224_73365312_2.fastq.gz
WTCHG_917224_73375313_1.fastq.gz
WTCHG_917224_73375313_2.fastq.gz
WTCHG_917224_73385314_1.fastq.gz
WTCHG_917224_73385314_2.fastq.gz
WTCHG_917224_73395315_1.fastq.gz
WTCHG_917224_73395315_2.fastq.gz
WTCHG_917224_73415317_1.fastq.gz
WTCHG_917224_73415317_2.fastq.gz
WTCHG_917224_73425318_1.fastq.gz
WTCHG_917224_73425318_2.fastq.gz
WTCHG_917224_73435319_1.fastq.gz
WTCHG_917224_73435319_2.fastq.gz

Input controls: .fastq files
WTCHG_917224_73145290_1.fastq.gz
WTCHG_917224_73145290_2.fastq.gz
WTCHG_917224_73155291_1.fastq.gz
WTCHG_917224_73155291_2.fastq.gz
WTCHG_917224_73165292_1.fastq.gz
WTCHG_917224_73165292_2.fastq.gz
WTCHG_917224_73205296_1.fastq.gz
WTCHG_917224_73205296_2.fastq.gz
WTCHG_917224_73215297_1.fastq.gz
WTCHG_917224_73215297_2.fastq.gz
WTCHG_917224_73225298_1.fastq.gz
WTCHG_917224_73225298_2.fastq.gz
WTCHG_917224_73235299_1.fastq.gz
WTCHG_917224_73235299_2.fastq.gz
WTCHG_917224_73255301_1.fastq.gz
WTCHG_917224_73255301_2.fastq.gz
WTCHG_917224_73265302_1.fastq.gz
WTCHG_917224_73265302_2.fastq.gz

WTCHG_917224_73275303_1.fastq.gz
WTCHG_917224_73275303_2.fastq.gz

Genome browser session
(e.g. UCSC)

IGV

## Methodology

| | |
|---|---|
| Replicates | 3 independent samples from healthy donors<br>3 independent samples from ARDS survivors 3-6 months post-hospital admission not treated with dexamethasone<br>4 independent samples from ARDS survivors 3-6 months post-hospital admission treated with dexamethasone |
| Sequencing depth | 75-bp paired-end sequencing. HiSeq 4000 platform (Illumina) |
| Antibodies | Refer to Table 2.<br>H3K4me3 C42D8 9751S 15 Cell Signaling 1:50 |
| Peak calling parameters | Raw sequencing reads were first quality trimmed using Trim Galore 0.6.7 using standard settings. Read mapping was performed with HISAT2 2.2.1 against the human genome (hg19) in paired-end mode. Reads mapping to blacklisted regions were removed with the bedtools 2.30 intersect tool and sorted by coordinate and indexed using samtools 1.15 sort and index, respectively. From the resulting BAM files, duplicated reads were removed using the MarkDuplicates tool from Picard with the setting REMOVE_DUPLICATES=TRUE. Any potential differences in sequencing depth were corrected by down-sampling all samples to the same number of fragments. Corresponding input files were used as controls. For each individual BAM file, H3K4me3 peaks were identified using the MACS3 3.0.0.a6 callpeak function with settings -q 0.05 -g hs -f BAMPE --broad --broad-cutoff 0.1. |
| Data quality | Successful sample fragmentation was confirmed by analysis with TapeStation (Agilent). MACS3 3.0.0.a6 callpeak function was used with settings -q 0.05 -g hs -f BAMPE --broad --broad-cutoff 0.1. Differential H3K4me3 peak analysis between conditions, and generation of the H3K4 signal heatmap, PCA plot, sample distance heatmap and enriched pathway analysis of differential peaks was performed with R 4.2.0 using the Bioconductor packages DiffBind, ChIPseeker, clusterProfiler, and ReactomePA. Any potential differences in sequencing depth were corrected by down-sampling all samples to the same number of fragments.<br>BAM files were merged per group with samtools merge to create subsequent metagene H3K4me3 profiles with ngs.plot 2.6163. This software performs 2 normalization steps: the coverage vectors (i. e. gene regions) are normalized to be equal length, and the vectors are normalized against the corresponding library size (i. e. the total read count for the reads that pass quality filters) to generate the Reads Per Million (RPM) mapped reads. Additionally, data from each individual was normalized by its corresponding input sample.<br>We represented 10000 bins around transcription start sites, which confirmed a higher concentration of peaks at these genomic areas across study groups, as expected according to previous studies. The random distribution of 10000 bins along the DNA revealed equivalent low abundance profiles. Also, despite the widespread reduction in H3K4me3, preserved levels of H3K4me3 were detected in certain genomic regions including the genes NBPF26, ACTN2, HRNR, and PLPRP5. |
| Software | Raw sequencing reads were first quality trimmed using Trim Galore 0.6.7.<br>Read mapping was performed with HISAT2 2.2.1 against the human genome (hg19) in paired-end mode.<br>Reads mapping to blacklisted regions were removed with the bedtools 2.30 intersect tool and sorted by coordinate and indexed using samtools 1.15.<br>Duplicated reads were removed using the MarkDuplicates tool from Picard.<br>BAM files were merged with samtools to create metagene H3k4me3 profiles with ngs.plot 2.61.<br>H3K4me3 peaks were identified using the MACS3 3.0.0.a6<br>H3K4 signal heatmap, PCA plot, sample distance heatmap and enriched pathway analysis of differential peaks was performed with R 4.2.0 using the Bioconductor packages DiffBind, ChIPseeker, and ReactomePA with prior annotation with HOMER 4.11.<br>Data was visualized with IGV. |

# Flow Cytometry

## Plots

Confirm that:

☒ The axis labels state the marker and fluorochrome used (e.g. CD4-FITC).

☒ The axis scales are clearly visible. Include numbers along axes only for bottom left plot of group (a 'group' is an analysis of identical markers).

☒ All plots are contour plots with outliers or pseudocolor plots.

☒ A numerical value for number of cells or percentage (with statistics) is provided.

## Methodology

| | |
|---|---|
| Sample preparation | For human studies, neutrophils were stained with Zombie Aqua Fixable viability dye (Biolegend) for 15 min at room temperature to be able to exclude dead cells. Human Fc block (Biolegend) was used for 15 min at 4 degrees C to avoid unspecific subsequent antibody binding. Cells were next stained with an antibody mix (Table 2) for 30 min at 4 degrees C. FACS buffer (2 % v/v FBS in DPBS) was used as vehicle for Fc block and antibody mixes and for washes after staining. Stained neutrophils were fixed with 4 % PFA for 15 min on ice and washed before data acquisition.<br>For mouse studies, cells were harvested from the bone marrow and stained against CD16/32 (Biolegend) for 20 minutes followed by a lineage antibody cocktail 20 minute incubation. Linage positive events are removed by using magnetic beads and columns (Miltenyi) or directly based on fluorescence. A specific antibody cocktail was then used for 20 minutes followed by live/dead staining (Thermo Scientific) before running into the flow cytometer. |

| | |
|---|---|
| Instrument | BD LSRFortessa flow cytometer (Beckton Dickinson). BD FACSAria Fusion cell sorter. |
| Software | FlowJo (10.8.1) was used to analyse flow cytometry data. |
| Cell population abundance | Cell abundances are included in Extended Data Fig. 5. |
| Gating strategy | Human neutrophil counts: SSC-H/SSC-A> FSC-H/FSC-A> SSC-A/FSC-A (Beads and Granulocytes). Any eosinophils were excluded from the granulocytes gate based on their autofluorescent properties, as previously reported. |
| | Human neutrophil phenotyping: SSC-H/SSC-A> FSC-H/FSC-A> SSC-A/Zombie Aqua>SSC-A/FSC-A (Cells)>CD66b+ CD49d->CD66b, CD16, and CD62L. |
| | Ex vivo phagocytosis: SSC-H/SSC-A> FSC-H/FSC-A> SSC-A/FSC-A (Neutrophils)>CFS-E+ |
| | Mouse infection blood counts: SSC-H/SSC-A> FSC-H/FSC-A> SSC-A/Zombie Aqua> SSC-A/FSC-A> SSC-A/CD45+> SSC-A/Ly6G+ |
| | Cut&Run: SSC-A/FSC-A> SSC-H/SSC-A> FSC-H/FSC-A> Zombie Aqua/Lin-> cKit+/Sca1-> CD34+/CD16/32high (GMP) or CD34+/CD16/32low (CMP). |
| | H3K4me3: SSC-A/FSC-A> SSC-H/SSC-A> FSC-H/FSC-A> SSC-A/Live/Dead-> Ly6G-/Lin-> CD16/32high/Ly6C+> cKithigh/CD34+> CD11blow/CD106- (ProNeu1) or CD11bhigh/CD106+ (ProNeu2). SSC-A/FSC-A> SSC-H/SSC-A> FSC-H/FSC-A> SSC-A/Live/Dead-> Ly6G-/Lin-> SiglecF-/CD11b+> cKit+/Gr1+> SSC-A /CD34- (PreNeu). |
| | PreNeu sorting: SSC-A/FSC-A> SSC-H/SSC-A> FSC-H/FSC-A> SSC-A/Live/Dead-> Ly6G-/Lin-> SiglecF-/CD11b+> cKit+/Gr1+> SSC-A /CD34- |
| | BCG vaccination study: SSC-A/FSC-A> SSC-H/SSC-A> FSC-H/FSC-A> SSC-A/Live/Dead-> Ly6G+ |

☒ Tick this box to confirm that a figure exemplifying the gating strategy is provided in the Supplementary Information.

