## [Peer Review File · Nature Immunology]

Hypoxia induces histone clipping and H3K4me3 loss in neutrophil progenitors resulting in long-term impairment of neutrophil immunity

Corresponding Author: Professor Sarah Walmsley

Version 0:

Reviewer comments:

Reviewer #1

(Remarks to the Author)

In this study, the authors perform molecular and functional analyses of neutrophils from acute ARDS patients or from ARDS survivors, as well as from individuals exposed to high altitudes. They propose that neutrophils in hypoxic conditions exhibit persistent changes in the expression and genomic activities of factors controlling histone methylation, leading to loss of this mark in differentiated cells and to subsequent dysfunction. This is an interesting idea that, in my view, is not sufficiently supported by the data.

As a general comment, this study seems interesting but is preliminary. Several claims are based on correlative evidence or on over-interpretation of the data. I will only focus on those major aspects in this review.

- Do neutrophils from ARDS survivors express lower levels of chromatin regulators? Is loss of H3K4me3 pervasive or selective? If pervasive, this should be quantified using spike-ins to exclude technical artifacts. If selective, unchanged or up-regulated regions in ARDS patients should be used as reference for differential enrichment analyses. The presented analyses are superficial and poorly informative.

- Functional differences between neutrophils from controls and post-ARDS patients are very small. Data are shown as fold change, which over-emphasize differences between groups; even so, fold change differences are almost negligible and poorly significant. Data in Extended Data Fig. 2h or Fig. 5g show no actual differences in *S. aureus* phagocytosis, contrary to what the authors claim. Even if one believes that neutrophils from ARDS survivors are functionally impaired, it is hard to assume these minor differences may underlie increased sensitivity to infection by post-ARDS patients.

- Key mechanistic analyses are lacking. How is (minor) decrease of enzymes connected with broad H3K4me3 changes? How is hypoxia linked to this phenotype? How are metabolic alterations linked to epigenomic alterations?

- Authors assume hypoxia-induced changes happen in neutrophil progenitors and are translated in differentiated cells. No data is shown to support this claim.

Reviewer #2

(Remarks to the Author)

The manuscript entitled "Hypoxia drives long-term reprogramming of newly formed blood neutrophils with consequence for inflammatory disease states" from Sanchez-Garcia and colleagues describes long-term perturbations in neutrophils following ARDS and hypoxia and associates these changes with altered epigenetic states and impaired effector function. While most analysis focuses on an ARDS cohort, the authors also enrolled a longitudinal altitude-induced hypoxia cohort and relate similar long-term changes to neutrophils in this group compared to ARDS. Another interesting and seemingly novel observation is the author's characterization of altered neutrophil function 6 months following ARDS, by metabolic proteomic studies and bacterial stimulation. However, beyond correlative description of epigenomic features in Fig 3 and 4 (inadequately analyzed and presented and without sufficient supportive data to interpret) the study lacks incisive functional

studies that link either epigenomic or metabolic phenotypes in progenitors (or neutrophils) to the phenotypes they describe.

Major points:

Conclusions drawn from comparing acute ARDS neutrophil programs to steady state should be highly qualified based on how different these cells are. For example, Fig 2 should be a supplemental figure. Any relation of these types of expression changes to overall epigenetic programs is highly speculative, especially given how dramatically different acute ARDS neutrophils are to steady state. It is interesting for the authors to present the data from MLL complex, SUZ12 and PHF2 in the context of the data presented in Fig 3 (combine those with Fig 3 is my suggestion)— these observations relate to a robust and durable phenotype making a putative functional link, while other data are not. The other data can be included in supplemental.

The quality of the ChIPseq data is challenging to interpret in the absence of more extensive QC metrics and the presentation of individual samples (replicates) and presentation of tracks over specific genes (for example, some favorites in the neutrophil degranulation category). In fact the equivalent differences between clinical groups across the metagene profile suggests differences in input or coverage between clinical groups rather than specific differences at promoter/regulatory regions. Statistics should be performed using per individual data for each clinical group and presented for a clear presentation of the within clinical group variance in H3K4me3 and H3K27me3 levels at (i) neutrophil gene sets (ii) a few favorite select gene promoters; (iii) control genes (or some kind of normalization should be described). Examples of unchanged peaks, increased and decreased peaks should be shown if possible including replicates for specific genes and QC metrics should be described in methods and/or presented in supplemental data. It is unclear to this reviewer what the following means, "Genome-wide H3K4me3 and 565 H3K27me3 profiles were generated using per condition merged BAM files..." were the tracks insufficient as replicates and required merging per each condition? The results as shown could be explained by simple differences in coverage across clinical groups attributable to cell viability or other global phenotypes. Normalizations should be employed to control for these potential effects in addition to the above mentioned recommendations (e.g. plot an average profile of 10000 random bins throughout the genome; control regions, etc).

Even if validated by replicates, statistics, normalizations, and extensive presentation of specific genes and programs differentially regulated by H3K4me3 across clinical groups, there is still no causal link between description of this epigenetic program and the phenotypes described. At this level of publication this correlative link should be supported by additional functional (e.g. gain and loss of function studies of MLL pathway and/or PRC2) and/or mechanistic data (more biochemical data linking these chromatin regulatory pathways to specific neutrophil genes in post ARDS conditions).

My understanding is that increased susceptibility of post-ICU patients to infections is well established and does not represent a new insight. If the data presented in Figure 4E are novel, beyond establishing these clinical features in the study's cohort for relation to the cellular and molecular data, the authors should highlight what was known and what additional insights are derived from these data. In general, alterations following sepsis (<https://doi.org/10.1038/s41590-023-01490-5>; published in NI) and covid (<https://doi.org/10.1016/j.cell.2023.07.019>) have established changes in hematopoiesis and neutrophils or their progenitors. In general background on the post ARDS population, what is known in terms of susceptibility to infection, and molecular programs is not well summarized in the introduction or discussion and the authors should improve this to better contextualize their study.

Again, at this level of publication it may be expected for the authors to provide direct evidence of progenitor reprogramming in their human studies by enriching rare circulating HSPC as described in the two studies above or at least extend their mouse studies where these progenitors are readily accessible to study how hypoxia durably reprograms neutrophil function at level of progenitor cells.

TF motif analysis presented in Fig 3F and Table 2 is unconventional and lacks supporting data and conventional visualization of these types of data. In table 2 it is not clear if authors used an adjusted p value, which would be appropriate here.

Minor comments:

"This was not explained by a global loss in histone H3, as ChIPseq analysis for 147 H3K27me3 showed preservation of marks when compared to healthy controls (Extended Data 148 Fig. 1b)."

The appropriate control for this would be H3 ChIP normalization— both a total reduction of H3 and proportional increase in H3K27me3 is possible. Perhaps authors should highlight that loss of histone modifications was not uniform across other PTMs, including an increase in H3K27me3.

typos — more than expected: "Out with [Other than?] changes in emergency haematopoiesis in 59 response to systemic hypoxia, a bias towards granulopoiesis has been reported in the myeloid 60 progenitors in response to BCG vaccination8. "

"We show that early in ARDS 80 mature blood neutrophils early in ARDS have altered abundance of epigenetic modifiers and 81 that in ARDS survivors 3-6 months following hospitalization this is associated with epigenetic 82 rewiring characterised by genome-wide loss of histone 3 lysine 4 trimethylation and impaired 83 neutrophil effector function, impacting on infection

outcomes. “

Reviewer #3

(Remarks to the Author)
Review for NI-A36279

In the submitted manuscript entitled "Hypoxia drives long-term reprogramming of newly formed blood neutrophils with consequence for inflammatory disease states" by Manuel A. Sanchez-Garcia et al., the authors, possessing internationally recognized expertise in the field of hypoxia research, propose that hypoxia triggers sustained epigenetic modifications, particularly a significant reduction in histone 3 lysine 4 trimethylation (H3K4me3), in neutrophils. These alterations result in enduring functional impairments in key neutrophil activities. The study's scope encompasses an examination of data obtained from clinical settings involving hypoxaemic ARDS patients, hypoxia-exposed mice, and healthy volunteers temporarily exposed to high altitudes. The topic is certainly within the scope of Nature Immunology. The cellular and molecular analyses carried out exhibit a state-of-the-art approach. While the conceptual framework is intriguing, the empirical foundation of the study lacks robust support, and the experimental design and data presentation reveal certain limitations.

Major concerns:

- 1. Novelty of the Central Mechanism:** The prevailing understanding that lysine residue methylation of histones orchestrates gene expression and epigenetic transmission within cells has been well-established (Klose RJ et al. *JmJC-domain-containing proteins and histone demethylation. Nat Rev Genet* 2006 Sep;7(9):715-27. doi: 10.1038). Earlier studies, notably those focused on pulmonary ischemia and reperfusion, have delineated a relationship between diminished H3K4me3 methylation and inflammatory responses, apoptosis, and endothelial barrier dysfunction, all of which could be inhibited by dexmedetomidine preconditioning (Hong H et al. *Dexmedetomidine preconditioning ameliorates lung injury induced by pulmonary ischemia/reperfusion by upregulating promoter histone H3K4me3 modification of KGF-2. 2021. Experimental Cell Research* 406(2):112762 DOI 10.1016/j.yexcr.2021.112762). As such, the present study's findings, which demonstrate that altered H3K4me3 profiles translate into enduring innate immune memory, do not yield striking novelty.
- 2. Mechanistic Insights Insufficiency:** The authors acknowledge a noteworthy limitation—namely, the absence of clarity surrounding the mechanisms underpinning the coordination of an epigenetic program following the confluence of systemic hypoxia and an inflammatory/infectious insult (Line 239-241). A salient opportunity exists to experimentally validate the restoration of neutrophil function impairment by modulating or inhibiting H3K4 trimethylation.
- 3. Clinical Cohort and Study Design Ambiguity:** The method and results sections allude to clinical cohorts with reference to study/ethical approval numbers (e.g. AMREC 15-HV012, 20/SS/0002) present in previous work (references 6 and 7). Clarity is warranted regarding the utilization of blood samples derived from prior studies, potentially categorizing the current effort as a follow-up study. Essential details like the primary outcome measures, study registration status, and prospective/post-hoc design classification are conspicuously absent. Furthermore, the time-points for blood collection during hypoxia seem not well-defined. Additionally, concerning line 468, was the neutrophil isolation from peripheral blood performed using a protocol similar to the study of ref. 6 or within the study of ref. 6?
- 4. Model Selection:** The high-altitude-induced hypoxia experiment, while elaborate and resource-intensive, yields limited insight into the proposed mechanism. Was the suppression of H3K4me3 also found in these volunteers? Given the manifold effects triggered by high-altitude exposure on diverse physiological systems, the suitability of comparing these individuals (e.g. alterations in fluid balances, hematopoiesis, innate fluid phase systems, adaptive immunity, thromboinflammatory responses) to ARDS patients poses a challenge. Recommending a validation of the principle mechanism within another hypoxic clinical context, such as post-cardio-pulmonary-resuscitation patients or those with acute myocardial infarction, could enrich the study's robustness. To assess the oxygen debt, blood gas analyses should be compared, especially the base excess of these patients/volunteers during the hypoxia-stress.
- 5. Data Presentation and Handling:** The figures and corresponding data present an intricate narrative that can be challenging to follow. Variability in n-sizes within the same cohort for various parameters, especially in Fig. 1, Fig. 2, and Fig. 4, raises queries about participant selection and the potential application of outlier tests. The rationale for these disparities requires elucidation, including a comprehensive understanding of which patients were integrated into specific graphs. The decision to employ a one-tailed test for Fig. 1e-g necessitates rationale.

Minor Concerns:

- 1. Title Precision:** The latter part of the title, "with consequences for inflammatory disease states," lacks specificity. Including a concise description of these consequences within the title would enhance clarity.
- 2. Purity of neutrophils in the sequencing experiments:** How pure were the neutrophils investigated? Did the authors remove eosinophils before the sequencing analyses were performed? It is well established that eosinophils will significantly alter/compromise the results.
- 3. Precision in Reporting Statistical Values:** Displaying p-values with an extensive number of decimal places, given the

modest n-size, raises statistical validity concerns. A reduction in decimal places could mitigate potential misinterpretations (see e.g. Fig. 2 B: $p=0.0166$ at a n-size of max. 8).

4. Experimental Protocol Adherence: Specifics concerning pain management, adherence to the principles of the 3Rs (replacement, reduction, refinement), and compliance with ARRIVE guidelines for animal research are absent and should be incorporated.

5. Distinction between Hypoxia and Hypoxemia (line 51 "systemic hypoxia (hypoxaemia)..."): A clearer distinction between hypoxia and hypoxemia is necessary, considering the nuanced differences in their definitions.

6. Ethical Clarification: Line 451-457: "This study was performed with informed consent obtained by proxy" – was it informed or written-informed? Line 456: the wording "where appropriate" should be more specified? What does "appropriate" mean?

7. Cautious Interpretation: Instances of overstating known facts, such as describing neutrophil half-life as "exceptional" (line 38) should be tempered for accurate representation. Similar, in line 169 "CD66b showed an elevated abundance of this protein..." should be specified as "slight elevation" and in line 180 "...we detect as specific increase..." should be rather worded "...we detect a minor increase" to avoid overinterpretation of the data.

8. Markers of Activation (line 201): The markers CD66b and CD62L, while indeed indicative of neutrophil activation, are not distinct to "mirror" a post-ARDS profile, given their broader relevance.

9. Repetition in Language: Repetition of "early in ARDS" within a single sentence (line 80-81) should be rectified for linguistic fluidity.

Decision Letter:

14th Sep 2023

Dear Sarah,

We have now finished reviewing your manuscript entitled "Hypoxia drives long-term reprogramming of newly formed blood neutrophils with consequence for inflammatory disease states", reference number NI-A36279.

Although the editors thought that the manuscript was interesting enough to send out for in-depth review, the reviewers were not in favor of publishing the paper in Nature Immunology, with all three noting concerns regarding the appropriateness of the technical approach and the robustness of the conclusions being drawn by the data. We are therefore returning the reviews to you with the hope that you find them useful when you prepare the paper for another journal.

Although we cannot publish your paper, it may be appropriate for another journal in the Nature Portfolio. If you wish to explore the journals and transfer your manuscript please use our manuscript transfer portal. You will not have to re-supply manuscript metadata and files, unless you wish to make modifications. For more information, please see our [manuscript transfer FAQ](http://www.nature.com/authors/author_resources/transfer_manuscripts.html?WT.mc_id=EMI_NPG_1511_AUTHORTRANSF&WT.ec_id=AUTHOR) page.

We realize that this is disappointing. I hope that you continue to consider Nature Immunology for your results most significant for the immunology community and wish you well in your future investigations.

Kind regards,

Laurie

Laurie A. Dempsey, Ph.D.
Senior Editor
Nature Immunology
l.dempsey@us.nature.com
ORCID: 0000-0002-3304-796X

Reviewers' comments:

Reviewer #1 (Remarks to the Author):

In this study, the authors perform molecular and functional analyses of neutrophils from acute ARDS patients or from ARDS survivors, as well as from individuals exposed to high altitudes. They propose that neutrophils in hypoxic conditions exhibit persistent changes in the expression and genomic activities of factors controlling histone methylation, leading to loss of this mark in differentiated cells and to subsequent dysfunction. This is an interesting idea that, in my view, is not sufficiently supported by the data.

As a general comment, this study seems interesting but is preliminary. Several claims are based on correlative evidence or on over-interpretation of the data. I will only focus on those major aspects in this review.

- Do neutrophils from ARDS survivors express lower levels of chromatin regulators? Is loss of H3K4me3 pervasive or selective? If pervasive, this should be quantified using spike-ins to exclude technical artifacts. If selective, unchanged or up-

regulated regions in ARDS patients should be used as reference for differential enrichment analyses. The presented analyses are superficial and poorly informative.

- Functional differences between neutrophils from controls and post-ARDS patients are very small. Data are shown as fold change, which over-emphasize differences between groups; even so, fold change differences are almost negligible and poorly significant. Data in Extended Data Fig. 2h or Fig. 5g show no actual differences in *S. aureus* phagocytosis, contrary to what the authors claim. Even if one believes that neutrophils from ARDS survivors are functionally impaired, it is hard to assume these minor differences may underlie increased sensitivity to infection by post-ARDS patients.

- Key mechanistic analyses are lacking. How is (minor) decrease of enzymes connected with broad H3K4me3 changes? How is hypoxia linked to this phenotype? How are metabolic alteration linked to epigenomic alterations?

- Authors assume hypoxia-induced changes happen in neutrophils progenitors and are translated in differentiated cells. No data is shown to support this claim.

Reviewer #2 (Remarks to the Author):

The manuscript entitled “Hypoxia drives long-term reprogramming of newly formed blood neutrophils with consequence for inflammatory disease states” from Sanchez-Garcia and colleagues describes long-term perturbations in neutrophils follow ARDS and hypoxia and associate these changes with altered epigenetic states and impaired effector function. While most analysis focuses on an ARDS cohort, the authors also enrolled a longitudinal altitude induced hypoxia cohort and relate similar long-term changes to neutrophils in this group compared to ARDS. Another interesting and seemingly novel observation is the author’s characterization of altered neutrophil function 6 months following ARDS, by metabolic proteomic studies and bacterial stimulation. However, beyond correlative description of epigenomic features in Fig 3 and 4l (inadequately analyzed and presented and without sufficient supportive data to interpret) the study lacks incisive functional studies that link either epigenomic or metabolic phenotypes in progenitors (or neutrophils) to the phenotypes they describe.

Major points:

Conclusions drawn from comparing acute ARDS neutrophil programs to steady state should be highly qualified based on how different these cells are. For example, Fig 2 should be a supplemental figure. Any relation of these types of expression changes to overall epigenetic programs is highly speculative, especially given how dramatically different acute ARDS neutrophils are to steady state. It is interesting for the authors to present the data from MLL complex, SUZ12 and PHF2 in the context of the data presented in Fig 3 (combine those with Fig 3 is my suggestion)— these observations relate to a robust and durable phenotype making a putative functional link, while other data are not. The other data can be included in supplemental.

The quality of the ChIPseq data is challenging to interpret in the absence of more extensive QC metrics and the presentation of individual samples (replicates) and presentation of tracks over specific genes (for example, some favorites in the neutrophil degranulation category). In fact the equivalent differences between clinical groups across the metagene profile suggests differences in input or coverage between clinical groups rather than specific differences at promoter/regulatory regions. Statistics should be performed using per individual data for each clinical group and presented for a clear presentation of the within clinical group variance in H3K4me3 and H3K27me3 levels at (i) neutrophil gene sets (ii) a few favorite select gene promoters; (iii) control genes (or some kind of normalization should be described). Examples of unchanged peaks, increased and decreased peaks should be shown if possible including replicates for specific genes and QC metrics should be described in methods and/or presented in supplemental data. It is unclear to this reviewer what the following means, “Genome-wide H3K4me3 and 565 H3K27me3 profiles were generated using per condition merged BAM files... “ were the tracks insufficient as replicates and required merging per each condition? The results as shown could be explained by simple differences in coverage across clinical groups attributable to cell viability or other global phenotypes. Normalizations should be employed to control for these potential effects in addition to the above mentioned recommendations (e.g. plot an average profile of 10000 random bins throughout the genome; control regions, etc).

Even if validated by replicates, statistics, normalizations, and extensive presentation of specific genes and programs differentially regulated by H3K4me3 across clinical groups, there is still no causal link between description of this epigenetic program and the phenotypes described. At this level of publication this correlative link should be supported by additional functional (e.g. gain and loss of function studies of MLL pathway and/or PRC2) and/or mechanistic data (more biochemical data linking these chromatin regulatory pathways to specific neutrophil genes in post ARDS conditions).

My understanding is that increased susceptibility of post-ICU patients to infections is well established and does not represent a new insight. If the data presented in Figure 4E are novel, beyond establishing these clinical features in the study’s cohort for relation to the cellular and molecular data, the authors should highlight what was known and what additional insights are derived from these data. In general, alterations following sepsis (<https://doi.org/10.1038/s41590-023-01490-5>; published in NI) and covid (<https://doi.org/10.1016/j.cell.2023.07.019>) have established changes in hematopoiesis and neutrophils or their progenitors. In general background on the post ARDS population, what is known in terms of susceptibility to infection, and molecular programs is not well summarized in the introduction or discussion and the authors should improve this to better contextualize their study.

Again, at this level of publication it may be expected for the authors to provide direct evidence of progenitor reprogramming in their human studies by enriching rare circulating HSPC as described in the two studies above or at least extend their mouse studies where these progenitors are readily accessible to study how hypoxia durably reprograms neutrophil function at the level of progenitor cells.

TF motif analysis presented in Fig 3F and Table 2 is unconventional and lacks supporting data and conventional visualization of these types of data. In table 2 it is not clear if authors used an adjusted p value, which would be appropriate here.

Minor comments:

“This was not explained by a global loss in histone H3, as ChIPseq analysis for 147 H3K27me3 showed preservaon of marks when compared to healthy controls (Extended Data 148 Fig. 1b).”

The appropriate control for this would be H3 ChIP normalization— both a total reduction of H3 and proportional increase in H3K27me3 is possible. Perhaps authors should highlight that loss of histone modifications was not uniform across other PTMs, including an increase in H3K27me3.

typos — more than expected: “Out with [Other than?] changes in emergency haematopoiesis in 59 response to systemic hypoxia, a bias towards granulopoiesis has been reported in the myeloid 60 progenitors in response to BCG vaccinaon8. “

“We show that early in ARDS 80 mature blood neutrophils early in ARDS have altered abundance of epigenec modifiers and 81 that in ARDS survivors 3-6 months following hospitalizaon this is associated with epigenec 82 rewiring characterised by genome-wide loss of histone 3 lysine 4 trimethylaon and impaired 83 neutrophil effector funcon, impacng on infecon outcomes. “

Reviewer #3 (Remarks to the Author):

Review for NI-A36279

In the submitted manuscript entitled "Hypoxia drives long-term reprogramming of newly formed blood neutrophils with consequence for inflammatory disease states" by Manuel A. Sanchez-Garcia et al., the authors, possessing internationally recognized expertise in the field of hypoxia research, propose that hypoxia triggers sustained epigenetic modifications, particularly a significant reduction in histone 3 lysine 4 trimethylation (H3K4me3), in neutrophils. These alterations result in enduring functional impairments in key neutrophil activities. The study's scope encompasses an examination of data obtained from clinical settings involving hypoxaemic ARDS patients, hypoxia-exposed mice, and healthy volunteers temporarily exposed to high altitudes. The topic is certainly within the scope of Nature Immunology. The cellular and molecular analyses carried out exhibit a state-of-the-art approach. While the conceptual framework is intriguing, the empirical foundation of the study lacks robust support, and the experimental design and data presentation reveal certain limitations.

Major concerns:

1. Novelty of the Central Mechanism: The prevailing understanding that lysine residue methylation of histones orchestrates gene expression and epigenetic transmission within cells has been well-established (Klose RJ et al. *JmJC-domain-containing proteins and histone demethylation. Nat Rev Genet* 2006 Sep;7(9):715-27. doi: 10.1038). Earlier studies, notably those focused on pulmonary ischemia and reperfusion, have delineated a relationship between diminished H3K4me3 methylation and inflammatory responses, apoptosis, and endothelial barrier dysfunction, all of which could be inhibited by dexmedetomidine preconditioning (Hong H et al. *Dexmedetomidine preconditioning ameliorates lung injury induced by pulmonary ischemia/reperfusion by upregulating promoter histone H3K4me3 modification of KGF-2.* 2021. *Experimental Cell Research* 406(2):112762 DOI 10.1016/j.yexcr.2021.112762). As such, the present study's findings, which demonstrate that altered H3K4me3 profiles translate into enduring innate immune memory, do not yield striking novelty.

2. Mechanistic Insights Insufficiency: The authors acknowledge a noteworthy limitation—namely, the absence of clarity surrounding the mechanisms underpinning the coordination of an epigenetic program following the confluence of systemic hypoxia and an inflammatory/infectious insult (Line 239-241). A salient opportunity exists to experimentally validate the restoration of neutrophil function impairment by modulating or inhibiting H3K4 trimethylation.

3. Clinical Cohort and Study Design Ambiguity: The method and results sections allude to clinical cohorts with reference to study/ethical approval numbers (e.g. AMREC 15-HV012, 20/SS/0002) present in previous work (references 6 and 7). Clarity is warranted regarding the utilization of blood samples derived from prior studies, potentially categorizing the current effort as a follow-up study. Essential details like the primary outcome measures, study registration status, and prospective/post-hoc design classification are conspicuously absent. Furthermore, the time-points for blood collection during hypoxia seem not

well-defined. Additionally, concerning line 468, was the neutrophil isolation from peripheral blood performed using a protocol similar to the study of ref. 6 or within the study of ref. 6?

4. Model Selection: The high-altitude-induced hypoxia experiment, while elaborate and resource-intensive, yields limited insight into the proposed mechanism. Was the suppression of H3K4me3 also found in these volunteers? Given the manifold effects triggered by high-altitude exposure on diverse physiological systems, the suitability of comparing these individuals (e.g. alterations in fluid balances, hematopoiesis, innate fluid phase systems, adaptive immunity, thromboinflammatory responses) to ARDS patients poses a challenge. Recommending a validation of the principle mechanism within another hypoxic clinical context, such as post-cardio-pulmonary-resuscitation patients or those with acute myocardial infarction, could enrich the study's robustness. To assess the oxygen debt, blood gas analyses should be compared, especially the base excess of these patients/volunteers during the hypoxia-stress.

5. Data Presentation and Handling: The figures and corresponding data present an intricate narrative that can be challenging to follow. Variability in n-sizes within the same cohort for various parameters, especially in Fig. 1, Fig. 2, and Fig. 4, raises queries about participant selection and the potential application of outlier tests. The rationale for these disparities requires elucidation, including a comprehensive understanding of which patients were integrated into specific graphs. The decision to employ a one-tailed test for Fig. 1e-g necessitates rationale.

Minor Concerns:

1. Title Precision: The latter part of the title, "with consequences for inflammatory disease states," lacks specificity. Including a concise description of these consequences within the title would enhance clarity.
2. Purity of neutrophils in the sequencing experiments: How pure were the neutrophils investigated? Did the authors remove eosinophils before the sequencing analyses were performed? It is well established that eosinophils will significantly alter/compromise the results.
3. Precision in Reporting Statistical Values: Displaying p-values with an extensive number of decimal places, given the modest n-size, raises statistical validity concerns. A reduction in decimal places could mitigate potential misinterpretations (see e.g. Fig. 2 B: $p=0.0166$ at a n-size of max. 8).
4. Experimental Protocol Adherence: Specifics concerning pain management, adherence to the principles of the 3Rs (replacement, reduction, refinement), and compliance with ARRIVE guidelines for animal research are absent and should be incorporated.
5. Distinction between Hypoxia and Hypoxemia (line 51 "systemic hypoxia (hypoxaemia)..."): A clearer distinction between hypoxia and hypoxemia is necessary, considering the nuanced differences in their definitions.
6. Ethical Clarification: Line 451-457: "This study was performed with informed consent obtained by proxy" – was it informed or written-informed? Line 456: the wording "where appropriate" should be more specified? What does "appropriate" mean?
7. Cautious Interpretation: Instances of overstating known facts, such as describing neutrophil half-life as "exceptional" (line 38) should be tempered for accurate representation. Similar, in line 169 "CD66b showed an elevated abundance of this protein..." should be specified as "slight elevation" and in line 180 "...we detect as specific increase..." should be rather worded "...we detect a minor increase" to avoid overinterpretation of the data.
8. Markers of Activation (line 201): The markers CD66b and CD62L, while indeed indicative of neutrophil activation, are not distinct to "mirror" a post-ARDS profile, given their broader relevance.
9. Repetition in Language: Repetition of "early in ARDS" within a single sentence (line 80-81) should be rectified for linguistic fluidity.

Version 1:

Reviewer comments:

Reviewer #1

(Remarks to the Author)

The authors have made significant progress by generating new data and analyses that demonstrate changes in H3K4me3 under hypoxia in mouse models. They also provide intriguing observations regarding histone clipping as a potential mechanism. Overall, the paper has improved, but it still falls short of fully addressing my initial concerns regarding its descriptive nature and functional implications. A key issue remains how the authors can quantify the reduction and non-selective loss of H3K4me3. The data on reduced H3K4me3 in neutrophil progenitors rely on flow cytometry, and the quantitative reliability of this methodology is uncertain. Additionally, the authors present evidence of histone clipping in a single bar plot, which complicates the assessment of its relevance. I would have expected a more thorough analysis of this.

Reviewer #3

(Remarks to the Author)

Review for NI-A36279

Concerning major issues of the original manuscript:

1. Novelty of the Central Mechanism

The authors highlight the long-term persistence of neutrophil functional impairment post-hypoxia/ARDS. The authors now

provide compelling new mechanistic evidence implicating N-terminal histone clipping in neutrophil progenitors as a novel driver of sustained H3K4me3 loss.

2. Mechanistic Insights Insufficiency

The authors acknowledge the initial lack of mechanistic detail and now present new preliminary data indicating that BCG vaccination post-hypoxia partially restores neutrophil H3K4me3 and improves infection outcomes in mice. While these data are not included in manuscript, their inclusion in the reviewer response is acceptable, given the complexity and scope of the study.

3. Ambiguity in Clinical Cohort and Study Design

The authors have adequately clarified the study design, sample provenance, timing, and methodological approaches. The revision resolves prior concerns regarding cohort characterization.

4. Model Selection

In response to concerns about the limited generalizability of the high-altitude model and the need for validation in other hypoxic clinical settings, the authors provide additional ChIP-qPCR data demonstrating H3K4me3 loss in altitude-exposed volunteers. They also justify the focus on murine mechanistic studies. Although additional clinical data (e.g., blood gas analyses or other clinical cohorts) were not provided the rationale is well-argued and reasonable in the light of practical constraints.

5. Data Presentation and Handling

The authors specified the title sufficiently. They clarified sample sizes and reasons for variation (limited cell availability, sequential assays) and stated no prior selection or outlier exclusion.

Concerning minor issues of the original manuscript:

All minor concerns appear to have been adequately addressed.

Overall Conclusion

The authors have provided comprehensive, transparent, and well-reasoned responses to both major and minor reviewer comments. The manuscript has been significantly strengthened by the addition of novel mechanistic data, clarification of study design and methodology, improved data presentation, and careful attention to terminology and ethical considerations. While some experimental validations (e.g., direct reversal of H3K4me3 loss) remain ongoing and are not included in the manuscript, the authors have acknowledged these limitations and provided supportive preliminary supportive data. Therefore, the responses are considered sufficient and appropriate to address the reviewer's concerns and support the manuscript's advancement.

Decision Letter:

23rd Jul 2025

Dear Sarah & Musa

Thank you for providing a point-by-point response to the remaining comments voiced by referee #1 on your revised manuscript entitled, "Hypoxia induces histone clipping and H3K4me3 loss in neutrophil progenitors resulting in long-term impairment of neutrophil immunity". I agree with your plan to include data from "experiments 2-4" as outlined in your revision plan, which include neutrophil H3K4me3 in a murine model of staphylococcal-induced abscess formation following hypoxic exposure and subsequent BCG vaccination, immunoblot analysis of BM neutrophil H3K4me3 abundance, and new analysis of LC-MS datasets for histone H3 peptide abundance. Please include the data from these additional experiments in your revised version of the manuscript.

We therefore invite you to revise your manuscript taking into account all reviewer and editor comments. Please highlight all changes in the manuscript text file in Microsoft Word format.

* If you have not done so already please begin to revise your manuscript so that it conforms to our Article format instructions at <http://www.nature.com/nia/authors/index.html>. Refer also to any guidelines provided in this letter.

* Please include a revised version of any required reporting checklist. It will be available to referees to aid in their evaluation of the manuscript goes back for peer review. They are available here:

Reporting summary:

Please note, Extended Data figures and tables are online-only (appearing in the online PDF and full-text HTML version of the paper), peer-reviewed display items that provide essential background to the Article but are not included in the printed version of the paper due to space constraints or being of interest only to a few specialists. A maximum of ten Extended Data display items (figures and tables) is typically permitted. When re-submitting your manuscript, please ensure that any supplementary figures and tables that are more critical to the manuscript's conclusions are converted to Extended data to increase these data's visibility.

Link Redacted

We hope to receive your revised manuscript within four weeks. If you cannot send it within this time, please let us know. We will be happy to consider your revision so long as nothing similar has been accepted for publication at Nature Immunology or published elsewhere.

Nature Immunology is committed to improving transparency in authorship. As part of our efforts in this direction, we are now requesting that all authors identified as 'corresponding author' on published papers create and link their Open Researcher and Contributor Identifier (ORCID) with their account on the Manuscript Tracking System (MTS), prior to acceptance. ORCID helps the scientific community achieve unambiguous attribution of all scholarly contributions. You can create and link your ORCID from the home page of the MTS by clicking on 'Modify my Springer Nature account'. For more information please visit www.springernature.com/orcid.

Kind regards,

Laurie

Laurie A. Dempsey, Ph.D.
Senior Editor
Nature Immunology
l.dempsey@us.nature.com
ORCID: 0000-0002-3304-796X

Referee expertise:

Referee #1:

Referee #2:

Referee #3:

Reviewers' Comments:

Reviewer #1 (Remarks to the Author):

The authors have made significant progress by generating new data and analyses that demonstrate changes in H3K4me3 under hypoxia in mouse models. They also provide intriguing observations regarding histone clipping as a potential mechanism. Overall, the paper has improved, but it still falls short of fully addressing my initial concerns regarding its descriptive nature and functional implications. A key issue remains how the authors can quantify the reduction and non-selective loss of H3K4me3. The data on reduced H3K4me3 in neutrophil progenitors rely on flow cytometry, and the quantitative reliability of this methodology is uncertain. Additionally, the authors present evidence of histone clipping in a single bar plot, which complicates the assessment of its relevance. I would have expected a more thorough analysis of this.

Reviewer #3 (Remarks to the Author):

Review for NI-A36279

Concerning major issues of the original manuscript:

1. Novelty of the Central Mechanism

The authors highlight the long-term persistence of neutrophil functional impairment post-hypoxia/ARDS. The authors now provide compelling new mechanistic evidence implicating N-terminal histone clipping in neutrophil progenitors as a novel driver of sustained H3K4me3 loss.

Incidental remark: the author's statement in response to reviewer #3 - that they "have now undertaken a 2-year program of work to address this knowledge gap" - may be informative but does not necessarily convey the extent or scientific depth of the progress made.

2. Mechanistic Insights Insufficiency

The authors acknowledge the initial lack of mechanistic detail and now present new preliminary data indicating that BCG vaccination post-hypoxia partially restores neutrophil H3K4me3 and improves infection outcomes in mice.

3. Ambiguity in Clinical Cohort and Study Design

The authors have adequately clarified the study design, sample provenance, timing, and methodological approaches. The revision resolves prior concerns regarding cohort characterization.

4. Model Selection

In response to concerns about the limited generalizability of the high-altitude model and the need for validation in other hypoxic clinical settings, the authors provide additional ChIP-qPCR data demonstrating H3K4me3 loss in altitude-exposed volunteers. They also justify the focus on murine mechanistic studies. Although additional clinical data (e.g., blood gas analyses or other clinical cohorts) were not provided the rationale is well-argued and reasonable in the light of practical constraints.

5. Data Presentation and Handling

The authors specified the title sufficiently. They clarified sample sizes and reasons for variation (limited cell availability, sequential assays) and stated no prior selection or outlier exclusion.

Concerning minor issues of the original manuscript:

All minor concerns appear to have been adequately addressed.

Overall Conclusion

The authors have provided comprehensive, transparent, and well-reasoned responses to both major and minor reviewer comments. The manuscript has been significantly strengthened by the addition of novel mechanistic data, clarification of study design and methodology, improved data presentation, and careful attention to terminology and ethical considerations. While some experimental validations (e.g., direct reversal of H3K4me3 loss) remain ongoing and are not included in the manuscript, the authors have acknowledged these limitations and provided supportive preliminary supportive data.

Therefore, the responses are considered sufficient and appropriate to address the reviewer's concerns and support the manuscript's advancement.

Version 2:

Decision Letter:

Our ref: NI-A36279B

1st Aug 2025

Dear Sarah,

Thank you for submitting your revised manuscript "Hypoxia induces histone clipping and H3K4me3 loss in neutrophil progenitors resulting in long-term impairment of neutrophil immunity" (NI-A36279B). I see that you clarified the text and data shown in figure 6 for the N-terminal cleavage of histone 3 during hypoxic conditions in the neutrophil progenitor subsets, in addition to the other clarifications in response to referee #1. Therefore we'll be happy in principle to publish it in Nature Immunology, pending minor revisions to comply with our editorial and formatting guidelines.

We will now perform detailed checks on your paper and will send you a checklist detailing our editorial and formatting

requirements in about a week. Please do not upload the final materials and make any revisions until you receive this additional information from us.

If you had not uploaded a Word file for the current version of the manuscript, we will need one before beginning the editing process; please email that to immunology@us.nature.com at your earliest convenience.

Thank you again for your interest in Nature Immunology Please do not hesitate to contact me if you have any questions.

Kind regards,

Laurie

Laurie A. Dempsey, Ph.D.
Senior Editor
Nature Immunology
l.dempsey@us.nature.com
ORCID: 0000-0002-3304-796X

Dear Dr. Dempsey,

Thank you for reconsidering our manuscript entitled "Hypoxia induces histone clipping and H3K4me3 loss in neutrophil progenitors resulting in long-term impairment of neutrophil immunity". Guided by the helpful critique provided by the reviewers, we have now undertaken a 2-year program of work in which we identify the mechanisms by which acute exposure to systemic hypoxia sustains long-term reprogramming of the neutrophil compartment. We provide a detailed point by point response to the specific concerns raised by each of the reviewers below. With this new work, we very much hope that our new manuscript can be considered for publication in Nature Immunology.

Although the editors thought that the manuscript was interesting enough to send out for in-depth review, the reviewers were not in favour of publishing the paper in Nature Immunology, with all three noting concerns regarding the appropriateness of the technical approach and the robustness of the conclusions being drawn by the data.

Our original submission provided evidence of long-term perturbations in neutrophil effector function in ARDS survivors and 3-4 months following healthy volunteer exposure to altitude-induced hypoxemia. It lacked detail regarding the global loss of H3K4me3 and the underlying mechanisms. We have expanded our work significantly and now demonstrate that H3K4me3 loss is widespread in neutrophils and results from hypoxia-induced N-terminal histone clipping. Mass spectrometry of sorted bone marrow populations localizes this clipping to a population of self-renewing preNeutrophil precursors downstream of GMPs. These are both highly exciting and novel findings with implications well beyond the clinic.

Reviewers' comments:

Reviewer #1 (Remarks to the Author):

In this study, the authors perform molecular and functional analyses of neutrophils from acute ARDS patients or from ARDS survivors, as well as from individuals exposed to high altitudes. They propose that neutrophils in hypoxic conditions exhibit persistent changes in the expression and genomic activities of factors controlling histone methylation, leading to loss of this mark in differentiated cells and to subsequent dysfunction. This an interesting idea that, in my view, is not sufficiently supported by the data.

As a general comment, thus study seems interesting but is preliminary. Several claims are based on correlative evidence or on over-interpretation of the data. I will only focus on those major aspects in this review.

We agree our original submission lacked mechanistic detail and stopped at the description of loss of H3K4me3 in ARDS survivors. We now provide evidence that i). H3K4me3 is lost in circulating neutrophils following altitude exposure (ChIP-qPCR) (new Fig. 4h-j), ii). loss of H3K4me3 is retained across species in murine models of hypoxic exposure (new Fig. 5g-i) iii). loss of H3K4me3 originates in bone marrow neutrophil committed progenitors downstream of GMP (new Fig. 5f-i), iv). loss of H3K4me3 in preNeutrophils is associated with N-terminal histone clipping (new Fig. 5k).

- Do neutrophils from ARDS survivors express lower levels of chromatin regulators? Is loss of H3K4me3 pervasive or selective? If pervasive, this should be quantified using spike-ins to exclude technical artifacts. If selective, unchanged or up-regulated regions in ARDS patients should be used as reference for differential enrichment analyses. The presented analyses are superficial and poorly informative

We have extended the ChIP-seq methods section on the manuscript to include further details on data processing and normalization steps. There was good input fragment size and distribution across all samples, and data were normalised for fragment size and total read count (new Extended Data Fig. 3a,b). We have also extended our analysis to provide individual gene track data (new Fig. 3e-g) and include evidence of unchanged tracks (new Extended Data Fig. 3d). Although not included within the new manuscript, when we compare our gene track control data to previously published human neutrophil H3K4me3 control data (Moorlag et al., 2020), we see similar gene track profiles between these different control data sets (Figure 1 below). Validation of loss of H3K4me3 in core neutrophil effector genes is also now provided in the post altitude cohort using ChIP-qPCR (including LYZ, RAB3D, CALM1, CALR, IFNAR2, CXCR2) (new Fig. 4h-j, Figure 3 below). Finally, the global loss of H3K4me3 is observed in neutrophil progenitors (proNeu1, proNeu2 and preNeu) by intracellular staining (new Fig. 5g-i), with LC-MS quantification of histone fragments detailing loss of the N-terminal histone tail removing the lysine required for the methylation mark to be present (new Fig. 5k).

Fig. 1: Exemplars of H3K4me3 ChIP-seq gene tracks from healthy donor circulating neutrophil data included in our study compared to pre-BCG vaccinated healthy donor neutrophil data from a previous study (Moorlag et al., 2020).

- Functional differences between neutrophils from controls and post-ARDS patients are very small. Data are shown as fold change, which over-emphasize differences between groups; even so, fold change differences are almost negligible and poorly significant. Data in Extended Data Fig. 2h or Fig. 5g show no actual differences in *S. aureus* phagocytosis, contrary to what the authors claim. Even if one believes that neutrophils from ARDS survivors are functionally impaired, it is hard to assume these minor differences may underlie increased sensitivity to infection by post-ARDS patients.

We respectfully disagree regarding the magnitude of the functional defects we observe. In homeostasis, 100 billion neutrophils are produced per day with a circulating half-life of 6-8 hours. As a highly abundant cell, relatively modest changes at a cellular level will therefore have the capacity to translate into biologically important nonlinear effects when considered across the whole population. We sampled in the region of 40 million cells per donor 3-6 months following recovery from ARDS and 3-4 months following a 7-day exposure period to altitude-induced hypoxemia. The ARDS cohort are a highly heterogenous group with respect to the trigger for the development of ARDS and the presence of comorbidities. That we detect conserved defects in neutrophil function despite this heterogeneity which translate to a longitudinal study of high altitude exposed volunteers and into a murine model system speaks to the biological importance of these observations and the magnitude of the underlying changes we observe in core neutrophil effector genes as a consequence of sustained loss of H3K4me3. We have extended the discussion to include reference to the magnitude of effects observed (see page 13 lines 301-310). In the original submission, we did not show any genetic loci from the ChIP-seq experiments making it difficult to convey how denuded of H3K4me3 neutrophil effector genes are. We now have individual gene track data, summarised for core effector genes below (new Fig. 3e-g, Figure 2 below), quantifying the magnitude of change in H3K4me3 in the ARDS survivors. Functional data is now shown as raw values rather than fold change for neutrophil phagocytosis and granule release (new Fig. 1i, g). Survival data is presented relative to unstimulated baseline (Fig. 1h).

Fig. 2: Individual H3K4me3 gene tracks of core neutrophil effector genes following ChIP-seq analysis of blood neutrophils isolated from healthy controls and ARDS survivors 3-6 months into recovery.

- Key mechanistic analyses are lacking. How is (minor) decrease of enzymes connected with broad H3K4me3 changes? How is hypoxia linked to this phenotype? How are metabolic alteration linked to epigenomic alterations?

We agree. As described above, we now provide a detailed mechanistic dissection where hypoxia induces N-terminal histone cleavage and H3K4me3 loss in neutrophil progenitors resulting in long-term impairment of neutrophil immunity.

- Authors assume hypoxia-induced changes happen in neutrophils progenitors and are translated in differentiated cells. No data is shown to support this claim.

We agree that this was a limitation of the previous submission. Utilisation of a murine hypoxia model has allowed us to study changes within the bone marrow compartment in response to hypoxia and trace the origin of the hypoxia-induced change to self-renewing neutrophil committed progenitors (new Fig. 5). Analysis of bone marrow GMP gene tracks by Cut&Run initially revealed the preservation of H3K4me3 marks in the GMP (new Fig. 5f). Excitingly however, we observe the loss of H3K4me3 to occur in the self-renewing proNeu1, proNeu2 and preNeu pool (new Fig. 5g-i). In keeping with evidence that neutrophil proteases mediate proteolytic clipping of the histone H3 amine tail in monocytes (Cheung et al. 2021), we observe the loss of H3K4me3 to be consequent upon hypoxia induced proteolytic cleavage of the N-terminal histone H3 tail in preNeu (new Fig. 5k).

Reviewer #2 (Remarks to the Author):

The manuscript entitled “Hypoxia drives long-term reprogramming of newly formed blood neutrophils with consequence for inflammatory disease states” from Sanchez-Garcia and colleagues describes long-term perturbations in neutrophils follow ARDS and hypoxia and associate these changes with altered epigenetic states and impaired effector function. While most analysis focuses on an ARDS cohort, the authors also enrolled a longitudinal altitude induced hypoxia cohort and relate similar long-term changes to neutrophils in this group compared to ARDS. Another interesting and seemingly novel observation is the author’s characterization of altered neutrophil function 6 months following ARDS, by metabolic proteomic studies and bacterial stimulation. However, beyond correlative description of epigenomic features in Fig 3 and 4I (inadequately analyzed and presented and without sufficient supportive data to interpret) the study lacks incisive functional studies that link either epigenomic or metabolic phenotypes in progenitors (or neutrophils) to the phenotypes they describe.

We agree. As detailed above in response to reviewer 1, our original submission lacked mechanistic detail and stopped at the description of loss of H3K4me3 in ARDS survivors. We now provide evidence that i). H3K4me3 is lost in circulating neutrophils following altitude exposure (ChIP-qPCR) (new Fig. 4h-j), ii). loss of H3K4me3 is retained across species in murine models of hypoxic exposure (new Fig. 5g-i) iii). loss of H3K4me3 originates in bone marrow neutrophil committed progenitors downstream of GMP (new Fig. 5f-i), iv). loss of H3k4me3 in preNeutrophils is associated with N-terminal histone clipping (new Fig. 5k).

Major points:

Conclusions drawn from comparing acute ARDS neutrophil programs to steady state should be highly qualified based on how different these cells are. For example, Fig 2 should be a supplemental figure. Any relation of these types of expression changes to overall epigenetic programs is highly speculative, especially given how dramatically different acute ARDS neutrophils are to steady state. It is interesting for the authors to present the data from MLL complex, SUZ12 and PHF2 in the context of the data presented in Fig 3 (combine those with Fig 3 is my suggestion)— these observations relate to a robust and durable phenotype making a putative functional link, while other data are not. The other data can be included in supplemental.

We have significantly advanced our mechanistic understanding of the hypoxic reprogramming of bone marrow neutrophil progenitors with consequence for neutrophil effector functions. As a consequence, the bulk of the data to which reviewer 2 refers has been moved to the supplemental results section (Extended Data Fig. 2).

The quality of the ChIP-seq data is challenging to interpret in the absence of more extensive QC metrics and the presentation of individual samples (replicates) and presentation of tracks over specific genes (for example, some favorites in the neutrophil degranulation category). In fact the equivalent differences between clinical groups across the metagene profile suggests differences in input or coverage between clinical groups rather than specific differences at promoter/regulatory regions. Statistics should be performed using per individual data for each clinical group and presented for a clear presentation of the within clinical group variance in H3K4me3 and H3K27me3 levels at (i) neutrophil gene sets (ii) a few favorite select gene promoters; (iii) control genes (or some kind of normalization should be described). Examples of unchanged peaks, increased and decreased peaks

should be shown if possible including replicates for specific genes and QC metrics should be described in methods and/or presented in supplemental data. It is unclear to this reviewer what the following means, “Genome-wide H3K4me3 and 565 H3K27me3 profiles were generated using per condition merged BAM files... “ were the tracks insufficient as replicates and required merging per each condition? The results as shown could be explained by simple differences in coverage across clinical groups attributable to cell viability or other global phenotypes. Normalizations should be employed to control for these potential effects in addition to the above mentioned recommendations (e.g. plot an average profile of 10000 random bins throughout the genome; control regions, etc).

We apologise for the lack of inclusion of QC metrics, which was an oversight, and our limited analysis and presentation of ChIP-seq data sets. As detailed above, we have extended the ChIP-seq methods section on the manuscript to include further details on data processing and normalization steps. There was good input fragment size and distribution across all samples, and data were normalised for fragment size and total read count (new Extended Data Fig. 3a,b). Also, each sample was normalized to its corresponding input control. We have included representation of the average profile of 10000 random bins as suggested (new Extended Data Fig. 3c). We now include a PCA plot and individual gene track data (Figure 2 above) to address within clinical group variance of H3K4me3 (new Fig. 3a, e-g). Also, this revised version of the manuscript includes examples of unchanged tracks (new Extended Data Fig. 3d), as suggested. Although not included within the new manuscript, when we compare our gene track control data to previously published human neutrophil H3K4me3 control data (Moorlag et al., 2020), we see similar gene track profiles between these different control data sets (Figure 1 above). Validation of loss of H3k4me3 in core neutrophil effector genes is provided in the post-altitude cohort using ChIP-qPCR (including LYZ, RAB3D, CALM1, CALR, IFNAR2, CXCR2) (new Fig. 4h-j, Figure 3 below). Finally, the global loss of H3K4me3 is observed in bone marrow neutrophil progenitors (proNeu1, proNeu2 and preNeu) by flow cytometry (new Fig. 5g-i), with LC-MS quantification of histone fragments detailing hypoxia induced proteolytic clipping of the N-terminal histone H3 tail in preNeu (new Fig. 5k).

Fig. 3: Paired longitudinal analysis of H3K4me3 modifications on blood neutrophil effector genes by ChIP-qPCR in volunteers before (Baseline) and 3-4 months after (Post-altitude) a 7-day period of altitude-induced hypoxemia.

Even if validated by replicates, statistics, normalizations, and extensive presentation of specific genes and programs differentially regulated by H3K4me3 across clinical groups, there is still no causal link between description of this epigenetic program and the phenotypes described. At this level of publication this correlative link should be supported by additional functional (e.g. gain and loss of function studies of MLL pathway and/or PRC2) and/or mechanistic data (more biochemical data linking these chromatin regulatory pathways to specific neutrophil genes in post ARDS conditions).

We agree. We now provide evidence that i). H3K4me3 is lost in circulating neutrophils following altitude exposure (ChIP-qPCR) (new Fig. 4h-j), ii). loss of H3K4me3 is retained across species in murine models of hypoxic exposure (new Fig. 5g-i) iii). loss of H3K4me3 originates in bone marrow neutrophil committed progenitors downstream of GMP (new Fig. 5f-i), iv). loss of H3k4me3 in preNeutrophils is associated with N-terminal histone clipping (new Fig. 5k).

My understanding is that increased susceptibility of post-ICU patients to infections is well established and does not represent a new insight. If the data presented in Figure 4E are novel, beyond establishing these clinical features in the study’s cohort for relation to the cellular and molecular data, the authors should highlight what was known and what additional insights are derived from these data. In general, alterations following sepsis (<https://doi.org/10.1038/s41590-023-01490-5>; published in NI) and covid (<https://doi.org/10.1016/j.cell.2023.07.019>) have established changes in hematopoiesis and neutrophils or their progenitors. In general background on the post ARDS population, what is known in terms of susceptibility to infection, and molecular programs is not well summarized in the introduction or discussion and the authors should improve this to better contextualize their study.

We agree, it is known that post-ICU patients demonstrate increased susceptibility to infection. We thought it important to report the increase in bacterial burden we observed within our ARDS survivors’ cohort but have now

more clearly framed this around what is already described in the literature (see page 10, line 320 and page 13, lines 298-300). The references highlighted above have also been incorporated in the revised manuscript (references 28 and 33, page 13, lines 296-298 and page 14, lines 313-315).

Again, at this level of publication it may be expected for the authors to provide direct evidence of progenitor reprogramming in their human studies by enriching rare circulating HSPC as described in the two studies above or at least extend their mouse studies where these progenitors are readily accessible to study how hypoxia durably reprograms neutrophil function at the level of progenitor cells.

Agreed. We have used this to direct the extension of our work to understand where reprogramming is occurring within the bone marrow progenitor pool and elucidate the mechanism by which H3K4me3 loss occurs, as detailed above.

TF motif analysis presented in Fig 3F and Table 2 is unconventional and lacks supporting data and conventional visualization of these types of data. In table 2 it is not clear if authors used an adjusted p value, which would be appropriate here.

We have removed this TF motif analysis as it did not provide any significant mechanistic insights.

Minor points:

“This was not explained by a global loss in histone H3, as ChIPseq analysis for 147 H3K27me3 showed preservation of marks when compared to healthy controls (Extended Data 148 Fig. 1b).”

We have now removed reference to this data and provide additional QC data as detailed above.

The appropriate control for this would be H3 ChIP normalization— both a total reduction of H3 and proportional increase in H3K27me3 is possible. Perhaps authors should highlight that loss of histone modifications was not uniform across other PTMs, including an increase in H3K27me3.

In addition to new QC data (new Extended Data Fig. 3), we also now provide details within our methods section regarding data normalization (see methods page 48, lines 876-884).

typos — more than expected: “Out with [Other than?] changes in emergency haematopoiesis in 59 response to systemic hypoxia, a bias towards granulopoiesis has been reported in the myeloid 60 progenitors in response to BCG vaccination8. “

“We show that early in ARDS 80 mature blood neutrophils early in ARDS have altered abundance of epigenetic modifiers and 81 that in ARDS survivors 3-6 months following hospitalization this is associated with epigenetic 82 rewiring characterised by genome-wide loss of histone 3 lysine 4 trimethylation and impaired 83 neutrophil effector function, impacting on infection outcomes. “

Sincere apologies, but we can't see these same typos highlighted above in our submitted version of the manuscript. We have corrected the “out with” to “besides” and have avoided repetition of “early in ARDS”.

Reviewer #3 (Remarks to the Author):

Review for NI-A36279

In the submitted manuscript entitled "Hypoxia drives long-term reprogramming of newly formed blood neutrophils with consequence for inflammatory disease states" by Manuel A. Sanchez-Garcia et al., the authors, possessing internationally recognized expertise in the field of hypoxia research, propose that hypoxia triggers sustained epigenetic modifications, particularly a significant reduction in histone 3 lysine 4 trimethylation (H3K4me3), in neutrophils. These alterations result in enduring functional impairments in key neutrophil activities. The study's scope encompasses an examination of data obtained from clinical settings involving hypoxaemic ARDS patients, hypoxia-exposed mice, and healthy volunteers temporarily exposed to high altitudes. The topic is certainly within the scope of Nature Immunology. The cellular and molecular analyses carried out exhibit a state-of-the-art approach. While the conceptual framework is intriguing, the empirical foundation of the study lacks robust support, and the experimental design and data presentation reveal certain limitations.

We acknowledge that this reviewer recognised the potential importance of the conceptual framework we presented at first submission and agree that our original submission lacked detail regarding the global loss of

H3K4me3 and the underlying mechanisms. As described above, we now provide evidence that i). H3K4me3 is lost in circulating neutrophils following altitude exposure (ChIP-qPCR) (new Fig. 4h-j), ii). loss of H3K4me3 is retained across species in murine models of hypoxic exposure (new Fig. 5g-i) iii). loss of H3K4me3 originates in bone marrow neutrophil committed progenitors downstream of GMP (new Fig. 5f-i), iv). loss of H3k4me3 in preNeutrophils is associated with N-terminal histone clipping (new Fig. 5k).

Major concerns:

1. Novelty of the Central Mechanism: The prevailing understanding that lysine residue methylation of histones orchestrates gene expression and epigenetic transmission within cells has been well-established (Klose RJ et al. JmjC-domain-containing proteins and histone demethylation. *Nat Rev Genet* 2006 Sep;7(9):715-27. doi: 10.1038). Earlier studies, notably those focused on pulmonary ischemia and reperfusion, have delineated a relationship between diminished H3K4me3 methylation and inflammatory responses, apoptosis, and endothelial barrier dysfunction, all of which could be inhibited by dexmedetomidine preconditioning (Hong H et al. Dexmedetomidine preconditioning ameliorates lung injury induced by pulmonary ischemia/reperfusion by upregulating promoter histone H3K4me3 modification of KGF-2. 2021. *Experimental Cell Research* 406(2):112762 DOI 10.1016/j.yexcr.2021.112762). As such, the present study's findings, which demonstrate that altered H3K4me3 profiles translate into enduring innate immune memory, do not yield striking novelty.

The novelty of our initial work was in describing long-term perturbations in neutrophil effector functions that extended beyond the duration of a hypoxic insult both in ARDS survivors and in a longitudinal study of volunteers exposed to 7 days of altitude-induced hypoxemia. We appreciate that we lacked understanding of the mechanisms responsible for these long-term perturbations of neutrophil function in our original submission.

2. Mechanistic Insights Insufficiency: The authors acknowledge a noteworthy limitation—namely, the absence of clarity surrounding the mechanisms underpinning the coordination of an epigenetic program following the confluence of systemic hypoxia and an inflammatory/infectious insult (Line 239-241). A salient opportunity exists to experimentally validate the restoration of neutrophil function impairment by modulating or inhibiting H3K4 trimethylation.

As detailed above, our initial submission lacked mechanistic insights and thus, guided by reviewer comments, we have undertaken a substantial program of work to advance our understanding, given the biological importance of our original observation. This detail is provided in the opening paragraph above.

REDACTED

3. Clinical Cohort and Study Design Ambiguity: The method and results sections allude to clinical cohorts with reference to study/ethical approval numbers (e.g. AMREC 15-HV012, 20/SS/0002) present in previous work (references 6 and 7). Clarity is warranted regarding the utilization of blood samples derived from prior studies, potentially categorizing the current effort as a follow-up study. Essential details like the primary outcome measures, study registration status, and prospective/post-hoc design classification are conspicuously absent. Furthermore, the time-points for blood collection during hypoxia seem not well-defined. Additionally, concerning line 468, was the neutrophil isolation from peripheral blood performed using a protocol similar to the study of ref.

6 or within the study of ref. 6?

We apologise for this lack of detail and have now provided additional clarity re study sampling in the methods section (page 41, lines 712-735 and page 43, lines 751-752). Of note, all samples included in this work were freshly collected under the existing ethical approvals listed, with the primary objective to assess neutrophil properties in acute disease and convalescence. In acute disease, blood samples were taken within 7 days of patients meeting Berlin criteria definition of moderate/severe ARDS. Neutrophil isolation was performed using dextran/ percoll isolation as described in reference 6.

4. Model Selection: The high-altitude-induced hypoxia experiment, while elaborate and resource-intensive, yields limited insight into the proposed mechanism. Was the suppression of H3K4me3 also found in these volunteers? Given the manifold effects triggered by high-altitude exposure on diverse physiological systems, the suitability of comparing these individuals (e.g. alterations in fluid balances, hematopoiesis, innate fluid phase systems, adaptive immunity, thromboinflammatory responses) to ARDS patients poses a challenge. Recommending a validation of the principle mechanism within another hypoxic clinical context, such as post-cardio-pulmonary-resuscitation patients or those with acute myocardial infarction, could enrich the study's robustness. To assess the oxygen debt, blood gas analyses should be compared, especially the base excess of these patients/volunteers during the hypoxia-stress.

We have new longitudinal ChIP-qPCR data for core neutrophil effector genes (including LYZ, RAB3D, CALM1, CALR, IFNAR2, CXCR2) detailing loss of H3K4me3 in the altitude cohort (new Fig. 4h-j and Figure 3 above). In addition, we provide new evidence in murine models of hypoxia of loss of H3K4me3 in bone marrow neutrophil progenitors (proNeu1, proNeu2 and preNeu) using intracellular staining (new Fig 5g-i). We chose to expand work in a murine model system rather than extend to different clinical cohorts as this allowed us to dissect the mechanisms by which hypoxia induces epigenetic alterations in neutrophil progenitors within the bone marrow with consequence for infection outcomes. This would not have been possible with clinical samples.

5. Data Presentation and Handling: The figures and corresponding data present an intricate narrative that can be challenging to follow. Variability in n-sizes within the same cohort for various parameters, especially in Fig. 1, Fig. 2, and Fig. 4, raises queries about participant selection and the potential application of outlier tests. The rationale for these disparities requires elucidation, including a comprehensive understanding of which patients were integrated into specific graphs. The decision to employ a one-tailed test for Fig. 1e-g necessitates rationale.

We have tried to clarify the narrative to make it easier to follow. We also apologise for any initial confusion regarding n numbers. Data is presented as individual data points, with consistent n numbers across data sets reported. Due to limited sample volumes, cell availability and high demands on cell numbers for "omics assays" it was not possible to conduct every assay on each patient sampled. Assays were performed sequentially as samples became available / cell number permitting with no prior selection (see methods page 42, lines 732-735). We provide a table detailing patient demographic data including aetiology of ARDS. We acknowledge that the ARDS cohort are a highly heterogeneous group with respect to the trigger for the development of ARDS and the presence of comorbidities. That we detect conserved defects in neutrophil function despite this heterogeneity which translate to a longitudinal study of altitude exposed volunteers and into a murine model system speaks to the biological importance of these observations and the magnitude of the underlying changes we observe in core neutrophil effector genes as a consequence of sustained loss of H3K4me3. We have now included reference to this heterogeneity within the discussion (page 13, lines 301-310). This new version of the manuscript does not include one-tailed tests.

Minor Concerns:

1. Title Precision: The latter part of the title, "with consequences for inflammatory disease states," lacks specificity. Including a concise description of these consequences within the title would enhance clarity.

We have amended our title in light of our new mechanistic insights.

2. Purity of neutrophils in the sequencing experiments: How pure were the neutrophils investigated? Did the authors remove eosinophils before the sequencing analyses were performed? It is well established that eosinophils will significantly alter/compromise the results.

Neutrophils demonstrated greater than 95% purity following dextran percoll isolation.

3. Precision in Reporting Statistical Values: Displaying p-values with an extensive number of decimal places, given the modest n-size, raises statistical validity concerns. A reduction in decimal places could mitigate potential misinterpretations (see e.g. Fig. 2 B: $p=0.0166$ at a n-size of max. 8).

We apologise and now report significance to 3 decimal places.

4. Experimental Protocol Adherence: Specifics concerning pain management, adherence to the principles of the 3Rs (replacement, reduction, refinement), and compliance with ARRIVE guidelines for animal research are absent and should be incorporated.

We have added new methods text, detailing our compliance with the ARRIVE guidelines and adherence to the principles of the 3Rs (replacement, reduction, refinement) (methods section page 42, lines 738-743).

5. Distinction between Hypoxia and Hypoxemia (line 51 “systemic hypoxia (hypoxaemia)...”): A clearer distinction between hypoxia and hypoxemia is necessary, considering the nuanced differences in their definitions.

We have tried to use hypoxemia to reference a reduction in arterial oxygenation and systemic hypoxia to reference the conditions to which the relevant organism is exposed.

6. Ethical Clarification: Line 451-457: “This study was performed with informed consent obtained by proxy” – was it informed or written-informed? Line 456: the wording “where appropriate” should be more specified? What does “appropriate” mean?

This was an oversight, details of how consent was obtained are included in the methods section (page 41, lines 724-729)

7. Cautious Interpretation: Instances of overstating known facts, such as describing neutrophil half-life as “exceptional” (line 38) should be tempered for accurate representation. Similar, in line 169 “CD66b showed an elevated abundance of this protein...” should be specified as “slight elevation” and in line 180 “...we detect as specific increase...” should be rather worded “..we detect a minor increase” to avoid overinterpretation of the data.

Apologies, we have removed the word exceptional. We have added the word “modest” to temper the interpretation of our results.

8. Markers of Activation (line 201): The markers CD66b and CD62L, while indeed indicative of neutrophil activation, are not distinct to “mirror” a post-ARDS profile, given their broader relevance.

We have reworded this text.

9. Repetition in Language: Repetition of “early in ARDS” within a single sentence (line 80-81) should be rectified for linguistic fluidity.

We have also addressed this repetition to improve linguistic fluidity.

Dear Dr Dempsey,

Thank you for the opportunity to revise our manuscript and address the comments from the referees. Please find below a point-by-point response in which we include new experimental data confirming by immunoblot the loss of H3K4me3, neutrophil H3K4me3 expression in a murine model of staphylococcal-induced abscess formation following hypoxic exposure and subsequent BCG vaccination and new analysis of LC-MS datasets for histone H3 peptide abundance. I very much hope that this allays the specific points raised by reviewer #1 in this regard.

Reviewer #1

(Remarks to the Author)

The authors have made significant progress by generating new data and analyses that demonstrate changes in H3K4me3 under hypoxia in mouse models. They also provide intriguing observations regarding histone clipping as a potential mechanism. Overall, the paper has improved, but it still falls short of fully addressing my initial concerns regarding its descriptive nature and functional implications.

We thank the reviewer for acknowledging our progress. We would like to highlight that we have presented evidence for the widespread loss of H3K4me3 in neutrophils across multiple models. We have validated this finding using four different techniques:

- ChIP-seq in ARDS patients (n=3-4 per group; Fig. 3).
- ChIP-qPCR in a longitudinal high-altitude cohort (n=8; Fig. 4 h-j).
- Flow Cytometry in a mouse model of systemic hypoxia (n=3; Fig. 5g-i).
- LC-MS, which revealed the mechanism: clipping of the N-terminal histone tail (n=4 with duplicates, New Fig. 6a-d).

Our LC-MS data show that the histone tail is physically removed. This clipping eliminates the methylation site, proving the loss of H3K4me3 is a biological reality, not a ChIP-seq loading artifact. We also now include new data which confirms that the loss of H3K4me3 is not a consequence of changes in total H3 (Rebuttal Fig. 1, New Fig. 6c).

Rebuttal Fig. 1: Abundance of histone 3 (H3) detected by LC-MS analysis of bone marrow preNeu (Ly6G⁺Lin⁻SiglecF⁻CD11b⁺cKit⁺Gr1⁺CD34⁻) isolated from mice exposed to 1 week of 10 % O₂ (Hpx), 10% O₂ with LPS challenge (HpxLPS), or normoxia controls (Nmx). Bone marrow harvested preNeu were subjected to chymotrypsin digestion prior to LC-MS quantification. Each data point represents a mouse. Data shown as mean ± s.d. No significant difference by 2-way ANOVA.

With regard to the functional implications of our work, I would like to highlight that we have extensively characterized the resulting defects in core neutrophil functions. Specifically, we have demonstrated significant impairments in neutrophil:

- Activation (Fig. 1c,d; Fig. 4d)
- Metabolism (Fig. 1e; Extended Data Fig. 1b,c)
- Granule Content & Release (Fig. 1f,g; Extended Data Fig. 1g; Fig. 4e,f; Extended data Fig. 4c)
- Cell Survival (Fig. 1h)
- Phagocytic Capacity (Fig. 1i; Fig. 4g)

Crucially, we have linked these functional defects directly to the loss of H3K4me3 at genes regulating granule synthesis, calcium signalling, and inflammatory pathways (Fig. 3e-g). To further strengthen the functional link, we have added the gain-of-function experiment from the original rebuttal to the revised version of the manuscript. In this experiment, BCG vaccination after hypoxic exposure improved infection outcomes and partially restored H3K4me3 in a murine abscess model (Rebuttal Fig. 2, New Fig. 6e-g).

Rebuttal Fig. 2: **a**, Mice were exposed to hypoxic lung injury with LPS nebulisation followed by 1 week of hypoxia (10 % O₂, HpxLPS). Following return to normoxia, mice were vaccinated with BCG (HpxLPS + ReNm BCG) or PBS control (HpxLPS + ReNm veh) on either day 8 (circle) or day 21 (triangle) and after 5 weeks recovery challenged with a subcutaneous infection of *S. aureus*. **b**, Abscess colony forming units (CFU) counts were performed after 48 h of infection. **c**, H3K4me3 MFI relative to H3 MFI in Ly6G⁺ circulating neutrophils were quantified by flow cytometry. Each data point represents one mouse. Data as mean ± s.d. Significant p-values depicted and obtained by Shapiro-Wilk normality test followed by two-tailed t-test.

A key issue remains how the authors can quantify the reduction and non-selective loss of H3K4me3. The data on reduced H3K4me3 in neutrophil progenitors rely on flow cytometry, and the quantitative reliability of this methodology is uncertain.

With respect to the H3K4me3 flow cytometry data, we ensured the quantitative reliability of this data using the following methodology:

- We stained murine bone marrow cells with flow cytometry validated antibodies for H3K4me3 and total H3.
- We used a precise gating strategy to identify specific low-abundance neutrophil progenitors (proNeu1, proNeu2, and preNeu), see representative gating plots below (Rebuttal Fig. 3; Extended Data Fig. 7c).
- We applied compensation and fluorescence-minus-one (FMO) controls to correct for signal overlap and background.

This is an established method for quantifying protein levels in rare cell populations (Gratama et al., 1998 PMID: 9773877).

Rebuttal Fig. 3: Pseudocolor plots containing gates depicting the sequential gating strategy followed to identify mouse bone marrow proNeu1, proNeu2 and preNeu neutrophil committed progenitors and to sort highly pure preNeu for subsequent LC-MS analysis. Percentages refer to Alive/Lin- gates.

To further validate the flow cytometry findings, we now provide new Western blot data in the revised manuscript. This experiment confirms the loss of H3K4me3 in mature bone marrow neutrophils after hypoxic lung injury (Rebuttal Fig. 4; n=3, New Fig. 5d,e).

Rebuttal Fig. 4: a, Mature bone marrow neutrophils were harvested from mice exposed to hypoxic lung injury with LPS nebulisation followed by 1 week of hypoxia (10 % O₂) and 5 weeks recovery in normoxia (HpxLPS + ReNm) or 6 weeks of normoxia (Nm). b, Quantification of H3K4me3 levels in these neutrophils by western blot normalized by total histone content. Each data point represents a mouse. Data as mean ± s.d. Significant p-values depicted (p<0.05) and obtained by Shapiro-Wilk normality test followed by two-tailed Mann-Whitney test.

Additionally, the authors present evidence of histone clipping in a single bar plot, which complicates the assessment of its relevance. I would have expected a more thorough analysis of this.

We agree that the bar graph in Figure 5 was insufficient. We have revised the figure to better illustrate our findings and provide a more comprehensive view of the data (New Fig. 6a-d).

The updated figure shows a new analysis of our existing LC-MS data, including

- A histogram of all histone peptides identified post-digestion.
- A quantification of total histone 3 abundance.
- A comparative histogram of N-terminal peptide abundance from hypoxic vs. normoxic mice.
- We have also extended our initial data sets to include a cohort of hypoxic lung injury challenged mice (LPS nebulisation followed by 1 week of hypoxia (10 % O₂)) to determine the relative contribution of hypoxia to our findings in ARDS mouse models.

We provide a proposed layout for the revised Figure 6 below (Rebuttal Fig. 5).

Rebuttal Fig. 5: **a**, Bone marrow preNeu (Ly6G⁻Lin⁻SiglecF⁻CD11b⁺cKit⁺Gr1⁺CD34⁻) were isolated by FACS from mice exposed to normoxia (Nmx, 21% O₂), hypoxia alone (Hpx, 10 % O₂) or hypoxic lung injury with LPS nebulisation followed by 1 week of hypoxia (10 % O₂, HpxLPS). Cells were lysed and chymotrypsin digestion was employed to ensure the generation of N-terminal histone peptides for subsequent LC-MS quantification. **b**, Quantification of the total number of histone peptides identified. **c**, Total absolute histone 3 intensity (H3) detected. **d**, Abundance of N-terminal H3 peptides normalized by total H3 at different peptide start sites within the H3 protein sequence in **(a)**. Highlighted region in red depicts the fraction containing the tri-methylation site lost in hypoxic conditions. Each mouse represents a data point (**b,c**) or n = 4 mice per experimental group (**d**). Data as mean ± s.d. Significant p-values depicted (for p<0.05) and obtained by 2-way ANOVA.

We will retain the essential control data for histone 2B, which shows no change in N-terminal abundance (Extended Data Fig. 5g) and will incorporate quantification of total histone 2 abundance for consistency (Rebuttal Fig. 6, New Extended Data Fig. 5f).

Rebuttal Fig. 6: Abundance of histone 2 (H2) detected by LC-MS analysis of bone marrow preNeu (Ly6G⁻Lin⁻SiglecF⁻CD11b⁺cKit⁺Gr1⁺CD34⁻) isolated from mice exposed to 1 week of 10 % O₂ (Hpx), 10% O₂ with LPS challenge (HpxLPS), or normoxia controls (Nmx). Bone marrow harvested preNeu were subjected to chymotrypsin digestion prior to LC-MS quantification. Each data point represents a mouse. Data shown as mean ± s.d. No significant difference by 2-way ANOVA.

Reviewer #3

(Remarks to the Author)
Review for NI-A36279

Concerning major issues of the original manuscript:

1. Novelty of the Central Mechanism

The authors highlight the long-term persistence of neutrophil functional impairment post-hypoxia/ARDS. The authors now provide compelling new mechanistic evidence implicating N-terminal histone clipping in neutrophil progenitors as a novel driver of sustained H3K4me3 loss.

2. Mechanistic Insights Insufficiency

The authors acknowledge the initial lack of mechanistic detail and now present new preliminary data indicating that BCG vaccination post-hypoxia partially restores neutrophil H3K4me3 and improves infection outcomes in mice. While these data are not included in manuscript, their inclusion in the reviewer response is acceptable, given the complexity and scope of the study.

3. Ambiguity in Clinical Cohort and Study Design

The authors have adequately clarified the study design, sample provenance, timing, and methodological approaches. The revision resolves prior concerns regarding cohort characterization.

4. Model Selection

In response to concerns about the limited generalizability of the high-altitude model and the need for validation in other hypoxic clinical settings, the authors provide additional ChIP-qPCR data demonstrating H3K4me3 loss in altitude-exposed volunteers. They also justify the focus on murine mechanistic studies. Although additional clinical data (e.g., blood gas analyses or other clinical cohorts) were not provided the rationale is well-argued and reasonable in the light of practical constraints.

5. Data Presentation and Handling

The authors specified the title sufficiently. They clarified sample sizes and reasons for variation (limited cell availability, sequential assays) and stated no prior selection or outlier

exclusion.

Concerning minor issues of the original manuscript:

All minor concerns appear to have been adequately addressed.

Overall Conclusion

The authors have provided comprehensive, transparent, and well-reasoned responses to both major and minor reviewer comments. The manuscript has been significantly strengthened by the addition of novel mechanistic data, clarification of study design and methodology, improved data presentation, and careful attention to terminology and ethical considerations.

While some experimental validations (e.g., direct reversal of H3K4me3 loss) remain ongoing and are not included in the manuscript, the authors have acknowledged these limitations and provided supportive preliminary supportive data.

Therefore, the responses are considered sufficient and appropriate to address the reviewer's concerns and support the manuscript's advancement.

We would like to acknowledge and thank reviewer 3 for their comments.